# Lottery Prior: Randomized Neural Compression for Zero-Shot Inverse Problems

Haotian Wu [1 2]   Di You [2]   Pier Luigi Dragotti [2]   Deniz Gündüz [2]

## Abstract

We study zero-shot inverse problems, where a clean signal is recovered from a single degraded observation without external training data. Contrary to the common belief that such problems require highly complex models, we show that a lightweight neural network, when combined with entropy and complexity regularization in a compression-based formulation, is sufficient for high-quality restoration. We propose Lottery Prior, a compression-based inverse solver that leverages architectural priors from random networks and induces a family of implicit priors through randomness, enabling ensemble-based refinement. We further derive non-asymptotic error bounds for compression-based maximum-likelihood inverse solvers, revealing how rate–distortion constraints act as implicit regularizers. Experiments on denoising, noisy super-resolution, and inpainting demonstrate that our method achieves state-of-the-art with significantly fewer effective parameters. Project page: https://eedavidwu.github.io/LotteryPrior/

## 1. Introduction

In an inverse problem, one aims to recover an unknown signal $\mathbf{x}$ from a degraded observation $\mathbf{y} = \mathcal{A}(\mathbf{x}, \mathbf{z})$, where $\mathcal{A}(\cdot)$ denotes a degradation operator and $\mathbf{z}$ is the noise component. Many inverse problems, such as denoising, super-resolution, and inpainting, can be formulated as energy minimization problems (see Fig. 2):

$$\textbf{P1:} \quad \mathbf{x}^* = \arg\min_{\hat{\mathbf{x}}} E(\hat{\mathbf{x}}, \mathbf{y}) + \lambda R(\hat{\mathbf{x}}), \quad (1)$$

where $E(\hat{\mathbf{x}}, \mathbf{y})$ is a task-dependent fidelity term, and $R(\hat{\mathbf{x}})$

[1]College of Electrical Engineering, Zhejiang University, Hangzhou, 310027, China. [2]Department of Electrical and Electronic Engineering, Imperial College London, London SW7 2AZ, U.K. Correspondence to: Di You <dy22@ic.ac.uk>.

*Proceedings of the 43rd International Conference on Machine Learning*, Seoul, South Korea. PMLR 306, 2026. Copyright 2026 by the author(s).

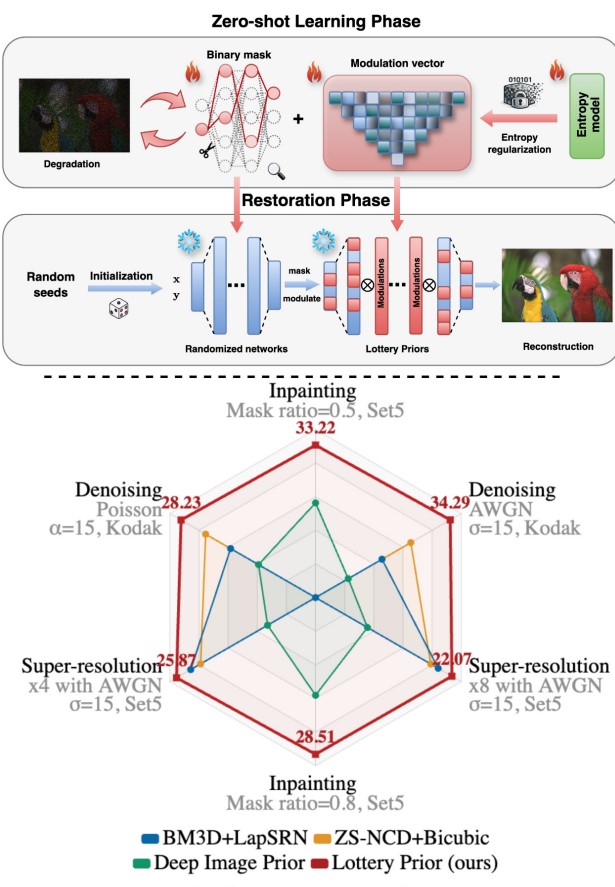

*Figure 1.* **Up**: Illustration of Lottery Priors for zero-shot inverse problems. During the learning phase, the degraded observation is fitted with a neural network regularized by a binary mask and entropy-constrained latent modulations. At restoration, a modulated subnetwork of a randomly initialized network reconstructs the image. Varying mask ratio and random initialization induce a family of lottery priors, enabling further ensemble-based refinement. **Down**: Performance comparison of different tasks, where Lottery Prior achieves state-of-the-art performance.

is a regularizer with hyper-parameter $\lambda$ that captures prior knowledge about the generic structure of the source.

Conventional inverse problem solvers rely on handcrafted models tailored to the forward operator, but can produce unnatural solutions in ill-posed settings (Hegde, 2018). Recent advances in representation learning, including genera-

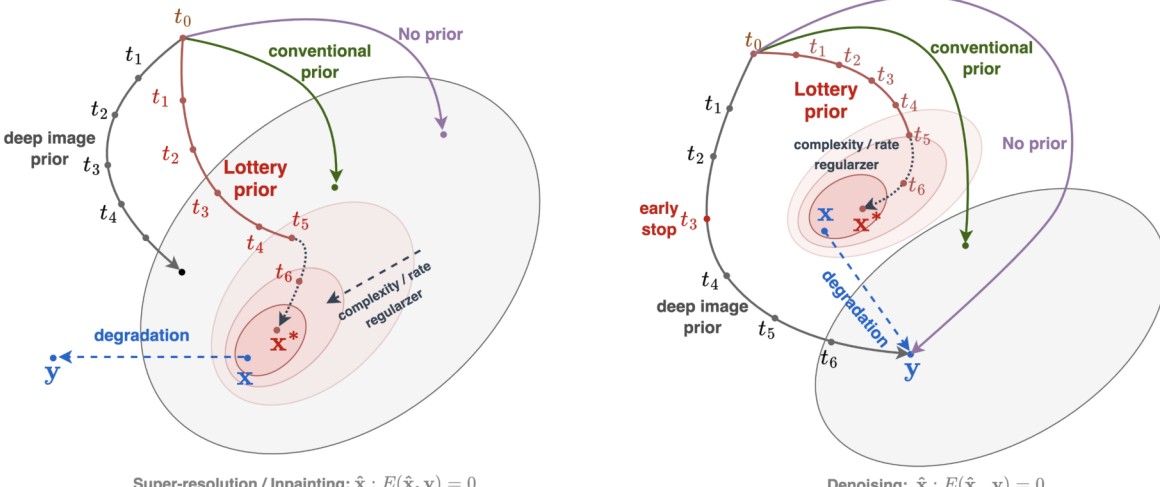

*Figure 2.* **Inverse problem with priors**. We consider reconstructing $\mathbf{x}$ from an observation $\mathbf{y}$. **Left**: For problems such as super-resolution or inpainting, multiple solutions satisfy $E(\hat{\mathbf{x}}, \mathbf{y}) = 0$ (gray region with null energy). Without a strong prior, optimization may converge far from the ground truth. Conventional priors and DIP guide the solution toward $\mathbf{x}$, while the proposed *lottery prior* further regularizes this solution via rate and complexity constraints. **Right**: For denoising, the ground truth has non-zero cost $E(\hat{\mathbf{x}}, \mathbf{y}) \approx \sigma^2$, where $\sigma^2$ is the noise variance. While DIP can reach low-energy solutions with extensive optimization, it tends to overfit and requires early stopping to recover $\mathbf{x}$. In contrast, *lottery priors* provide additional regularization that mitigates overfitting without early stopping. Moreover, inherent randomness induces multiple solutions near the ground truth, enabling ensemble-based refinement in both cases.

tive models (Asim et al., 2020), transformers (Zamir et al., 2022), and diffusion models (Chung et al., 2023), have substantially improved reconstruction performance by learning powerful priors. However, despite their success on standard benchmarks, these methods face significant practical challenges in terms of *data efficiency* and *deployment complexity* (Qayyum et al., 2022). They typically require large-scale paired training data and computationally intensive models, and are sensitive to distribution shifts, often necessitating retraining and data recollection, which limits practical deployment. To reduce reliance on paired data, self-supervised methods learn restorations directly from noisy observations without clean ground truth (Soltanayev & Chun, 2018; Batson & Royer, 2019). However, they generally require large noisy datasets, which can be costly or impractical to collect in scenarios such as medical imaging or remote sensing. This has motivated growing interest in zero-shot inverse solvers, which recover clean signals from a single degraded observation by enforcing measurement consistency while favoring solutions on the natural signal manifold.

A notable example is compression-based zero-shot denoising, which naturally instantiates this principle by exploiting the greater compressibility of clean signals relative to noise (Donoho, 2002). By restricting reconstructions to a codebook, compression induces a structure prior without external training. Formally, lossy compression optimizes a rate–distortion (RD) trade-off (Cover, 1999):

$$\mathbf{P2:} \quad \min D(\hat{\mathbf{x}}, \mathbf{x}) + \lambda R, \qquad (2)$$

where $D(\cdot)$ denotes a distortion measure and $R(\cdot)$ is the

corresponding rate term, with $\lambda$ controlling the trade-off. Viewed through Eq. (1), and taking the noisy observation $\mathbf{y}$ as the compression target, Eq. (2) can be interpreted as a zero-shot denoising solver, with the rate term acting as an implicit regularizer that favors structured, natural solutions. Recent work (Zafari et al., 2026) further demonstrates this potential by training neural compression models directly on patches from a single noisy image. However, this autoencoder-based and patch-driven approach introduces substantial complexity and limits the use of global image context, which often leads to suboptimal reconstructions. Moreover, without additional priors, such methods are largely limited to denoising and do not readily extend to more general inverse problems.

Contrary to the common belief that zero-shot inverse problems require complex models to learn multi-scale features, we show that a lightweight neural network with entropy constraints and flexible compression parameterizations is sufficient for high-quality restoration. This suggests the potential for an inverse solver that is simultaneously *good* (reconstruction quality), *fast* (inference speed), and *cheap* (restoration mechanism), exemplifying a clear "*less is more*" paradigm. Concretely, our approach is motivated by two observations: (i) untrained neural networks inherently encode useful structural priors through their architectures, as evidenced by deep image prior (DIP) (Ulyanov et al., 2020), and (ii) random networks can serve as low-cost, flexible implicit codecs (Wu et al., 2025).

Building on these insights, we propose *Lottery Prior*, a

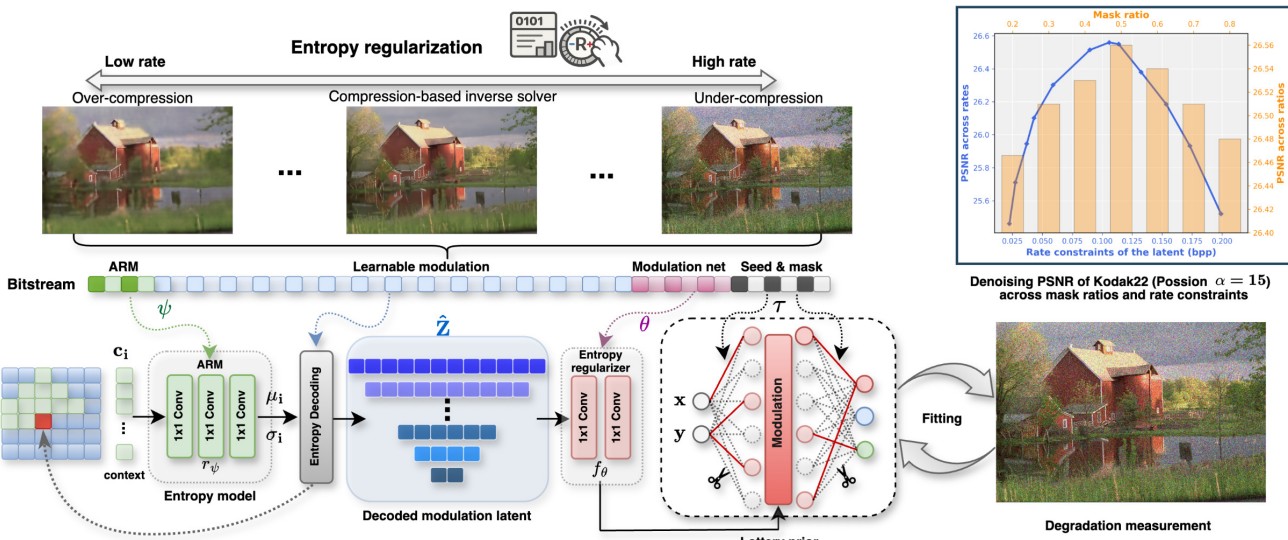

*Figure 3.* Illustration of the image restoration process in *Lottery Prior*. Each degraded observation is fitted using an implicit codec parameterized by a learnable latent and a subnetwork of a randomized neural network. An entropy model imposes rate constraints on the latent, enabling a well-calibrated rate–distortion trade-off for effective restoration. At inference, the binary mask and random initialization seed configure the synthesis network, while latent-driven modulations guide the image reconstruction. Empirically, a mask ratio of 50% provides a strong architectural prior with favorable statistical encoding capacity and improved restoration performance.

randomized neural compression-based inverse solver that combines architectural priors with entropy and complexity regularization. The inherent randomness induces a family of implicit priors whose ensemble can further enhance restoration performance. Specifically, this is formulated as:

$$\textbf{P3:} \quad (\mathbf{Z}^*, \boldsymbol{\tau}^*) = \arg\min_{\mathbf{Z}, \boldsymbol{\tau}} \Bigg[ \underbrace{D(\mathcal{A}(g_{\mathbf{w}_0 \odot \boldsymbol{\tau}}(\mathbf{Z})), \mathbf{y})}_{\text{Measurement alignment}}$$

$$+ \underbrace{\lambda R(\mathbf{Z})}_{\text{Entropy regularizer}} + \underbrace{\beta J(\tau)}_{\text{Complexity regularizer}} \Bigg], \quad (3)$$

$$\text{with} \quad \hat{\mathbf{x}} = g_{\mathbf{w}_0 \odot \boldsymbol{\tau}^*}(\mathbf{Z}^*),$$

where $g_{\mathbf{w}_0 \odot \boldsymbol{\tau}}$ denotes a randomly initialized neural network with parameters $\mathbf{w}_0$ masked via a binary vector $\boldsymbol{\tau} \in \{0, 1\}^{|\mathbf{w}_0|}$. The network $g$ reconstructs the signal from the latent $\mathbf{Z}$, which is subject to an entropy constraint $R(\mathbf{Z})$, while $J(\boldsymbol{\tau})$ enforces a sparsity-based complexity constraint on the architecture. Here, entropy and complexity regularization act as implicit priors, where mask optimization plays a DIP-like role, and different random initializations induce an ensemble of solutions for improved reconstruction.

We show that this formulation is highly competitive across various zero-shot inverse problems, including denoising, noisy super-resolution, and inpainting (see Fig. 1). Notably, model complexity is tightly controlled by rate and structural priors, resulting in significantly fewer effective parameters than prior approaches. *To the best of our knowledge, this is the first work to explicitly study the priors induced by low-cost implicit codecs for various inverse problems.* These

findings open a promising direction toward lightweight plug-and-play inverse solvers grounded in compression-based principles. The main contributions are:

- We propose *Lottery Prior*, a zero-shot inverse solver that unifies random-network architectural priors with entropy and complexity regularization, together inducing a family of implicit priors with ensemble-like behavior, thereby improving the restoration performance.

- We derive non-asymptotic measurement- and signal-domain error bounds for compression-based maximum-likelihood inverse solvers, showing how rate–distortion constraints act as implicit regularizers with provable guarantees, including denoising as a special case.

- Extensive experiments show that our method achieves state-of-the-art performance on various zero-shot inverse problems such as denoising, noisy super-resolution, and inpainting, while using significantly fewer effective parameters.

## 2. Related work

**Neural data compression.** Neural data compression methods can be broadly categorized into two paradigms: AE-based and implicit codecs. AE-based codecs replace hand-crafted transforms with learned auto-encoders (Ballé et al., 2018; Jiang et al., 2023), achieving strong RD performance but requiring extensive training and offering limited zero-shot flexibility. In contrast, implicit codecs optimize a para-

metric function per signal and encode its parameters, enabling efficient decoding. Representative methods, such as COIN++ (Dupont et al., 2022), COMBINER (He et al., 2024), COOL-CHIC family (Leguay et al., 2023; Kim et al., 2024; Ladune et al., 2026), LotteryCodec (Wu et al., 2025), and MoRIC (Li et al., 2026) achieve strong RD performance with efficient decoding. Together, these works highlight the rapid progress of neural compression toward practical codecs that jointly optimize RD performance and efficiency.

**Zero-shot inverse problem.** Zero-shot inverse problems aim to recover a clean signal from a single degraded observation without any external training data or pretrained source knowledge. Untrained priors, such as DIP (Ulyanov et al., 2020) and Deep Decoder (DD) (Heckel & Hand, 2019), exploit architectural bias and have been applied to various inverse problems. However, most zero-shot work focuses on denoising, including single-image adaptations of self-supervised methods (Batson & Royer, 2019; Quan et al., 2020) and compression-based denoising, which leverages the higher compressibility of structured signals over noise (Donoho, 2002; Weissman & Ordentlich, 2005). While recent zero-shot neural codecs (Zafari et al., 2026) train directly on the observed signal, they remain AE-based and patch-driven, limiting the use of architectural priors, global context, and extension beyond denoising. Dataset-free formulations have also been explored for inverse problems such as compressive sensing and blind super-resolution (Pang et al., 2020; Wang et al., 2022; Shocher et al., 2018; Yue et al., 2022), but remain comparatively underexplored. Meanwhile, "zero-shot" is also used for pretrained foundation- or diffusion-model approaches (Terris et al., 2025a; Spagnoletti et al., 2025; Chung et al., 2023) that handle unseen degradation tasks without specific fine-tuning, or adapt through self-supervised optimization from a small number of measurements. In contrast, our work considers a dataset-free, instance-level setting.

# 3. Theoretical Foundations of Compression-Based Inverse Solvers

## 3.1. Lossy compression

Let $\mathcal{Q} \subset \mathbb{R}^n$ be a signal class, and let $(f, g)$ be a lossy compression code of rate $R$, i.e., $f : \mathcal{Q} \to \{1, \ldots, 2^R\}$ and $g : \{1, \ldots, 2^R\} \to \mathbb{R}^n$. We define the codebook $\mathcal{C} = \{g(i) : i = 1, \ldots, 2^R\}$ and the signal-domain distortion as:

$$\delta = \sup_{\mathbf{x} \in \mathcal{Q}} \frac{1}{n} \big\| \mathbf{x} - g(f(\mathbf{x})) \big\|_2^2. \tag{4}$$

Distortion $\delta$ characterizes a reconstruction error guarantee over $\mathcal{Q}$ for the fixed codebook $\mathcal{C}$. This lossy compression model can serve as an implicit structure-aware prior, exploiting the fact that clean signals are more compressible.

## 3.2. Compression-based ML estimator

Consider a linear inverse problem that aims to reconstruct the clean signal $\mathbf{x} \in \mathbb{R}^n$ from an observation $\mathbf{y} \in \mathbb{R}^m$:

$$\mathbf{y} = \mathbf{A}\mathbf{x} + \mathbf{z}, \mathbf{z} \sim \mathcal{N}(0, \sigma^2 I_m), \tag{5}$$

where $\mathbf{A} \in \mathbb{R}^{m \times n}$ is a known linear degradation operator, and $\mathbf{z} \in \mathbb{R}^m$ represents additive Gaussian noise.

To analyze this, we consider a compression-based maximum-likelihood (ML) approach. Given an observation $\mathbf{y}$ and a lossy compression code $(f, g)$ over $\mathcal{C}$ for the signal class $\mathcal{Q}$, the compression-based ML estimator is given as:

$$\hat{\mathbf{x}} = \arg\max_{\mathbf{c} \in \mathcal{C}} p(\mathbf{y} \mid \mathbf{c}) = \arg\min_{\mathbf{c} \in \mathcal{C}} \| \mathbf{y} - \mathbf{A}\mathbf{c} \|_2^2. \tag{6}$$

Derivations, along with other noise models, are provided in Appendix A.1. The proposed ML estimator leverages the rate constraint to restrict reconstructions to the codebook, thereby implicitly regularizing the solution toward structured signals. Under Gaussian noise, it reduces to a least-squares projection onto $\mathcal{C}$ in measurement domain. We characterize the induced measurement-domain distortion as:

$$\delta_A = \sup_{\mathbf{x} \in \mathcal{Q}} \frac{1}{n} \big\| \mathbf{A}\big(g(f(\mathbf{x})) - \mathbf{x}\big) \big\|_2^2. \tag{7}$$

## 3.3. Theoretical foundations

We derive non-asymptotic error bounds for the proposed inverse solver, with denoising as a special case and signal-domain guarantees under mild assumptions.

**Theorem 3.1** (Error bound for compression-based solver in general inverse problem)**.** *Assume that $\mathbf{x} \in \mathcal{Q}$ and that $(f, g)$ is a lossy compression code for $\mathcal{Q}$ with rate $R$ and distortion $\delta$. Consider the inverse problem $\mathbf{y} = \mathbf{A}\mathbf{x} + \mathbf{z}$, where $\mathbf{z} \sim \mathcal{N}(0, \sigma^2 I_m)$. Let $\hat{\mathbf{x}}$ denote the compression-based ML estimator, and let $\delta_A$ denote the measurement-domain distortion. Then, for any $\eta > 0$, the following holds with probability at least $1 - 2^{-\eta R}$:*

$$\frac{1}{\sqrt{n}} \big\| \mathbf{A}(\hat{\mathbf{x}} - \mathbf{x}) \big\|_2 \leq \sqrt{\delta_A} + 2\sigma\sqrt{\frac{2\ln 2(2+\eta)R}{n}}. \tag{8}$$

*In particular, if the operator norm satisfies $\|\mathbf{A}\|_{\mathrm{op}} \leq L$:*

$$\frac{1}{\sqrt{n}} \big\| \mathbf{A}(\hat{\mathbf{x}} - \mathbf{x}) \big\|_2 \leq L\sqrt{\delta} + 2\sigma\sqrt{\frac{2\ln 2(2+\eta)R}{n}}. \tag{9}$$

*Proof of Theorem 3.1 is provided in Appendix A.2.* □

*Remark* 3.2. The assumption $|\mathbf{A}|_{\mathrm{op}} \leq L$ is mild and is only used to relate $\delta_A$ to $\delta$, as it merely fixes the operator scale and can be enforced by appropriate normalization.

Theorem 3.1 provides a *measurement-domain fidelity guarantee* of the compression-based inverse solver. The bound consists of two components: (i) a distortion term that quantifies the approximation error induced by the compression codebook, and (ii) a rate-dependent noise term that scales with the noise level $\sigma$ and the code rate $R$, capturing the statistical uncertainty induced by noise when the estimator is restricted to a finite-rate codebook. While stated for general linear inverse problems, the result specializes naturally to denoising by setting $\mathbf{A} = \mathbf{I}$, yielding a signal-domain error guarantee, as presented in Corollary 3.3.

**Corollary 3.3** (Denoising specialization)**.** *Consider the denoising setting* $\mathbf{y} = \mathbf{x} + \mathbf{z}$ *with* $\mathbf{z} \sim \mathcal{N}(0, \sigma^2 \mathbf{I}_n)$*. Then* $\delta_A = \delta$*, and Theorem 3.1 directly yields, for any* $\eta > 0$*, with probability at least* $1 - 2^{-\eta R}$*,*

$$\frac{1}{\sqrt{n}} \|\hat{\mathbf{x}} - \mathbf{x}\|_2 \ \leq \ \sqrt{\delta} \ + \ 2\sigma \sqrt{\frac{2 \ln 2 (2 + \eta) R}{n}}. \quad (10)$$

*Proof and analysis is provided in Appendix A.3.* □

*Remark* 3.4. In the denoising setting, our bound reduces to the same *two-term* structure, as existing analysis such as (Zafari et al., 2026), while arising as a direct specialization from a more general inverse-problem framework. A detailed comparison in Appendix A.3 shows that the relative tightness of two bounds depends on the operating regime, characterized by the confidence level $\eta$ and compression rate $R$. In particular, our bound is more favorable in high-confidence or low-rate regimes, typical in zero-shot denoising to mitigate noise overfitting, while existing bounds (Zafari et al., 2026) can be tighter at large rates and low confidence regime.

To obtain an explicit signal-domain error bound, we impose an identifiability condition on the degradation operator over the codebook, leading to the following corollary.

**Corollary 3.5** (Signal-domain error bound for compression-based solver in general inverse problem)**.** *Assume the conditions of Theorem 3.1. Suppose that there exists* $\alpha > 0$ *such that* $\|\mathbf{A}\mathbf{d}\|_2 \geq \alpha \|\mathbf{d}\|_2, \forall \mathbf{d} \in \mathcal{D}$*, where* $\mathcal{D} \triangleq \{\mathbf{c_1} - \mathbf{c_2} : \mathbf{c_1}, \mathbf{c_2} \in \mathcal{C}\}$*. Then, for any* $\eta > 0$*, the following holds with probability at least* $1 - 2^{-\eta R}$*:*

$$\frac{1}{\sqrt{n}} \|\hat{\mathbf{x}} - \mathbf{x}\|_2 \ \leq \ \sqrt{\delta} \ + \ \frac{2}{\alpha} \sqrt{\delta_A} \ + \ \frac{2\sigma}{\alpha} \sqrt{\frac{2 \ln 2 (2 + \eta) R}{n}}. \quad (11)$$

*In particular, if* $\|\mathbf{A}\|_{\mathrm{op}} \leq L$*, then* $\delta_A \leq L^2 \delta$ *and hence*

$$\frac{1}{\sqrt{n}} \|\hat{\mathbf{x}} - \mathbf{x}\|_2 \ \leq \ \left(1 + \frac{2L}{\alpha}\right) \sqrt{\delta} \ + \ \frac{2\sigma}{\alpha} \sqrt{\frac{2 \ln 2 (2 + \eta) R}{n}}. \quad (12)$$

*Proof of Corollary 3.5 are provided in Appendix A.4.* □

*Remark* 3.6. The condition $\|\mathbf{A}\mathbf{d}\|_2 \geq \alpha \|\mathbf{d}\|_2$ is an *injectivity* assumption of the forward operator restricted to the codebook difference set, which is standard in model-based inverse-problem analysis. It holds trivially for denoising ($\mathbf{A} = \mathbf{I}$). While $\mathbf{A}$ may be non-injective for tasks such as inpainting or super-resolution, learned compression codebooks often induce structured sets $\mathcal{D}$ on which this condition holds empirically (e.g., random-mask inpainting). The bound is thus most informative when $\mathbf{A}$ is well behaved on $\mathcal{D}$; otherwise, the proposed scheme can still be used as zero-shot denoisers within plug-and-play frameworks (Terris et al., 2024; Zhang et al., 2021).

Notably, the bound is non-asymptotic and does not require an optimal compression code, indicating that effective restoration is achievable with a well-calibrated RD trade-off. While the above analysis assumes a fixed compression codebook, in practice, such codebooks need not be pre-designed or learned from external data. Instead, they can be learned directly from the degraded observation via entropy regularization, which empirically biases the representation toward natural signal structure. This insight motivates our proposed *Lottery Prior*, which constructs a *family* of implicit codebooks from a single observation in a zero-shot manner.

# 4. Lottery Prior

We now introduce *Lottery Prior* (Fig. 4), a zero-shot inverse solver that enforces compression-based regularization via entropy control and randomized architectural priors, while maintaining measurement consistency. We next detail (i) entropy regularization, (ii) randomized subnetwork selection, and (iii) zero-shot optimization with prior ensembling.

### 4.1. Entropy regularization

**Multi-scale latent representation.** To encourage multi-scale self-similarity, we introduce a hierarchical learnable latent $\mathbf{Z}$ using an $L$-level pyramid structure:

$$\mathbf{Z} \triangleq \{\mathbf{z}_1, \mathbf{z}_2, \ldots, \mathbf{z}_L\}, \text{with} \quad \mathbf{z}_k \in \mathbb{R}^{\frac{H}{2^{k-1}} \times \frac{W}{2^{k-1}}}, \quad (13)$$

where $H$ and $W$ are the image height and width. The resulting latent provides a compact representation of source structures across scales, a key property of natural images.

**Auto-regressive probability model.** To impose entropy constraints, we adopt a factorized auto-regressive model (ARM) $r_\psi(\cdot)$ to estimate the distribution of the quantized latent $\hat{\mathbf{Z}}$. Each element $\hat{z}_{i,j}$ (the $j$-th element of $\mathbf{z}_i$) is conditioned on its $C$ spatial neighbors $\mathbf{c_{i,j}} \in \mathbb{Z}^C$ as $p_\psi(\hat{\mathbf{z}}) = \prod_{i,j} p_\psi(\hat{z}_{i,j}|\mathbf{c_{i,j}})$, where $p_\psi(\hat{\mathbf{z}})$ is modeled as:

$$p_\psi(\hat{z}_{i,j}|\mathbf{c_{i,j}}) = \int_{\hat{z}_{i,j}-0.5}^{\hat{z}_{i,j}+0.5} g(z)dz, \quad (14)$$

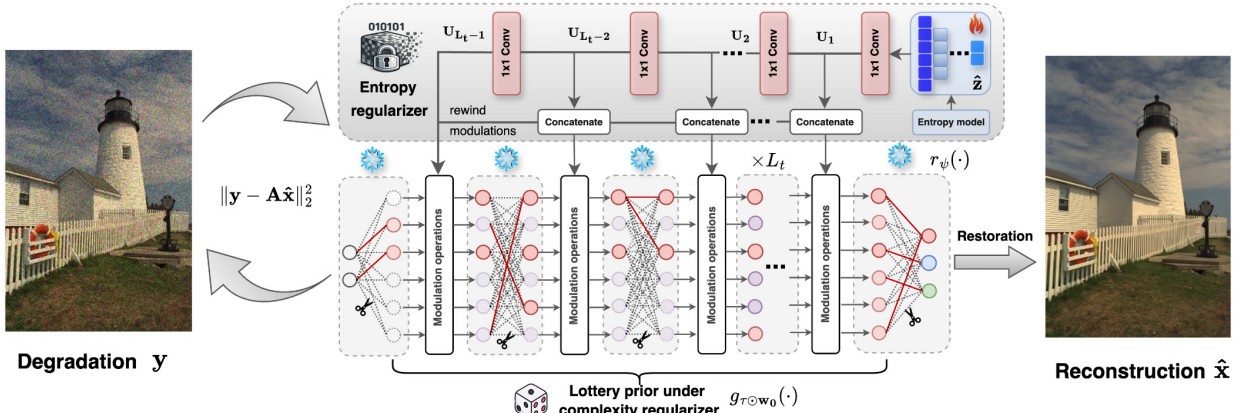

*Figure 4.* Architecture of Lottery Prior. A randomly initialized network maps coordinates to signal values by selecting subnetworks via a learned binary mask. Two implicit priors are enforced: an entropy prior, implemented through an entropy-regularized latent that generates layer-wise modulations, and an architectural prior, realized by subnetwork selection within a frozen random network. Solid red lines denote active weights, while dashed lines indicate masked connections.

where $g \sim \mathcal{L}(\mu_{i,j}, \sigma_{i,j})$ is a Laplace distribution with parameters $\mu_{i,j}, \sigma_{i,j} = r_\psi(\mathbf{c_{i,j}})$. The resulting rate term is:

$$R(\hat{\mathbf{z}}) = -\log_2 p_\psi(\hat{\mathbf{Z}}) = -\sum_{i,j} \log_2 p_\psi(\hat{z}_{i,j} | \mathbf{c_{i,j}}). \quad (15)$$

Beyond rate regularization, the ARM here also injects spatial context, which is particularly beneficial for inpainting, as later demonstrated in our experiments.

### 4.2. Randomized subnetwork selection

For an over-parameterized network $g_{\mathbf{W_o}}$, which maps pixel coordinates to signal values, we seek a high-performing subnetwork $g_{\tau \odot \mathbf{W_o}}$ via a learned mask $\tau$, which serves as a prior by learning signal statistics into the network structure while remaining consistent with the degradation.

**Subnetwork selection.** The randomized network $g_{\tau \odot \mathbf{W_o}}$ consists of $L_t$ masked linear layers with latent modulations, where Fourier initialization (Shi et al., 2024) is adopted to mitigate spectral bias while preserving stable variance. Subnetworks are selected via a learnable score matrix $\mathbf{P}$ with the same shape as $\mathbf{W_0}$, where each element scores the corresponding weight. The binary mask $\tau$ is obtained by activating the top $r_a\%$ of weights according to $\mathbf{P}$ while zeroing out the rest. Formally, the output of neuron $k$ in layer $i$, denoted as $v_k^{(i)}$, is given by: $v_k^{(i)} = \sigma\left(\sum_{j=1}^{d_{i-1}} \tau_{k,j}^{(i-1)} w_{kj}^{(i-1)} m(v_j^{(i-1)})\right)$, where $\tau_{k,j}$ is obtained by thresholding $\mathbf{P}$, $m(\cdot)$ denotes modulation function, $\sigma$ the activation function, and $w_{kj}^{(i-1)}$ the weight connecting the $k$-th neuron in $i$-th layer to the $j$-th neuron of the $(i-1)$-th layer. The scores in $\mathbf{P}$ are optimized using a straight-through estimator (Wortsman et al., 2019; Wu et al., 2025) under measurement-consistency constraints between $\mathbf{A}\hat{\mathbf{x}}$ and $\mathbf{y}$, together with entropy-based rate regularization.

**Latent modulation.** To simplify the learning process and improve restoration quality, we introduce a lightweight modulation network $f_\theta(\cdot)$ that maps entropy-constrained latents to layer-wise modulation signals. As shown in Fig. 4, the quantized $\hat{\mathbf{Z}}$ is upsampled via transposed convolutions, yielding $\mathbf{U_0} = [\mathbf{u_1}; \mathbf{u_2}; \ldots; \mathbf{u_L}] \in \mathbb{Z}^{L \times HW}$, which are processed by convolutional operations in $f_\theta$ to produce layer-wise modulations $\mathbf{U_i}$. Following the rewind strategy introduced in (Wu et al., 2025), modulation for $i$-th layer of $g_{\tau \odot \mathbf{W_0}}$ is constructed by reverse concatenation: $\mathbf{M_i} = \mathrm{Concat}(\mathbf{U_{L_t-1}}, \mathbf{U_{L_t-2}}, \ldots, \mathbf{U_{L_t-i}})$, and concatenated with the layer output $\mathbf{F}_i$ to produce $\mathbf{G_i} = m(\mathbf{F_i}) \triangleq \mathrm{Concat}(\mathbf{M_i}, \mathbf{F_i})$. Intuitively, this multi-scale modulation reinforces a DIP-style architectural prior by propagating structured information across layers, enabling effective subnetwork selection and improved restoration.

### 4.3. Zero-shot learning and prior ensemble

**Optimization strategy.** *Lottery Prior* is optimized in a zero-shot manner by learning the parameter set $\mathbf{\Omega} \triangleq \{\mathbf{Z}, \psi, \theta, \tau\}$ from a single degraded observation $\mathbf{y}$. Compared with Eq. (3), additional parameters $\psi$ and $\theta$ are introduced to facilitate the restoration process. The optimization minimizes the following objective:

$$\mathcal{L}(\mathbf{\Omega}) = D(\mathbf{y}, \mathbf{A}\hat{\mathbf{x}}) + \lambda R(\hat{\mathbf{Z}}), \text{ with } \hat{\mathbf{x}} = g_{\tau \odot \mathbf{w_0}}(f_\theta(\hat{\mathbf{Z}})), \quad (16)$$

where $D(\cdot)$ is the mean-squared error enforcing measurement consistency, and $\lambda$ is chosen empirically according to the degradation level. Training details are provided in Appendix B.4.

**Restoration stage.** To exploit prior diversity, we vary random seeds and mask ratios to obtain a family of reconstructions $\{\hat{\mathbf{x}_1}, \cdots, \hat{\mathbf{x}_k}\}$, which are aggregated as: $\mathbf{x}^* =$

*Table 1.* Denoising performance comparison under AWGN and Poisson Noise, average PSNR(dB) and SSIM are reported. Best results are in **bold**, second-best are underlined. The number of effective network parameters reflects the size of the architectural prior and indicates restoration efficiency at inference.

| Noise level $\sigma$ or $\alpha$ | Method | Network parameters | AWGN, $\mathcal{N}(0, \sigma^2)$ | | | Poisson, Poisson$(\alpha x)/\alpha$ | | |
|---|---|---|---|---|---|---|---|---|
| | | | Set11 | Set13 | Kodak24 | Set11 | Set13 | Kodak24 |
| 15 | JPEG2K | - | 27.45 / 0.7699 | 26.69 / 0.7543 | 27.86 / 0.7457 | 22.35 / 0.5882 | 21.76 / 0.5494 | 22.56 / 0.5249 |
| | BM3D | - | 32.22 / **0.8991** | 31.15 / 0.8808 | 32.37 / 0.8754 | **26.66** / **0.7505** | 25.64 / 0.6912 | 27.04 / 0.6900 |
| | DIP | 2.2M | 29.11 / 0.7990 | 30.31 / 0.8570 | 31.42 / 0.8454 | 23.69 / 0.5863 | 25.14 / 0.6916 | 26.37 / 0.6761 |
| | DD | 0.1M | 28.83 / 0.8215 | 29.22 / 0.8371 | 28.71 / 0.8016 | 24.37 / 0.6629 | 24.96 / 0.7006 | 25.59 / 0.6679 |
| | S2S | 1.0M | 26.81 / 0.8158 | 20.61 / 0.6879 | 23.08 / 0.7695 | 21.75 / 0.6872 | 19.23 / 0.6553 | 22.52 / 0.7418 |
| | ZS-N2S | 0.2M | 28.92 / 0.8495 | 18.18 / 0.5690 | 18.68 / 0.5540 | 25.06 / 0.7051 | 21.23 / 0.6066 | 22.24 / 0.6170 |
| | ZS-N2N | 22.3K | 30.01 / 0.8169 | 30.95 / 0.8701 | 32.30 / 0.8650 | 24.04 / 0.5766 | 25.37 / 0.6878 | 26.80 / 0.6757 |
| | ZS-NCD | 0.4M | 31.35 / 0.8580 | 31.93 / **0.8983** | 33.18 / 0.9026 | 25.65 / 0.7132 | **26.44** / 0.7434 | 27.64 / 0.7432 |
| | Lottery Prior | 0.7-3.3K | **32.30** / 0.8922 | **32.74** / 0.8963 | **34.29 / 0.9118** | 26.16 / 0.7327 | 26.35 / **0.7702** | **28.23 / 0.7778** |
| 25 | JPEG2K | - | 24.91 / 0.6997 | 24.32 / 0.6676 | 25.43 / 0.6550 | 23.03 / 0.6108 | 22.65 / 0.5952 | 23.58 / 0.5680 |
| | BM3D | - | **29.79 / 0.8523** | 28.81 / 0.8213 | 29.98 / 0.8092 | 22.70 / 0.5741 | 22.17 / 0.5992 | 24.13 / 0.5931 |
| | DIP | 2.2M | 26.60 / 0.7128 | 27.85 / 0.7837 | 28.90 / 0.7738 | 24.94 / 0.6512 | 26.13 / 0.7289 | 27.49 / 0.7243 |
| | DD | 0.1M | 26.93 / 0.7530 | 27.40 / 0.7832 | 27.62 / 0.7496 | 25.48 / 0.7022 | 26.04 / 0.7373 | 26.56 / 0.7060 |
| | S2S | 1.0M | 23.32 / 0.7306 | 17.95 / 0.5998 | 20.69 / 0.6949 | 23.40 / 0.7355 | 20.18 / 0.6927 | 23.09 / 0.7674 |
| | ZS-N2S | 0.2M | 27.30 / 0.7971 | 20.39 / 0.6200 | 20.89 / 0.6156 | 26.01 / 0.7478 | 21.19 / 0.6312 | 21.47 / 0.6277 |
| | ZS-N2N | 22.3K | 27.18 / 0.7173 | 28.36 / 0.8001 | 29.54 / 0.7798 | 25.40 / 0.6432 | 26.75 / 0.7455 | 28.21 / 0.7374 |
| | ZS-NCD | 0.4M | 28.93 / 0.8079 | 29.33 / 0.8351 | 30.60 / 0.8144 | 27.10 / 0.7431 | 27.60 / 0.7827 | 28.77 / 0.7677 |
| | Lottery Prior | 0.7-3.3K | 29.73 / 0.8420 | **30.00** / 0.8414 | **31.48 / 0.8580** | **27.64 / 0.7734** | **27.89 / 0.8088** | **29.55 / 0.8115** |

$\mathcal{G}\big(\{\hat{\mathbf{x}}_i\}_{i=1}^k\big)$. We use simple linear averaging for denoising and noisy super-resolution, which is sufficient for state-of-the-art performance, and report single-prior results for other inverse problems, such as image inpainting. Significant performance improvements are expected with more advanced aggregation strategies (Terris et al., 2025b).

# 5. Experimental results

We evaluated *Lottery Prior* on three canonical inverse-problems: (i) Denoising, including AWGN $\mathcal{N}(0, \sigma^2)$ with $\sigma \in \{15, 25\}$ and Poisson noise Poisson$(\alpha x)$ with $\alpha \in \{15, 25\}$; (ii) Noisy super-resolution, using $\times 4$ and $\times 8$ bicubic downsampling followed by additive Gaussian noise ($\sigma = 15$); and (iii) Inpainting, covering text removal, region completion and random pixel masking. Experimental setup and implementations are detailed in Appendix B.4.

## 5.1. Denoising under AWGN and Poisson Noise

We evaluate denoising performance under AWGN and Poisson noise on Set11 (Zhang et al., 2017), Set13 (Zeyde et al., 2010) ($192 \times 192$ central crops), and Kodak24. Comparisons against representative zero-shot methods (see Appendix B for baseline details), including JPEG2K, BM3D, DIP, DD, ZS-N2S, S2S, ZS-N2N, and ZS-NCD, are reported in Table 1. Overall, *Lottery Prior* consistently achieves the best performance across all noise levels and datasets. Pixel-masking approaches such as ZS-N2S and S2S exhibit significant degradation on high-resolution images, likely due to blind-spot masking that limits detail recovery. Notably, *Lottery Prior* shows an increasing advantage over the previ-

*Table 2.* Super-resolution PSNR on Set5 with AWGN ($\sigma = 15$).

| | Baby | Bird | Butterfly | Head | Woman | **Avg.** |
|---|---|---|---|---|---|---|
| **$4\times$ super-resolution** | | | | | | |
| No prior | 21.56 | 20.59 | 17.25 | 21.47 | 19.64 | 20.10 |
| Bicubic | 25.04 | 24.41 | 19.89 | 24.63 | 22.64 | 23.32 |
| BM3D+Bicubic | 28.17 | 25.56 | 20.33 | 27.00 | 23.88 | 24.99 |
| DIP | 24.07 | 21.37 | 19.17 | 21.53 | 20.70 | 21.37 |
| Ours | **28.66** | **26.06** | **22.48** | **27.26** | **24.87** | **25.87** |
| LapSRN | 26.27 | 25.07 | 20.61 | 25.71 | 23.41 | 24.21 |
| BM3D+LapSRN | 28.21 | 25.51 | 20.92 | 27.03 | 24.19 | 25.17 |
| **$8\times$ super-resolution** | | | | | | |
| | Baby | Bird | Butterfly | Head | Woman | **Avg.** |
| No prior | 20.46 | 18.85 | 15.50 | 20.84 | 17.64 | 18.66 |
| Bicubic | 23.26 | 21.55 | 16.29 | 23.59 | 19.88 | 20.91 |
| BM3D+Bicubic | 24.81 | **21.68** | 16.32 | 24.99 | 20.30 | 21.62 |
| DIP | 21.01 | 20.15 | 16.55 | 21.00 | 19.01 | 19.55 |
| Ours | **25.46** | 21.66 | **17.28** | **25.06** | **20.90** | **22.07** |
| LapSRN | 24.25 | 21.51 | 16.18 | 24.32 | 20.26 | 21.30 |
| BM3D+LapSRN | 25.20 | 21.42 | 16.21 | 24.98 | 20.49 | 21.66 |

ous state-of-the-art ZS-NCD as image resolution increases, highlighting the benefit of our global structural prior over patch-wise neural compression. Moreover, our method is lightweight, avoids sensitive early stopping, and yields more stable restorations (see Appendix C.3). These results indicate strong potential for integrating lightweight implicit codecs into future plug-and-play zero-shot inverse solvers.

## 5.2. Super-Resolution with Noisy Observations

We report the performance of noisy super-resolution on Set5 in Table 2. More metrics, such as LPIPS and MS-SSIM,

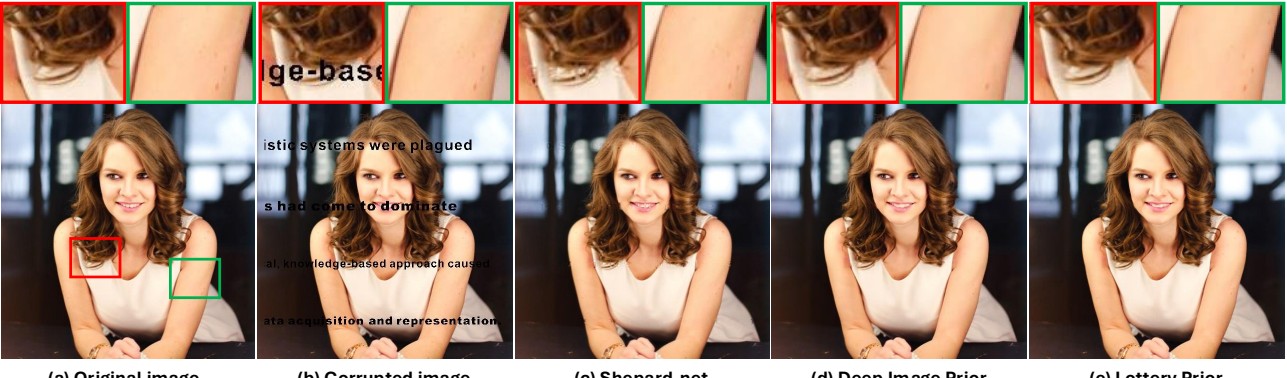

| (a) Original image | (b) Corrupted image | (c) Shepard-net | (d) Deep Image Prior | (e) Lottery Prior |

*Figure 5.* Compared with Shepard-net (Ren et al., 2015) and DIP, *Lottery Prior* preserves finer details without halo artifacts or early stopping, while DIP loses fidelity due to early stopping.

are reported in Table 11 of our appendix. DIP performs poorly in this challenging task due to its sensitivity to noise and reliance on early stopping, often requiring task-specific training strategies or architectural modifications, which limit its applicability to composite degradations. We additionally included a supervised super-resolution baseline LapSRN, combined with BM3D denoising. Despite operating in a zero-shot setting, *Lottery Prior* consistently achieves the highest PSNR across all test images. This demonstrates the robustness of compression-induced rate regularization. Qualitative comparisons in Figs. 8 and 9 further illustrate the superiority of *Lottery Prior*. In our experiments, the proposed method already demonstrates leading performance in all three metrics (PSNR, MS-SSIM, and LPIPS), and, in some cases, surpasses even strong supervised priors

### 5.3. Image Inpainting

Next, we evaluate the inpainting performance on three tasks: *(i) text removal*, *(ii) random mask inpainting*, and *(iii) region inpainting*. Text removal results in Fig. 5 show that *Lottery Prior* preserves finer details with fewer artifacts than both supervised methods and DIP, while requiring no early stopping and operating in a zero-shot manner. Random mask inpainting results are reported in Table 3 and Fig. 6, where our method demonstrates strong performance with accurate detail completion, benefiting from auto-regressive entropy modeling over nearby contextual cues. Additional visual results are provided in Fig.10.

Region inpainting results in Figs. 11 and 12 further show that large missing areas can be plausibly completed using surrounding context, indicating that randomized masked networks act as an implicit structural prior similar to DIP. Fig. 12 illustrates the trade-off between rate constraints and inpainting fidelity. While generation capability is naturally limited by available context and model capacity, all results here use only a single prior; leveraging multiple lottery

*Table 3.* Random-mask inpainting PSNR on Set5 (mask ratio $\rho$).

|  | Baby | Bird | Butterfly | Head | Woman | **Avg.** |
|---|---|---|---|---|---|---|
| Deep image prior ($\rho = 0.8$) | 29.56 | 29.93 | 23.87 | 26.33 | 25.99 | 27.14 |
| Lottery prior ($\rho = 0.8$) | **31.25** | **30.66** | **25.11** | **27.99** | **27.53** | **28.51** |
| Deep image prior ($\rho = 0.5$) | 31.87 | 35.38 | 28.31 | 29.66 | 31.13 | 31.27 |
| Lottery prior ($\rho = 0.5$) | **35.46** | **36.28** | **30.61** | **30.65** | **33.09** | **33.22** |

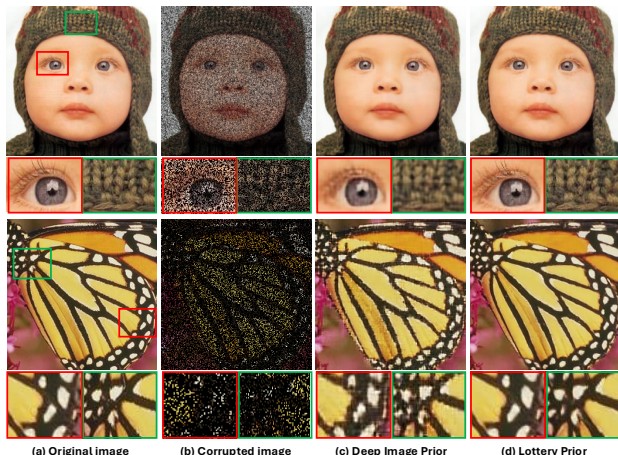

| (a) Original image | (b) Corrupted image | (c) Deep Image Prior | (d) Lottery Prior |

*Figure 6.* Visualization of inpainting results with mask ratios 0.5 (top) and 0.8 (bottom). *Lottery Prior* combines auto-regressive entropy modeling for spatial context with a global implicit prior from the randomized codec, enabling effective detail completion.

priors with stronger architectures and aggregation strategies is expected to yield further gains.

### 5.4. Task and modality adaptation

We further evaluated our method on JPEG artifact removal using AFHQ dataset (the first image from each class), with results reported in Table 5. The results show that our framework is not restricted to linear operators and consistently outperforms DIP. Although zero-shot JPEG artifact removal is challenging due to its nonlinear quantization degradation,

*Table 4.* Task adaptation experiments of RAM and Lottery Prior on demosaicing tasks, where the models are evaluated on classic samples (randomly selected from Set5/13 for testing; remaining samples used for RAM finetuning). RAM is pretrained on the LSDIR dataset (84,991 high-quality images) across 6 inverse tasks (such as deblur or inpainting) and then fine-tuned in a self-supervised manner (10 measurements), while Lottery Prior is trained using a single prior from a single degradation without external data.

| Classic samples (randomly selected from Set5/13 for testing; remaining samples used for finetuning) | | | | | | |
|---|---|---|---|---|---|---|
| PSNR | Baby | Butterfly | Woman | Coastguard | Foreman | **Avg.** |
| **Lottery prior** | **33.17** | **31.15** | **32.77** | **31.59** | **32.43** | **32.22** |
| RAM (Pretrained) | 32.58 | 30.77 | 31.86 | 30.63 | 31.08 | 31.38 |
| RAM (Finetuned N=10) | 37.62 | 33.68 | 37.71 | 38.77 | 36.58 | 36.87 |

*Table 5.* Noisy JPEG artifact removal ($Q = 5, \sigma = 25$).

| PSNR / MS-SSIM | Cat | Dog | Wild | **Avg.** |
|---|---|---|---|---|
| Deep image prior | 25.17/0.81 | 24.09/0.77 | 23.13/0.77 | 24.13/0.79 |
| **Lottery prior** | **26.62/0.88** | **25.27/0.82** | **24.73/0.81** | **25.54/0.83** |

*Table 6.* Runtime (sec/1k steps) and complexity (MACs/pixel) for zero-shot methods to denoise a Set11 image with $\sigma = 15$ and mask ratio 0.5, on NVIDIA 3090Ti GPU and AMD 5900X CPU.

| Methods | Run time | Required steps | Per-step complexity |
|---|---|---|---|
| **ZS-NCD** | 96.91 | 20k | 2350.08k |
| **DIP** | 22.38 | 2k-4k | 899.78k |
| **Lottery prior** | 12.38 | 5k-10k | 10.37k |

the proposed compression-based prior remains effective.

We also compare with RAM (Terris et al., 2025a) on unseen tasks and modalities, including demosaicing (Table 4) and microscopy imaging (Table 12). RAM achieves strong results, especially after self-supervised fine-tuning, while our method performs consistently without pretraining. Since RAM relies on extensive supervised pretraining, we report these results only to calibrate our dataset-free method against pretrained foundation-model priors. Additional MRI experiments are provided in Table 13 of the Appendix, further demonstrating task and modality adaptation.

Note that highly ill-conditioned degradations, such as JPEG artifact removal and MRI reconstruction, remain challenging in the pure zero-shot setting. While our method shows initial effectiveness, fully recovering signals using only compressibility and untrained priors remains difficult. Promising directions include approximate forward models, perceptual coding objectives, and iterative or hybrid schemes to better balance data consistency and prior regularization.

### 5.5. Ablation study and complexity analysis

Detailed ablation studies for each component are reported in Tables 8, 9, and 10, validating the proposed design and showing that a moderate model size is preferred in practice. We further note that, under the randomized network setting, different architectures can induce diverse random codec priors; judiciously combining such priors may further improve performance, which we leave for future work.

We report runtime and computational complexity in Table 6 using the first three Set11 images, with backward MACs approximated as twice the forward MACs (Baydin et al., 2018). Compared to ZS-NCD (0.4M parameters, officially reported 40 min/image) and lightweight DIP variants (1-2 min/image; 2.2M parameters for the original model), our

method is more efficient due to its compact design (0.7-3.3k parameters) and avoidance of patch-wise redundancy, yielding lower per-iteration cost and stronger performance. Step-wise results in Table 14 further show a flexible time-performance trade-off. Our framework also supports flexible prior aggregation: a single prior (5-10k steps, ∼2 min) suffices for structured tasks such as inpainting, while aggregating a few priors (e.g., 5 mask ratios, ∼5-10 min) further improves denoising.

## 6. Conclusion and future work

We presented *Lottery Prior*, a lightweight zero-shot framework for inverse problems that combines entropy-based compression regularization with randomized architectural priors. By constraining the solution space via rate control and leveraging the structural bias of random networks, it induces a family of implicit priors from a single observation without external training. We provide non-asymptotic error bounds for compression-based solvers, showing how RD constraint acts as an implicit regularizer with provable guarantees. Extensive experiments demonstrate state-of-the-art results on denoising, noisy super-resolution, and inpainting with significantly fewer effective parameters.

**Limitations and future work.** While *Lottery Prior* excels on composite degradations, its gains are less pronounced in the case of severely ill-conditioned forward operators. From an information-theoretic perspective, these settings are dominated by operator-induced ambiguity rather than signal uncertainty, reducing the effectiveness of entropy-based regularization. A promising direction for future work is to integrate *Lottery Prior* into plug-and-play frameworks that explicitly handle the forward operator.

## Impact Statement

This paper presents work whose goal is to advance the field of Machine Learning. There are many potential societal consequences of our work, none of which we feel must be specifically highlighted here.

## Acknowledgments

This work was started while Haotian Wu was with Imperial College London. This work received funding from the UKRI for the projects INFORMED-AI (EP/Y028732/1) and AI-R (ERC Consolidator Grant, EP/X030806/1).

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

# Appendix

## Appendix Contents

## A. Proof of theorem and corollary

### A.1. Derivations of ML estimator

**Recall of [The ML estimator and noise model]:** In this paper, we consider the standard Gaussian linear inverse problem:

$$\mathbf{y} = \mathbf{Ax} + \mathbf{z}, \mathbf{z} \sim \mathcal{N}(0, \sigma^2 I_m). \tag{17}$$

Given a noisy observation $\mathbf{y}$ and a lossy compression code $(f, g)$ with codebook $\mathcal{C}$ for the signal class $\mathcal{Q}$, the compression-based ML estimator is given as:

$$\boxed{\hat{\mathbf{x}} = \arg\max_{\mathbf{c} \in \mathcal{C}} p(\mathbf{y} \mid \mathbf{c}) = \arg\min_{\mathbf{c} \in \mathcal{C}} \|\mathbf{y} - \mathbf{Ac}\|_2^2.} \tag{18}$$

We emphasize that the derived ML estimator is specific to Gaussian noise.

*Proof.* Under the additive Gaussian noise model $\mathbf{z} \sim \mathcal{N}(0, \sigma^2 \mathbf{I}_m)$, the conditional distribution of the observation $\mathbf{y}$ given a candidate signal $\mathbf{c} \in \mathcal{C}$ is:

$$p(\mathbf{y} \mid \mathbf{c}) = \frac{1}{(2\pi\sigma^2)^{m/2}} \exp\left(-\frac{1}{2\sigma^2}\|\mathbf{y} - \mathbf{Ac}\|_2^2\right). \tag{19}$$

Taking the logarithm yields the log-likelihood:

$$\log p(\mathbf{y} \mid \mathbf{c}) = -\frac{1}{2\sigma^2}\|\mathbf{y} - \mathbf{Ac}\|_2^2 - \frac{m}{2}\log(2\pi\sigma^2). \tag{20}$$

Since the second term is independent of $\mathbf{c}$, maximizing the log-likelihood over $\mathbf{c} \in \mathcal{C}$ is equivalent to minimizing the squared Euclidean distance:

$$\arg\max_{\mathbf{c} \in \mathcal{C}} p(\mathbf{y} \mid \mathbf{c}) = \arg\min_{\mathbf{c} \in \mathcal{C}} \|\mathbf{y} - \mathbf{Ac}\|_2^2. \tag{21}$$

$$\square$$

Thus, under Gaussian noise, the compression-based ML estimator reduces to a least-squares projection onto the codebook by evaluating the likelihood of the observation under the conditional model associated with each candidate signal.

*Remark* A.1 (Other noise models). For noise models other than Gaussian, the maximum-likelihood (ML) estimator generally takes a different form. For instance, under Poisson noise with observations $y_i \sim \text{Poisson}(\alpha(\mathbf{Ax})_i)$ (each element of the measured signal), the corresponding ML estimator minimizes the Poisson negative log-likelihood:

$$\hat{\mathbf{x}} = \arg\min_{\mathbf{c} \in \mathcal{C}} \sum_i \left[\alpha(\mathbf{Ac})_i - y_i \log(\mathbf{Ac})_i\right]. \tag{22}$$

Theoretical guarantees for such noise models can be derived using arguments analogous to the Gaussian case, relying on appropriate inequalities. For clarity and interpretability, we therefore focus our theoretical guarantees on linear inverse problems with Gaussian noise in this work.

### A.2. Proof of Theorem 3.1

**Recall of Theorem 3.1 [Compression-based ML for General Inverse Problem]:** Assume that $\mathbf{x} \in \mathcal{Q}$ and that $(f, g)$ is a lossy compression code for $\mathcal{Q}$ with rate $R$ and distortion $\delta$. Consider the inverse problem $\mathbf{y} = \mathbf{A}\mathbf{x} + \mathbf{z}$, where $\mathbf{z} \sim \mathcal{N}(0, \sigma^2 I_m)$. Let $\hat{\mathbf{x}}$ denote the compression-based ML estimator, and $\delta_A$ denote the measurement-domain distortion. Then, for any $\eta > 0$, the following holds with probability at least $1 - 2^{-\eta R}$:

$$\frac{1}{\sqrt{n}} \left\| \mathbf{A}(\hat{\mathbf{x}} - \mathbf{x}) \right\|_2 \; \leq \; \sqrt{\delta_A} \; + \; 2\sigma \sqrt{\frac{2\ln 2(2+\eta)R}{n}}. \tag{23}$$

In particular, if the operator norm of $\mathbf{A}$ satisfies $\|\mathbf{A}\|_{\mathrm{op}} \leq L$, then

$$\frac{1}{\sqrt{n}} \left\| \mathbf{A}(\hat{\mathbf{x}} - \mathbf{x}) \right\|_2 \; \leq \; L\sqrt{\delta} \; + \; 2\sigma \sqrt{\frac{2\ln 2(2+\eta)R}{n}}. \tag{24}$$

*Proof of Theorem 3.1.* Let $\mathbf{x_c} = g(f(\mathbf{x})) \in \mathcal{C}$. By definition of $\hat{\mathbf{x}}$,

$$\|\mathbf{y} - \mathbf{A}\hat{\mathbf{x}}\|_2^2 \leq \|\mathbf{y} - \mathbf{A}\mathbf{x_c}\|_2^2. \tag{25}$$

Substituting $\mathbf{y} = \mathbf{A}\mathbf{x} + \mathbf{z}$ into two sides of Eqn. (25) yields

$$\|\mathbf{A}\mathbf{x} + \mathbf{z} - \mathbf{A}\hat{\mathbf{x}}\|_2^2 \leq \|\mathbf{A}\mathbf{x} + \mathbf{z} - \mathbf{A}\mathbf{x_c}\|_2^2, \tag{26}$$

i.e.,

$$\|\mathbf{z} - \mathbf{A}(\hat{\mathbf{x}} - \mathbf{x})\|_2^2 \leq \|\mathbf{z} + \mathbf{A}(\mathbf{x} - \mathbf{x_c})\|_2^2. \tag{27}$$

Expanding both sides leads to:

$$\|\mathbf{z} - \mathbf{A}(\hat{\mathbf{x}} - \mathbf{x})\|_2^2 = \|\mathbf{z}\|_2^2 - 2\mathbf{z}^\top \mathbf{A}(\hat{\mathbf{x}} - \mathbf{x}) + \|\mathbf{A}(\hat{\mathbf{x}} - \mathbf{x})\|_2^2, \tag{28}$$

and

$$\|\mathbf{z} + \mathbf{A}(\mathbf{x} - \mathbf{x_c})\|_2^2 = \|\mathbf{z}\|_2^2 + 2\mathbf{z}^\top \mathbf{A}(\mathbf{x} - \mathbf{x_c}) + \|\mathbf{A}(\mathbf{x} - \mathbf{x_c})\|_2^2. \tag{29}$$

Removing $\|\mathbf{z}\|_2^2$ and rearranging yields:

$$\|\mathbf{A}(\hat{\mathbf{x}} - \mathbf{x})\|_2^2 \leq \|\mathbf{A}(\mathbf{x} - \mathbf{x_c})\|_2^2 + 2\mathbf{z}^\top \mathbf{A}\big((\hat{\mathbf{x}} - \mathbf{x}) + (\mathbf{x} - \mathbf{x_c})\big) = \|\mathbf{A}(\mathbf{x_c} - \mathbf{x})\|_2^2 + 2\mathbf{z}^\top \mathbf{A}(\hat{\mathbf{x}} - \mathbf{x_c}). \tag{30}$$

We therefore have:

$$\|\mathbf{A}(\hat{\mathbf{x}} - \mathbf{x})\|_2^2 \leq \|\mathbf{A}(\mathbf{x_c} - \mathbf{x})\|_2^2 + 2\mathbf{z}^\top \mathbf{A}(\hat{\mathbf{x}} - \mathbf{x_c}). \tag{31}$$

To allow for a tractable probabilistic analysis, we then aim to upper bound it by a supremum over a finite set. Recall that there exist indices $i, j \in \{1, \ldots, 2^R\}$ such that $\hat{\mathbf{x}} = g(i)$ and $\mathbf{x_c} = g(j)$, and therefore $\hat{\mathbf{x}} - \mathbf{x_c} = g(i) - g(j) \in \mathcal{D}$, where $\mathcal{D} \triangleq \{\mathbf{c_i} - \mathbf{c_j} : \mathbf{c_i}, \mathbf{c_j} \in \mathcal{C}\}$ for any $i, j$. Note that since $\mathcal{C}$ is a finite codebook with $|\mathcal{C}| = 2^R$, the difference set $\mathcal{D}$ is finite with $|\mathcal{D}| \leq 2^{2R}$. Consequently, for the particular random vector $\hat{\mathbf{x}} - \mathbf{x_c}$, we have:

$$\mathbf{z}^\top \mathbf{A}(\hat{\mathbf{x}} - \mathbf{x_c}) \leq \sup_{\mathbf{d} \in \mathcal{D}} \mathbf{z}^\top \mathbf{A}\mathbf{d}. \tag{32}$$

Given an arbitrary $\mathbf{d} \in \mathbb{R}^n$, since $\mathbf{z} \sim \mathcal{N}(0, \sigma^2 \mathbf{I_m})$ and $\mathbf{A}\mathbf{d} \in \mathbb{R}^m$ is deterministic when $\mathbf{d}$ is fixed, the random variable $\mathbf{z}^\top \mathbf{A}\mathbf{d}$ can be given as:

$$\mathbf{z}^\top \mathbf{A}\mathbf{d} = \langle \mathbf{z}, \ \mathbf{A}\mathbf{d} \rangle = (\mathbf{A}\mathbf{d})^\top \mathbf{z}, \tag{33}$$

which is a Gaussian with the corresponding mean and variance as:

$$\mathbb{E}[\mathbf{z}^\top \mathbf{A}\mathbf{d}] = (\mathbf{A}\mathbf{d})^\top \mathbb{E}[\mathbf{z}] = 0, \tag{34}$$

$$\mathrm{Var}(\mathbf{z}^\top \mathbf{Ad}) = \mathbb{E}\big[(\mathbf{z}^\top \mathbf{Ad})^2\big] = \mathbb{E}\big[(\mathbf{Ad})^\top \mathbf{zz}^\top (\mathbf{Ad})\big] = (\mathbf{Ad})^\top \mathbb{E}[\mathbf{zz}^\top](\mathbf{Ad}). \tag{35}$$

Since $\mathbb{E}[\mathbf{zz}^\top] = \sigma^2 \mathbf{I_m}$, it can be simplified as:

$$\mathrm{Var}(\mathbf{z}^\top \mathbf{Ad}) = (\mathbf{Ad})^\top (\sigma^2 \mathbf{I_m})(\mathbf{Ad}) = \sigma^2 \|\mathbf{Ad}\|_2^2. \tag{36}$$

Therefore, for any fixed $\mathbf{d}$, we have

$$X_d \triangleq \mathbf{z}^\top \mathbf{Ad} \sim \mathcal{N}\big(0,\ \sigma^2 \|\mathbf{Ad}\|_2^2\big). \tag{37}$$

Let $U_d \triangleq X_d/(\sigma \|\mathbf{Ad}\|_2) \sim \mathcal{N}(0,1)$. Using the standard Gaussian tail bound $\mathrm{Pr}(U_d \geq t) \leq e^{-t^2/2}$ for any $t > 0$, we obtain:

$$\mathrm{Pr}\left(X_d \geq \sigma \|\mathbf{Ad}\|_2 t\right) = \mathrm{Pr}(U_d \geq t) \leq e^{-t^2/2}. \tag{38}$$

Applying a union bound over the finite set $\mathcal{D}$ yields

$$\mathrm{Pr}(\exists \mathbf{d} \in \mathcal{D}:\ X_d \geq \sigma \|\mathbf{Ad}\|_2 t) \leq \sum_{\mathbf{d} \in \mathcal{D}} e^{-t^2/2} = |\mathcal{D}| e^{-t^2/2}. \tag{39}$$

Equivalently, with probability at least $1 - |\mathcal{D}| e^{-t^2/2}$, we obtain that for all $\mathbf{d} \in \mathcal{D}$ we satisfy $X_d \leq \sigma \|\mathbf{Ad}\|_2 t$.

Since $|\mathcal{C}| = 2^R$, we have $|\mathcal{D}| \leq |\mathcal{C}|^2 = 2^{2R}$. Hence,

$$\mathrm{Pr}\big(\exists \mathbf{d} \in \mathcal{D}:\ \mathbf{z}^\top \mathbf{Ad} \geq \sigma \|\mathbf{Ad}\|_2\, t\big) \leq 2^{2R} e^{-t^2/2}. \tag{40}$$

We next choose a $t$ such that the right-hand side is upper bounded by a form of $2^{-\eta R}$, i.e.,

$$2^{2R} e^{-t^2/2} \leq 2^{-\eta R}. \tag{41}$$

Taking logarithms on both sides yields

$$\frac{t^2}{2} \geq (2+\eta)R \ln 2, \tag{42}$$

and therefore we set

$$t \triangleq \sqrt{2(2+\eta)R \ln 2}. \tag{43}$$

With this choice, it follows that with probability at least $1 - 2^{-\eta R}$, the following inequality holds for all $\mathbf{d} \in \mathcal{D}$:

$$\mathbf{z}^\top \mathbf{Ad} \leq \sigma \|\mathbf{Ad}\|_2\, t. \tag{44}$$

That is, with probability at least $1 - 2^{-\eta R}$ [1]:

$$\mathbf{z}^\top \mathbf{A}(\hat{\mathbf{x}} - \mathbf{x_c}) \leq \sigma \|\mathbf{A}(\hat{\mathbf{x}} - \mathbf{x_c})\|_2 t, \tag{45}$$

where $t \triangleq \sqrt{2(2+\eta)R \ln 2}$.

Substituting this bound into (31) yields:

$$\|\mathbf{A}(\hat{\mathbf{x}} - \mathbf{x})\|_2^2 \leq \|\mathbf{A}(\mathbf{x_c} - \mathbf{x})\|_2^2 + 2\sigma t \|\mathbf{A}(\hat{\mathbf{x}} - \mathbf{x_c})\|_2. \tag{46}$$

To further simplify, let $\mathbf{e} \triangleq \hat{\mathbf{x}} - \mathbf{x}$ and $\mathbf{e_c} \triangleq \mathbf{x_c} - \mathbf{x}$. Since $\hat{\mathbf{x}} - \mathbf{x_c} = \mathbf{e} - \mathbf{e_c}$, by the triangle inequality, we have

$$\|\mathbf{A}(\hat{\mathbf{x}} - \mathbf{x_c})\|_2 \leq \|\mathbf{Ae}\|_2 + \|\mathbf{Ae_c}\|_2. \tag{47}$$

---

[1] The probability guarantee differs from previous work focusing on denoising (Zafari et al., 2026) only in constant factors. Specifically, the additional constant factor appearing in expressions such as $2^{-\eta R+2}$ in previous work (Zafari et al., 2026) arises from the use of a union bound over two events together with a two-sided Gaussian tail bound. In contrast, our analysis employs a one-sided tail bound tailored to the more general linear inverse problem setting. This choice avoids unnecessary constant slack in the probability estimate while preserving the same error scaling and recovering the denoising result as a special case.

By replacing Eqn. (47) into Eqn. (46), we have:

$$\|\mathbf{Ae}\|_2^2 \leq \|\mathbf{Ae_c}\|_2^2 + 2\sigma t(\|\mathbf{Ae}\|_2 + \|\mathbf{Ae_c}\|_2). \tag{48}$$

Eqn. (48) can be rearranged and written as:

$$(\|\mathbf{Ae}\|_2 - \sigma t)^2 \leq (\|\mathbf{Ae_c}\|_2 + \sigma t)^2, \tag{49}$$

which implies

$$\|\mathbf{Ae}\|_2 \leq \|\mathbf{Ae_c}\|_2 + 2\sigma t. \tag{50}$$

Thus, we obtain the following measurement-domain error bound:

$$\boxed{\begin{aligned} \frac{1}{\sqrt{n}} \left\|\mathbf{A}(\hat{\mathbf{x}} - \mathbf{x})\right\|_2 &\leq \frac{1}{\sqrt{n}} \left\|\mathbf{A}(\mathbf{x_c} - \mathbf{x})\right\|_2 + 2\sigma\sqrt{\frac{2\ln 2\,(2+\eta)\,R}{n}} \\ &\leq \sqrt{\delta_A} + 2\sigma\sqrt{\frac{2\ln 2\,(2+\eta)\,R}{n}} \end{aligned}} \tag{51}$$

Finally, if $\|\mathbf{A}\|_{\mathrm{op}} \leq L$, recalling that $\|\mathbf{x} - \mathbf{x_c}\|_2^2 \leq n\delta$, it follows that $\|\mathbf{Ae_c}\|_2 \leq L\sqrt{n\delta}$. Consequently, we obtain

$$\boxed{\frac{1}{\sqrt{n}}\|\mathbf{A}(\hat{\mathbf{x}} - \mathbf{x})\|_2 \leq L\sqrt{\delta} + 2\sigma\sqrt{\frac{2\ln 2(2+\eta)R}{n}}.} \tag{52}$$

$\square$

*Remark* A.2 (One-sided vs. two-sided Gaussian tail bounds). In previous analysis over denoising task (Zafari et al., 2026), the proof employs a two-sided Gaussian tail bound over the form $\langle \mathbf{z}, \mathbf{d}/\|\mathbf{d}\|_2 \rangle \sim \mathcal{N}(0, \sigma^2)$, leading to probability bounds of the type $\Pr(|\langle \mathbf{z}, \mathbf{d}/\|\mathbf{d}\|_2 \rangle| \geq t) \leq 2\exp(-t^2/(2\sigma^2))$, where the inequalities involving absolute values arise from intermediate steps Eqn. (27) to (28). Notably, here two union-bound operations are operated.

In contrast, our analysis focuses on $\Pr(\langle \mathbf{z}, \mathbf{Ad}/\|\mathbf{Ad}\|_2 \rangle \geq t) \leq \exp(-t^2/(2\sigma^2))$, using a one-sided Gaussian tail bound $\mathbb{P}(\langle \mathbf{z}, \mathbf{u} \rangle \geq t) \leq \exp(-t^2/(2\sigma^2))$, with one union-bound operation introduced. This makes a difference of factor 2 before the exponentiation operations.

For intuition, we compare the true codeword $\mathbf{x}$ with a candidate $\mathbf{c} \in \mathcal{C}$. A decoding error occurs only if: $\|\mathbf{y} - \mathbf{Ac}\|_2^2 \leq \|\mathbf{y} - \mathbf{Ax}\|_2^2$. Substituting $\mathbf{y} = \mathbf{Ax} + \mathbf{z}$ and simplifying yields the equivalent condition $2\langle \mathbf{z}, \mathbf{A}(\mathbf{c} - \mathbf{x}) \rangle \geq \|\mathbf{A}(\mathbf{c} - \mathbf{x})\|_2^2$. Letting $\mathbf{u} = \mathbf{A}(\mathbf{c} - \mathbf{x})$, the error event becomes $\langle \mathbf{z}, \mathbf{u} \rangle \geq \frac{1}{2}\|\mathbf{u}\|_2^2$. Since $\langle \mathbf{z}, \mathbf{u} \rangle \sim \mathcal{N}(0, \sigma^2\|\mathbf{u}\|_2^2)$ (see Eqn. (33)), a decoding error requires an unusually large positive noise deviation along $\mathbf{u}$, so only one-sided Gaussian tail bound suffices.

### A.3. Proof and analysis of Corollary 3.3

*Proof of Corollary 3.3.* Setting $\mathbf{A} = \mathbf{I}_n$ in Theorem 3.1 gives $\|\mathbf{A}(\hat{\mathbf{x}} - \mathbf{x})\|_2 = \|\hat{\mathbf{x}} - \mathbf{x}\|_2$. Moreover, $\delta_A = \sup_{\mathbf{x} \in \mathcal{Q}} \frac{1}{n}\|\mathbf{A}(g(f(\mathbf{x})) - \mathbf{x})\|_2^2 = \sup_{\mathbf{x} \in \mathcal{Q}} \frac{1}{n}\|g(f(\mathbf{x})) - \mathbf{x}\|_2^2 = \delta$. Substituting these into the theorem yields the conclusion. $\square$

We compare the denoising specialization of our measurement-domain bound with the corresponding result in previous zero-shot compression-based denoising literature (Zafari et al., 2026).

**Our bound.** In the denoising setting $\mathbf{A} = \mathbf{I}_n$ (hence $\delta_A = \delta$), for any $\eta > 0$, with probability at least $1 - 2^{-\eta R}$,

$$\frac{1}{\sqrt{n}}\|\hat{\mathbf{x}} - \mathbf{x}\|_2 \leq \sqrt{\delta} + 2\sigma\sqrt{\frac{2\ln 2\,(2+\eta)\,R}{n}}. \tag{53}$$

**Previous result.** The denoising bound reported in (Zafari et al., 2026) can be written as: for any $\tilde{\eta} > 0$, with probability at least $1 - 2^{-\tilde{\eta}R+2}$,

$$\frac{1}{\sqrt{n}}\|\hat{\mathbf{x}} - \mathbf{x}\|_2 \leq \sqrt{\delta} + 2\sigma\sqrt{\frac{2\ln 2\,R}{n}}\left(1 + 2\sqrt{\tilde{\eta}}\right). \tag{54}$$

To compare the two bounds under the *same* scenario, we parameterize both results by a common $\eta > 0$ via

$$\tilde{\eta} = \eta + \frac{2}{R}. \tag{55}$$

Then, we can have $1 - 2^{-\tilde{\eta}R+2} = 1 - 2^{-(\eta+2/R)R+2} = 1 - 2^{-\eta R}$, so both (53) and (54) hold with probability at least $1 - 2^{-\eta R}$ after substituting (55) into (54).

After unifying the probability, both bounds share the same distortion term $\sqrt{\delta}$ and the same scaling $\sigma\sqrt{R/n}$, and differ only in the confidence-dependent constants. Define

$$C_{\text{ours}}(\eta) = \sqrt{2 + \eta}, \qquad C_{\text{pre}}(\eta, R) = 1 + 2\sqrt{\eta + \frac{2}{R}}. \tag{56}$$

Under the same confidence level $1 - 2^{-\eta R}$, our bound is tighter if and only if

$$C_{\text{ours}}(\eta) \leq C_{\text{pre}}(\eta, R) \quad \Longleftrightarrow \quad \sqrt{2 + \eta} \leq 1 + 2\sqrt{\eta + \frac{2}{R}}. \tag{57}$$

**Regime analysis.** Since $C_{\text{pre}}(\eta, R)$ is minimized as $R \to \infty$, the most stringent condition is obtained in this limit, yielding

$$\sqrt{2 + \eta} \leq 1 + 2\sqrt{\eta}. \tag{58}$$

This inequality holds if and only if

$$\eta \geq \eta^{\star} \triangleq \frac{11 - 4\sqrt{7}}{9} \approx 0.0463. \tag{59}$$

Consequently, the following regimes can be identified:

- **High-confidence regime.** If $\eta \geq \eta^{\star}$, then $C_{\text{ours}}(\eta) \leq C_{\text{pre}}(\eta, R)$ for all $R > 0$, and our bound is tighter.

- **Low-confidence, small-rate regime.** If $\eta < \eta^{\star}$, then (57) holds provided that

$$R \leq \frac{8}{3(1 - \eta) - 2\sqrt{2 + \eta}}, \tag{60}$$

  and our bound remains tighter.

- **Low-confidence, large-rate regime.** Only if $\eta < \eta^{\star}$ and

$$R > \frac{8}{3(1 - \eta) - 2\sqrt{2 + \eta}}, \tag{61}$$

  then $C_{\text{ours}}(\eta) > C_{\text{pre}}(\eta, R)$, and the bound in (Zafari et al., 2026) becomes tighter.

**Summary.** Our theoretical results provide a more general framework for a compression-based solver, which naturally specializes to the denoising setting and recovers the error bound of the same two-term structure as prior results, consisting of a distortion term and a rate-dependent term. Their relative tightness depends on the confidence parameter $\eta$ and the compression rate $R$. In particular, our bound is tighter under high-confidence requirements or small compression rates, which are typical operating regimes in zero-shot denoising to mitigate noise overfitting, whereas for large rates and low-confidence requirements the bound in (Zafari et al., 2026) is tighter.

## A.4. Proof of Corollary 3.5

**Recall of Corollary 3.5 [Signal-domain error bound for compression-based solver in linear inverse problem]:** Assume the conditions of Theorem 3.1. In addition, suppose that there exists $\alpha > 0$ such that

$$\|\mathbf{Ad}\|_2 \geq \alpha\|\mathbf{d}\|_2, \qquad \forall \mathbf{d} \in \mathcal{D}, \tag{62}$$

where $\mathcal{D} \triangleq \{c_1 - c_2 : c_1, c_2 \in \mathcal{C}\}$. Then, for any $\eta > 0$, with probability at least $1 - 2^{-\eta R}$, it holds:

$$\boxed{\frac{1}{\sqrt{n}}\|\hat{\mathbf{x}} - \mathbf{x}\|_2 \leq \sqrt{\delta} + \frac{2}{\alpha}\sqrt{\delta_A} + \frac{2\sigma}{\alpha}\sqrt{\frac{2\ln 2(2+\eta)R}{n}}.} \tag{63}$$

In particular, if $\|\mathbf{A}\|_{\mathrm{op}} \leq L$, then $\delta_A \leq L^2\delta$ and hence

$$\boxed{\frac{1}{\sqrt{n}}\|\hat{\mathbf{x}} - \mathbf{x}\|_2 \leq \left(1 + \frac{2L}{\alpha}\right)\sqrt{\delta} + \frac{2\sigma}{\alpha}\sqrt{\frac{2\ln 2(2+\eta)R}{n}}.} \tag{64}$$

*Proof of Corollary 3.5.* Recall that $\mathbf{x_c} = g(f(\mathbf{x})) \in \mathcal{C}$ denotes the reconstruction of $\mathbf{x}$ by the compression code. By the triangle inequality,

$$\|\hat{\mathbf{x}} - \mathbf{x}\|_2 \leq \|\hat{\mathbf{x}} - \mathbf{x_c}\|_2 + \|\mathbf{x_c} - \mathbf{x}\|_2. \tag{65}$$

Since $\mathbf{x_c} \in \mathcal{C}$ and $\hat{\mathbf{x}} \in \mathcal{C}$, their difference belongs to the codeword-difference set $\mathcal{D}$. Thus, by the restricted injectivity assumption,

$$\|\hat{\mathbf{x}} - \mathbf{x_c}\|_2 \leq \frac{1}{\alpha}\|\mathbf{A}(\hat{\mathbf{x}} - \mathbf{x_c})\|_2. \tag{66}$$

Moreover, from triangle inequality, we have

$$\|\mathbf{A}(\hat{\mathbf{x}} - \mathbf{x_c})\|_2 \leq \|\mathbf{A}(\hat{\mathbf{x}} - \mathbf{x})\|_2 + \|\mathbf{A}(\mathbf{x} - \mathbf{x_c})\|_2. \tag{67}$$

Combining (65)–(67) yields:

$$\|\hat{\mathbf{x}} - \mathbf{x}\|_2 \leq \frac{1}{\alpha}\|\mathbf{A}(\hat{\mathbf{x}} - \mathbf{x})\|_2 + \frac{1}{\alpha}\|\mathbf{A}(\mathbf{x} - \mathbf{x_c})\|_2 + \|\mathbf{x_c} - \mathbf{x}\|_2. \tag{68}$$

Now, based on Theorem 3.1 (Eqn. (50)), we have:

$$\frac{1}{\sqrt{n}}\|\mathbf{A}(\hat{\mathbf{x}} - \mathbf{x})\|_2 \leq \frac{1}{\sqrt{n}}\|\mathbf{A}(\mathbf{x_c} - \mathbf{x})\|_2 + 2\sigma\sqrt{\frac{2\ln 2(2+\eta)R}{n}}. \tag{69}$$

Substituting (69) into (68) and dividing by $\sqrt{n}$, we obtain

$$\frac{1}{\sqrt{n}}\|\hat{\mathbf{x}} - \mathbf{x}\|_2 \leq \frac{1}{\alpha}\left[\frac{1}{\sqrt{n}}\|\mathbf{A}(\mathbf{x_c} - \mathbf{x})\|_2 + 2\sigma\sqrt{\frac{2\ln 2(2+\eta)R}{n}}\right] + \frac{1}{\alpha}\frac{1}{\sqrt{n}}\|\mathbf{A}(\mathbf{x} - \mathbf{x_c})\|_2 + \frac{1}{\sqrt{n}}\|\mathbf{x_c} - \mathbf{x}\|_2$$
$$= \frac{2}{\alpha} \cdot \frac{1}{\sqrt{n}}\|\mathbf{A}(\mathbf{x_c} - \mathbf{x})\|_2 + \frac{2\sigma}{\alpha}\sqrt{\frac{2\ln 2(2+\eta)R}{n}} + \frac{1}{\sqrt{n}}\|\mathbf{x_c} - \mathbf{x}\|_2. \tag{70}$$

Finally, using the signal-domain distortion definition,

$$\frac{1}{\sqrt{n}}\|\mathbf{x_c} - \mathbf{x}\|_2 \leq \sqrt{\delta}, \tag{71}$$

we can have:

$$\boxed{\begin{aligned} \frac{1}{\sqrt{n}}\|\hat{\mathbf{x}} - \mathbf{x}\|_2 &\leq \frac{2}{\alpha} \cdot \frac{1}{\sqrt{n}}\|\mathbf{A}(\mathbf{x_c} - \mathbf{x})\|_2 + \frac{2\sigma}{\alpha}\sqrt{\frac{2\ln 2(2+\eta)R}{n}} + \sqrt{\delta} \\ &= \frac{2}{\alpha}\sqrt{\delta_A} + \frac{2\sigma}{\alpha}\sqrt{\frac{2\ln 2(2+\eta)R}{n}} + \sqrt{\delta} \end{aligned}} \tag{72}$$

Considering that the operator-norm bound $\|\mathbf{A}\|_{\mathrm{op}} \leq L$ gives

$$\frac{1}{\sqrt{n}}\|\mathbf{A}(\mathbf{x_c} - \mathbf{x})\|_2 \leq \|\mathbf{A}\|_{\mathrm{op}} \cdot \frac{1}{\sqrt{n}}\|\mathbf{x_c} - \mathbf{x}\|_2 \leq L\sqrt{\delta}, \tag{73}$$

we obtain:

$$\frac{1}{\sqrt{n}}\|\hat{\mathbf{x}} - \mathbf{x}\|_2 \leq \frac{2}{\alpha} \cdot \frac{1}{\sqrt{n}}\|\mathbf{A}(\mathbf{x_c} - \mathbf{x})\|_2 + \frac{2\sigma}{\alpha}\sqrt{\frac{2\ln 2(2+\eta)R}{n}} + \sqrt{\delta}$$
$$\leq (1 + \frac{2L}{\alpha})\sqrt{\delta} + \frac{2\sigma}{\alpha}\sqrt{\frac{2\ln 2(2+\eta)R}{n}} \tag{74}$$

$\square$

## B. Implementation details

### B.1. Dataset and baseline

**Dataset.** Following (Zafari et al., 2026), we evaluate denoising performance under AWGN and Poisson noise on Set11 (Zhang et al., 2017), Set13 (Zeyde et al., 2010) ($192 \times 192$ crops), and Kodak24 (Kodak, 1991). Specifically, the Set11 dataset consists of 11 grayscale images (C.man, House, Peppers, Starfish, Monarch, Airplane, Parrot, Barbara, Boats, Pirate, and Couple), where seven images have a resolution of $256 \times 256$, and the remaining four images have a resolution of $512 \times 512$. Set13 dataset consists of 13 images (Baboon, Barbara, Bridge, Coastguard, Comic, Face, Flowers, Foreman, Man, Monarch, Peppers, PPT3, and Zebra), where each image is center-cropped into $192 \times 192$ patches. Kodak dataset consists of 24 images, each with a resolution of $512 \times 768$. For super-resolution and inpainting, we follow (Ulyanov et al., 2020) using Set5 (Bevilacqua et al., 2012) and other images as in their original papers.

**Baseline.** For denoising, we compare our approach against representative zero-shot solvers, including both traditional and learning-based methods. All baselines are dataset-free and require no external training. For non-learning methods, we include JPEG2K and BM3D. For learning-based methods, we evaluate Deep Image Prior (DIP) (Ulyanov et al., 2020), Deep Decoder (DD) (Heckel & Hand, 2019), Zero-Shot Noise2Self (ZS-N2S) (Batson & Royer, 2019), Self2Self (S2S) (Quan et al., 2020), Zero-Shot Noise2Noise (ZS-N2N) (Mansour & Heckel, 2023), and Zero-Shot Neural Compression Denoiser (ZS-NCD) (Zafari et al., 2026). The performance comparison is summarized in Table 1. The baseline results were obtained using their official implementations. For alternative architectures, we employ their optimal results or use the better result between our reproduction and the officially reported results from their original papers/codes.

For noisy super-resolution, we compare against representative classical and zero-shot baselines, including Bicubic interpolation, No-Prior optimization, BM3D, BM3D+Bicubic, DIP, BM3D+LapSRN, and ZS-NCD+LapSRN (Lai et al., 2017). For inpainting, DIP serves as the primary zero-shot baseline. For text removal, we additionally compare with Shepard-Net (Ren et al., 2015). For region inpainting, we include a representative generative-model-based approach, Local–Global GAN (Iizuka et al., 2017), to assess performance against methods with explicit generative priors.

### B.2. Model architecture

We summarize the detailed architecture of the employed implicit codec in Table 7. For the auto-regressive entropy model, we use 32 contextual elements as input and adopt a three-layer network composed of linear (or $1 \times 1$ convolutional) layers with GELU activations. The network follows the configuration: $(32 \times 32) \rightarrow \text{GELU} \rightarrow (32 \times 32) \rightarrow \text{GELU} \rightarrow (32 \times 32)$.

For the modulation network $f_\theta$, we set the number of synthesis layers to $L_t = 4$ and use $L = 7$ multi-resolution latent components with an $8 \times 8$ upsampling kernel. The network maps the upsampled latent features through a lightweight MLP with dimensions: $(7 \times 48) \rightarrow \text{GELU} \rightarrow (48 \times 3) \rightarrow \text{GELU} \rightarrow (3 \times 3)$.

The randomized network takes pixel coordinates as input and consists of $L_t = 4$ layers. Coordinates are first embedded using Fourier-feature positional encoding, yielding a 24-dimensional representation, which is concatenated with the original coordinates to form a 26-dimensional input. The network architecture is: $(26 \times 32) \rightarrow \text{GELU} \rightarrow ([32 + 3 + 3] \times 24) \rightarrow \text{GELU} \rightarrow ([24 + 3 + 3 + 3] \times 16) \rightarrow ([16 + 3 + 3 + 3 + 48] \times 3) \rightarrow \text{Tanh}$, where layer-wise modulation features produced by the modulation network are concatenated in a rewind fashion. This design enables efficient structure-aware modulation while keeping the network lightweight.

### B.3. Fourier initializations for random network

We adopt the Fourier-based initialization (Shi et al., 2024) to facilitate the identification of a *Lottery Prior* that mitigates the low-frequency bias of MLPs and preserves informative sign patterns while maintaining stable variance (Zhou et al., 2019; Glorot & Bengio, 2010; He et al., 2016). Concretely, weight matrix $\mathbf{W}^{(\mathbf{i})} \in \mathbb{R}^{d_i \times d_{i-1}}$ of each $i$ layer is reparameterized as $\mathbf{W}^{(\mathbf{i})} = \mathbf{\Lambda}^{(\mathbf{i})} \mathbf{B}^{(\mathbf{i})}$, where $\mathbf{\Lambda}^{(\mathbf{i})} \in \mathbb{R}^{d_i \times M}$ denotes the coefficient matrix, and $\mathbf{B}^{(\mathbf{i})} \in \mathbb{R}^{M \times d_{i-1}}$ contains $M$ Fourier bases, defined by distinct frequency–phase pairs. We employ $P$ phase shifts and both low- and high-frequency components, yielding $M = P(F_{\text{high}} + F_{\text{low}})$. Throughout this paper, we set $P = 32$, $F_{\text{high}} = 256$, and $F_{\text{low}} = 32$.

### B.4. Training details

We select $\lambda$ empirically according to the degradation level ($\mathbf{A}$ is given), with more severe degradations requiring stronger rate constraints (higher compression). In principle, degradation levels can be estimated using standard blind restoration techniques (You & Dragotti, 2024).

For denoising, we adopt the Lagrangian coefficient search strategy of (Zafari et al., 2026). Under the AWGN model $\mathbf{y} = \mathbf{x} + \mathbf{z}$, we have $\frac{1}{n}\mathbb{E}\|\mathbf{y} - \hat{\mathbf{x}}\|_2^2 = \sigma^2 + \frac{1}{n}\mathbb{E}\|\mathbf{x} - \hat{\mathbf{x}}\|_2^2 + \frac{2}{n}\mathbb{E}[\mathbf{z}^\top(\mathbf{x} - \hat{\mathbf{x}})]$. In effective denoising regimes the cross term is negligible, so the residual energy $\frac{1}{n}\|\mathbf{y} - \hat{\mathbf{x}}\|_2^2$ is close to the noise variance. We therefore choose $\lambda$ by matching the residual to an estimate of $\sigma^2$.

For noisy super-resolution and inpainting, we find that the method is robust to the choice of $\lambda$, and thus a single $\lambda$ is used for each degradation level.

We report $\lambda$ in later visualization section C and data-points section E. Other hyperparameters are reported in Table 7.

### B.5. Ensemble of priors

A simple way to aggregate priors (obtained from different random seeds and mask ratios) is to directly average them, which achieves state-of-the-art performance for compression-based zero-shot denoising and noisy super-resolution.

We also provide a linear averaging approach for improved denoising performance. Given a set of candidate reconstructions $\{\hat{\mathbf{x}}_i\}_{i=1}^k$, we evaluate their measurement consistency with respect to the noisy observation $\mathbf{y}$. This is quantified by the mean squared error in the denoising case: $e_i = \|\hat{\mathbf{x}}_i - \mathbf{y}\|_2^2$.

We then compute aggregation weights via a gating mechanism,

$$\gamma_i = \frac{\exp(-\tau (e_i - \min_j e_j))}{\sum_{j=1}^{K} \exp(-\tau (e_j - \min_\ell e_\ell))}, \tag{75}$$

where the temperature parameter is set to $\tau = 50$ to control the sharpness of the selection. This yields a prior-aggregated estimate $\bar{\mathbf{x}} = \sum_{i=1}^{K} \gamma_i \hat{\mathbf{x}}_i$. Finally, a data-anchoring proximal step balances fidelity to the noisy observation and the aggregated prior:

$$\mathbf{x}^* = \frac{\mathbf{y} + \lambda\bar{\mathbf{x}}}{1 + \lambda}, \tag{76}$$

where $\lambda = 30$ controls the effect of the ensemble prior. For more details, we refer to our project page. Note that we use simple linear aggregation for denoising, which is sufficient to achieve strong performance; more advanced aggregation strategies are left for future work. To show the effect of the ensemble, we provide a visualization in Fig. 7.

### B.6. Pseudocode for the algorithm

This section provides a detailed learning and inference algorithm of Lottery Prior, as shown in Algorithm 1 and Algorithm 2.

*Table 7.* Hyper-parameter settings.

| Hyper parameter | Initial values | Final values |
|---|---|---|
| **Quantization – Stage I** | | |
| Number of learning steps | $5 * 10^4$ | |
| Learning rate $\beta$ | $10^{-2}$ | 0 |
| Scheduler for learning rate | Cosine scheduler | |
| Temperature $T$ for soft rounding | 0.3 | 0.1 |
| Noise strength $\alpha$ for Kumaraswamy noise | 2.0 | 1.0 |
| Scheduler for Soft-rounding and Kumaraswamy noise | Linear scheduler | |
| **Quantization – Stage II** | | |
| Number encoding steps | $10^4$ | |
| Learning rate | $10^{-4}$ | $10^{-8}$ |
| Decay learning rate if loss has not improved for this many steps | 40 | |
| Decay factor | 0.8 | |
| Temperature $T$ for soft rounding | $10^{-4}$ | |
| **Architecture – Latent modulations** | **Values** | |
| Number of latent vectors $L$ | 7 | |
| Initialization of $\mathbf{z}$ | 0 | |
| **Architecture – ModNet** | **Values** | |
| Output channels of the $1 \times 1$ convolutions | $\{48, 3\}$ | |
| Number of $3 \times 3$ residual convolutions | 2 | |
| **Architecture – Entropy model** | | |
| Output channels of $1 \times 1$ convolutions | $\{32, 32, 2\}$ | |
| Activation function | GELU | |
| Log-scale of Laplace is shifted before $\exp$ | 4 | |
| Scale parameter of Laplace is clipped to | $[10^{-2}, 150]$ | |
| **Architecture – randomized network** | **Values** | |
| Output dimensions of MLP layers | $\{32, 24, 16, 3\}$ | |
| Mask ratios | $\{0.2, 0.8\}$ with 0.05 intervals | |
| Initialization of MLP | FFN initialization | |
| FFN initialization phase number $P$ | 32 | |
| FFN initialization low/high frequency number $F$ | 32/256 | |
| Initialization of score matrix $\mathbf{P}$ | Kaiming uniform initialization | |
| Learning rate $\alpha$ for $\mathbf{P}$ | 0.1 | |
| Scheduler | Cosine scheduler | |

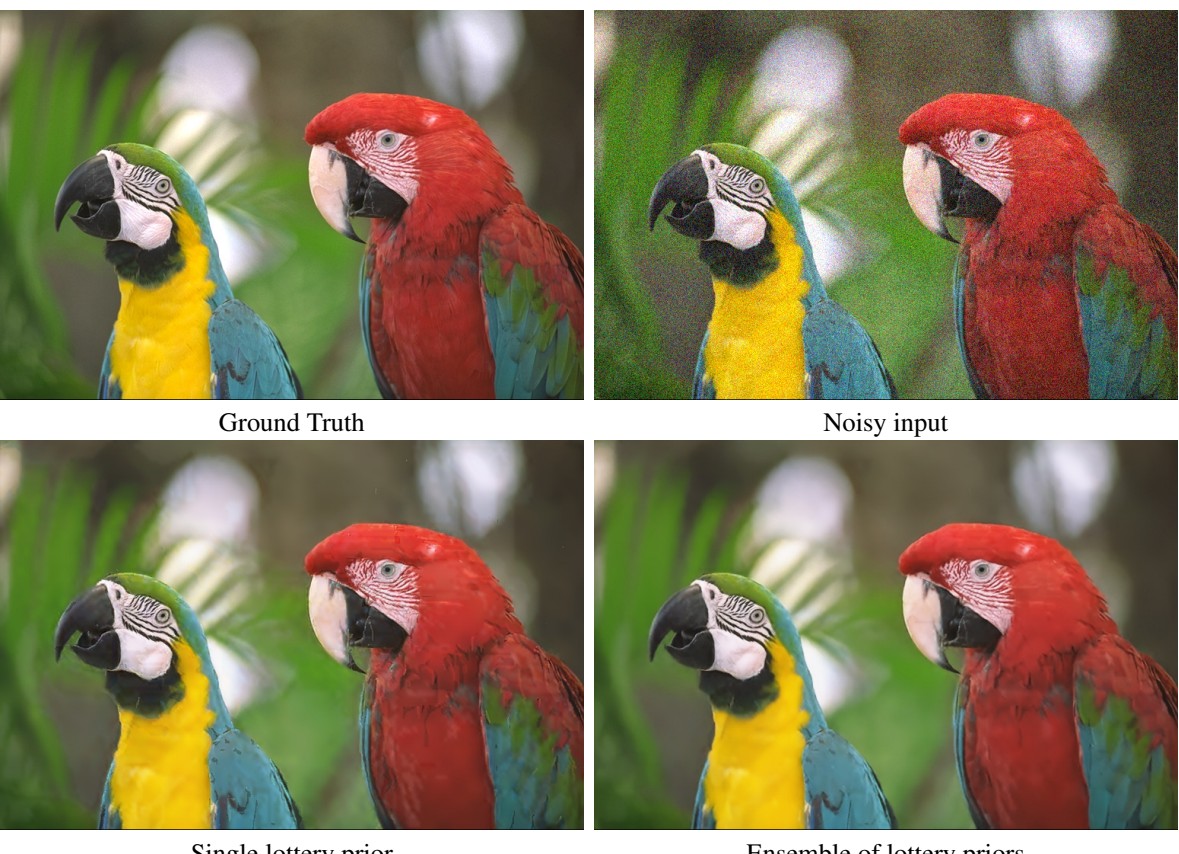

*Figure 7.* Denoising on Kodak under AWGN ($\sigma = 25$). Prior ensembling improves restoration and effectively suppresses artifacts.

---

**Algorithm 1** Zero-shot learning stage of the Lottery Prior

---

**Input:** Observed measurement $\mathbf{y}$, degradation operator $\mathbf{A}$, random seed $e$,
learning rates $\alpha$ for mask scores ($\mathbf{P}$),
learning rates $\beta$ for modulations ($\mathbf{z}$), and networks ($\theta$ and $\psi$),
cosine scheduler for learning rate for Stage I and II: $\mathcal{C}_1, \mathcal{C}_2$,
linear scheduler for soft-rounding temperature and Kumaraswamy noise strength for Stage I and II: $l_1, l_2$,
**Output:** $\hat{\mathbf{Z}}, \theta, \tau$.

---

1:    $\mathbf{w_0} = \mathbf{\Lambda B}, \mathbf{P} \sim \mathcal{U}_k$,                *// Initialization for $g(\cdot)$ and $\mathbf{P}$ based on $e$*
2: **for** the $i$-th step within the Stage I **do**
3:      $\tau = h(\mathbf{P}), g = g_{\tau \odot \mathbf{w_0}}$          *// Configure network by masking $k\%$ weights based on $\mathbf{P}$*
4:      $\hat{\mathbf{Z}} = \mathcal{S}_T(\mathbf{Z}) + \mathbf{u}_{kum}$,               *// Quantization-aware training*
5:      $\hat{\mathbf{x}} = g_{\tau \odot \mathbf{w_0}}(f_\theta(\hat{\mathbf{Z}}))$,             *// Modulate the reconstruction*
6:      $\mathcal{L} = D(\mathbf{y}, \mathbf{A}\hat{\mathbf{x}}) + \lambda R_\psi(\hat{\mathbf{Z}})$,        *// Compute the RD cost loss function*
7:      $\mathbf{P} \leftarrow \mathbf{P} - \alpha \nabla_{\mathbf{P}} \mathcal{L}$,               *// Update the scores for mask*
8:      $\mathbf{Z} \leftarrow \mathbf{Z} - \beta \nabla_{\mathbf{Z}} \mathcal{L}$,               *// Update latent modulations*
9:      $\theta \leftarrow \theta - \beta \nabla_\theta \mathcal{L}$,                *// Update modulation net*
10:     $\psi \leftarrow \psi - \beta \nabla_\psi \mathcal{L}$,                  *// Update ARM*
11:     $\alpha = \mathcal{C}_1(\alpha, i), \beta = \mathcal{C}_1(\beta, i)$           *// Update learning rate*
12:     $T = l_1(T, i), u_{kum} = l_1(u_{kum}, i)$       *// Update noise strength*
13: **end for**
14: **for** the $i$-th step within the Stage II **do**
15:     $\tau = h(\mathbf{P}), g = g_{\tau \odot \mathbf{w_0}}$          *// Mask $k\%$ weights based on updated $\mathbf{P}$*
16:     $\hat{\mathbf{Z}} = Q(\mathbf{Z})$,                       *// Hard rounding*
17:     $\hat{\mathbf{x}} = g_{\tau \odot \mathbf{w_0}}(f_\theta(\hat{\mathbf{Z}}))$,            *// Modulate the reconstruction*
18:     $\mathcal{L} = D(\mathbf{y}, \mathbf{A}\hat{\mathbf{x}}) + \lambda R_\psi(\hat{\mathbf{Z}})$,       *// Compute the RD cost loss*
19:     $\mathbf{P} \leftarrow \mathbf{P} - \alpha \nabla_{\mathbf{P}} \mathcal{L}$,               *// Update the scores for mask*
20:     $\mathbf{Z} \leftarrow \mathbf{Z} - \beta \nabla_{\mathbf{z}} \mathcal{L}$,               *// Update latent modulations*
21:     $\theta \leftarrow \theta - \beta \nabla_\theta \mathcal{L}$,                 *// Update modulation net*
22:     $\psi \leftarrow \psi - \beta \nabla_\psi \mathcal{L}$,                  *// Update ARM*
23:     $\alpha = \mathcal{C}_2(\alpha, i), \beta = \mathcal{C}_2(\beta, i)$           *// Update learning rate*
24:     $T = l_2(T, i), u_{kum} = l_2(u_{kum}, i)$       *// Update noise strength*
25: **end for**
26: $\Omega = \{e, \hat{\mathbf{Z}}, \theta, \tau\}$,                     *// Save the parameter set*

---

**Algorithm 2** Restoration stage of *Lottery Prior*

---

**Input:** Random seed $e$, and a well-trained set $\{\hat{\mathbf{Z}}, \theta, \tau\}$.
**Output:** Reconstruction of image $\hat{\mathbf{x}}$.

---

1:    $\mathbf{w_0} = \mathbf{\Lambda B}, g = g_{\tau \odot \mathbf{w_0}}$         *// Configure randomized network with a given seed $e$*
2:    $\hat{\mathbf{x}} = g_{\tau \odot \mathbf{w_0}}(f_\theta(\hat{\mathbf{Z}}))$,              *// Reconstruct the source*

---

## C. Visualization

### C.1. Super-resolution with noisy observation

We visualize $\times 4$ ($\lambda = 7e - 2$) and $\times 8$ ($\lambda = 4e - 1$) super-resolution results on Set5 under additive white Gaussian noise ($\sigma = 15$) in Fig. 8 and Fig 9, respectively. Our method achieves high reconstruction quality and produces clean results comparable to the BM3D+LapSRN pipeline, which uses a supervised super-resolution network.

### C.2. Inpainting

For text-removal inpainting, we employ $\lambda = 8e - 4$. For random-mask inpainting, we employ $\lambda = 1e - 4$. We visualize our inpainting results with different mask ratios ($\rho$) on Set5 in Fig. 10. Our approach yields cleaner textures and fewer artifacts, especially under the heavier masking setting. We also evaluate our approach on region inpainting in Figs 11 ($\lambda = 5e - 3$) and 12 ($\lambda$ reported in the figure), highlighting the effectiveness of our approach.

### C.3. Denoising

We visualize denoising results on Kodak under additive white Gaussian noise ($\sigma = 15, 25$) in Figs. 13 and 14, respectively. We further present Kodak results under Poisson noise ($\alpha = 15$) in Fig. 15, and Set11 results under additive white Gaussian noise ($\sigma = 25$) in Fig. 16. Our approach achieves high fidelity and produces clean results comparable to the supervised Restormer approach.

## D. Ablation studies and more experiments

### D.1. Ablation study

**Model component.** We conduct ablation studies over each component (Kodak01 with AWGN $\sigma = 25$). We sequentially remove each component from the proposed Lottery Prior, with results reported in Table 8. The ablation over the ensemble mechanism demonstrates the importance of flexible priors generated from our Lottery Priors. Ablation on the randomized network without the modulation network shows the pure effect of the entropy regularizer and the network priors, revealing that the proposed modulation mechanism can further enhance performance. We also found that lightweight ARM here can result in slightly worse performance.

*Table 8.* Ablation study on different mechanisms, conducted on the Kodak01 image using an ARM-32 model and a mask ratio of 50%.

| Model Variant | Denoising PSNR |
|---|---|
| *Lottery Prior* scheme | 29.15 dB |
| $\Rightarrow$ Single prior w/o ensemble (ratio=0.5) | 27.18 dB |
| $\Rightarrow$ randomized network w/o modulation network | 26.78 dB |
| $\Rightarrow$ cheaper ARM with dimension 24 | 26.89 dB |
| $\Rightarrow$ Single prior w/o entropy model | Not work |

Furthermore, we conducted ablations on optimizer choice (Adam/AdamW/RMSProp) and architectural configurations (backbone width) to assess sensitivity, using the first three images of Set11 for denoising ($\sigma = 15$, mask ratio 0.5).

**Optimizers.** As shown in Table 9, across the optimizers (Adam, AdamW, RMSProp), performance varies by only 0.12 dB, with only minor differences in convergence speed, confirming that the method is robust to optimizer choice. We therefore adopt the default Adam optimizer.

**Architecture.** We vary the random network backbone width ($0.5 \times / 1 \times / 2 \times$). As shown in Table 10, narrower models slightly underfit, while wider models yield only marginal gains or may overfit (across a $4 \times$ range in backbone width, performance varies by only 0.06 dB, indicating that the rate constraint, not the specific architecture, is the primary regularizer, consistent with the minimum description length principle and our theoretical framework. From an optimization perspective,

an appropriate model size is preferred, neither too small nor excessively large. We want to highlight that under the randomized network setting, different architectures can induce a family of random codec priors; judiciously combining multiple such priors may further improve performance, which we leave for future work.

*Table 9.* Ablation over different optimizers.

| Lottery Prior | Adam | AdamW | RMSprop |
|---|---|---|---|
| PSNR(dB) | 32.60 | 32.59 | 32.48 |
| SSIM | 0.8834 | 0.8828 | 0.8792 |

*Table 10.* Ablation over the width of random network.

| Width of each layer | $\times0.5$ | $\times1$ | $\times2$ | Combine Priors |
|---|---|---|---|---|
| PSNR(dB) | 32.54 | 32.60 | 32.59 | 32.90 |
| SSIM | 0.8826 | 0.8834 | 0.8820 | 0.8872 |

### D.2. Additional experiments over super-resolution tasks

Additional metrics for the super-resolution task are reported in Table 11.

*Table 11.* Average PSNR / MS-SSIM/ LPIPS on Set5 with AWGN ($\sigma = 15$) for the super-resolution tasks.

|  | $4\times$ super-resolution | $8\times$ super-resolution |
|---|---|---|
| Methods | PSNR / MS-SSIM / LPIPS | PSNR / MS-SSIM / LPIPS |
| No prior | 20.10/0.7125/0.5682 | 18.66/0.6102/0.6037 |
| Bicubic | 23.32/0.8129/0.6129 | 20.91/0.7158/0.7433 |
| BM3D+Bicubic | 24.99/0.9078/0.4316 | 21.62/0.7795/0.6296 |
| Deep image prior | 21.37/0.7610/0.5064 | 19.55/0.6460/0.6443 |
| **Lottery prior** | **25.87/0.9182/0.3042** | **22.07/0.7951/0.5180** |
| LapSRN | 24.21/0.8465/0.4158 | 21.30/0.7418/0.5787 |
| BM3D+LapSRN | 25.17 /0.9096/0.3392 | 21.66/0.7803/0.5189 |

### D.3. Additional experiments over modality and foundation models

**Demosaicing and microscopy image.** We include additional comparisons with a representative foundation model, RAM. We evaluate on Mouse Nuclei fluorescence microscopy denoising (Buchholz et al., 2020)(Table 12) and natural-image demosaicing (Table 4), both unseen during RAM's pretraining. For fine-tuning, RAM uses an additional N=10 measurements, following Table 3 of (Terris et al., 2025a). Our experiments show that RAM achieves strong performance without fine-tuning across unseen tasks and modalities. In particular, it is competitive with BM3D on unseen-modality denoising and also achieves strong results on the unseen task of demosaicing, benefiting from its incorporation of forward-operator knowledge and extensive pretraining across diverse modalities and restoration tasks. In contrast, Lottery Prior does not rely on any external data or pretraining, yet still achieves strong performance on both tasks, outperforming pretrained RAM by an average of 0.85 dB on microscopy denoising and 0.84 dB on demosaicing, which validates the generality of the proposed method. At the same time, RAM improves substantially after few-shot self-supervised fine-tuning (N=10), highlighting the advantage of pretrained priors combined with fine-tuning. We note that RAM is not directly comparable to our setting, as it relies on extensive supervised pretraining across diverse datasets and relevant tasks, whereas our method is purely dataset-free and instance-level. We therefore present these results to calibrate our method against pretrained and finetuned foundation-model priors, highlighting the different assumptions in prior knowledge and training resources.

**MRI.** To assess our method in a medical imaging setting, we conduct experiments on fastMRI (Zbontar et al., 2018; Knoll et al., 2020) single-coil knee reconstruction (320x320-cropped Knee-file1000000 as an example) under $4\times$ and $8\times$ acceleration. The undersampled measurements follow $y = M\mathcal{F}(x) + n$, where $\mathcal{F}$ denotes the Fourier transform, $M$ the sampling mask, $n$ the acquisition noise ($n = 0$ in our experiments). In this setting, the complex-valued image is

*Table 12.* Modality adaptation experiments of the foundation models and Lottery Prior on Mouse Nuclei fluorescence microscopy image denoising (first 5 samples from the dataset for validation, and randomly sampled 10 from the remaining as a fine-tuning set). Reported values are the average PSNR, and Lottery Prior is averaged with 3 priors, each with 10k steps.

| Noise level | BM3D | Lottery Prior | RAM (Pretrained) | RAM (Finetuned) |
|---|---|---|---|---|
| $\sigma = 15$ | 33.91 | **34.13** | 33.89 | 34.98 |
| $\sigma = 25$ | 29.85 | **30.42** | 29.57 | 29.91 |

parameterized by a single lottery prior and optimized under a $k$-space data-consistency objective with a rate constraint. The PSNR results are reported in Table 13, showing that the proposed Lottery Prior can extend to medical imaging inverse problems beyond natural image settings. Our method achieves consistent improvement over DIP ($+0.3$ dB at $4\times$, $+1.26$ dB at $8\times$, demonstrating that the compression-based prior generalizes to MRI reconstruction. We note that both dataset-free approaches (DIP and ours) naturally have a performance ceiling compared to methods that exploit domain-specific training data or imaging physics priors. Bridging this gap through hybrid approaches (e.g., integrating Lottery Prior as a regularizer within physics-informed unrolled networks (Huang et al., 2023)) is a compelling direction for future work.

*Table 13.* Experiment over MRI reconstruction (Knee) from 4x and x8 undersampled k-space measurement.

| PSNR performance | x4 | x8 |
|---|---|---|
| **Deep image prior** | 26.15 | 23.92 |
| **Lottery prior** | **26.45** | **25.18** |

### D.4. Complexity analysis

We report the detailed runtime and computational complexity in Table 6 (first 3 images of Set11 for denoising with $\sigma = 15$, mask ratio 0.5 under the same configuration of Table 1).

**Latency and complexity.** To ensure fairness across different optimization schedules, we measure GPU runtime and complexity under a fixed budget of 1000 iterations. Compared to ZS-NCD, which trains a 0.4M-parameter network over overlapping patches (officially reported 40 minutes per image), our method is significantly more efficient due to its lightweight design and avoidance of extensive patch-wise redundancy. Compared to DIP, which relies on a heavy multi-scale architecture (2.2M parameters), we consider its lightweight variant for fairness (1–2 minutes). In contrast, our model is highly compact (0.7–3.3k parameters), leading to substantially lower per-iteration cost and better performance.

**Learning steps.** We further analyze performance across optimization steps (reported in Table 14) and observe that satisfactory results are achieved within 5–10k iterations, while longer runs provide additional but diminishing gains, indicating a favorable time–performance trade-off that can be flexibly adjusted based on the available computational budget.

*Table 14.* Denoising performance across various zero-shot learning steps.

| Learning steps | 500 | 1000 | 3000 | 5000 | 7000 | 10000 | 20000 | 30000 | 50000 |
|---|---|---|---|---|---|---|---|---|---|
| **PSNR** | 31.32 | 31.75 | 32.19 | 32.34 | 32.39 | 32.42 | 32.49 | 32.55 | 32.60 |
| **SSIM** | 0.8649 | 0.8710 | 0.8766 | 0.8793 | 0.8798 | 0.8802 | 0.8817 | 0.8825 | 0.8834 |

**Prior aggregation.** Our framework also supports flexible prior aggregation: combining a few randomized priors (e.g., 5 mask ratios, $\sim 10$–20 minutes) improves denoising, while a single prior (5–10k steps, $\sim 2$ minutes) suffices for structured tasks such as inpainting, avoiding over-smoothing, and achieving state-of-the-art performance. We note that the ensemble-based refinement introduces additional computational overhead. However, this component is optional and not required by the core method. In practice, a single run already achieves competitive (and for some tasks, such as inpainting, state-of-the-art) performance.

To summarize: in terms of total wall-clock time for a single image: ZS-NCD requires $\sim 40$ min (20K steps), DIP requires $\sim 1 - 2$ min (2-4K steps), and a single Lottery Prior requires $\sim 1 - 2$ min (5-10K steps at much lower per-step cost). For

ensemble refinement with 5 priors, total time increases to $\sim 5-10$ min, which remains competitive with ZS-NCD while achieving superior performance.

## E. Detailed datapoints

We report detailed results for each experiment. For baselines, we use the better result between our reproduction and the officially reported results from their original papers/codes, ensuring a fair comparison to prior methods.

**Denoising-Set11**   For denoising on Set11, results under AWGN are reported in Table 15 ($\lambda = 0.004/0.011$ for $\sigma = 15/25$), while results under Poisson noise are shown in Table 16 ($\lambda = 0.04/0.025$ for $\alpha = 15/25$).

**Denoising-Set13**   For denoising on Set13, results under AWGN are reported in Table 17 ($\lambda = 0.0015/0.004$ for $\sigma = 15/25$), while results under Poisson noise are shown in Table 18 ($\lambda = 0.013/0.008$ for $\alpha = 15/25$).

**Denoising-Kodak**   For denoising on Kodak, results under AWGN ($\lambda = 0.0015/0.0035$ for $\sigma = 15/25$) are reported in Table 19, while results under Poisson noise are shown in Table 20 ($\lambda = 0.011/0.0075$ for $\alpha = 15/25$).

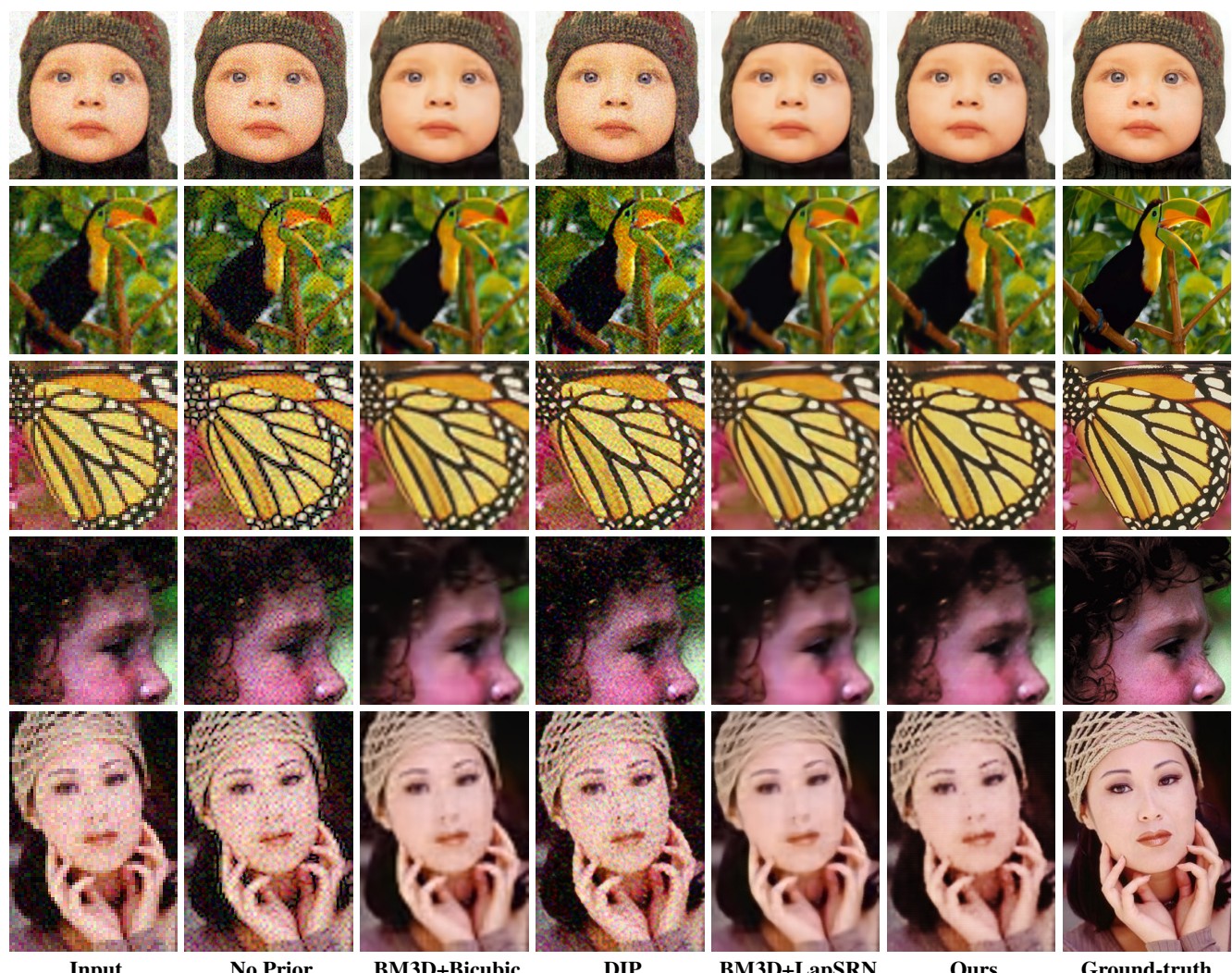

| Input | No Prior | BM3D+Bicubic | DIP | BM3D+LapSRN | Ours | Ground-truth |

*Figure 8.* $\times 4$ super-resolution results on Set5 under additive white Gaussian noise ($\sigma = 15$). Our method achieves high reconstruction quality and produces clean results comparable to the BM3D+LapSRN pipeline, which uses a supervised super-resolution network.

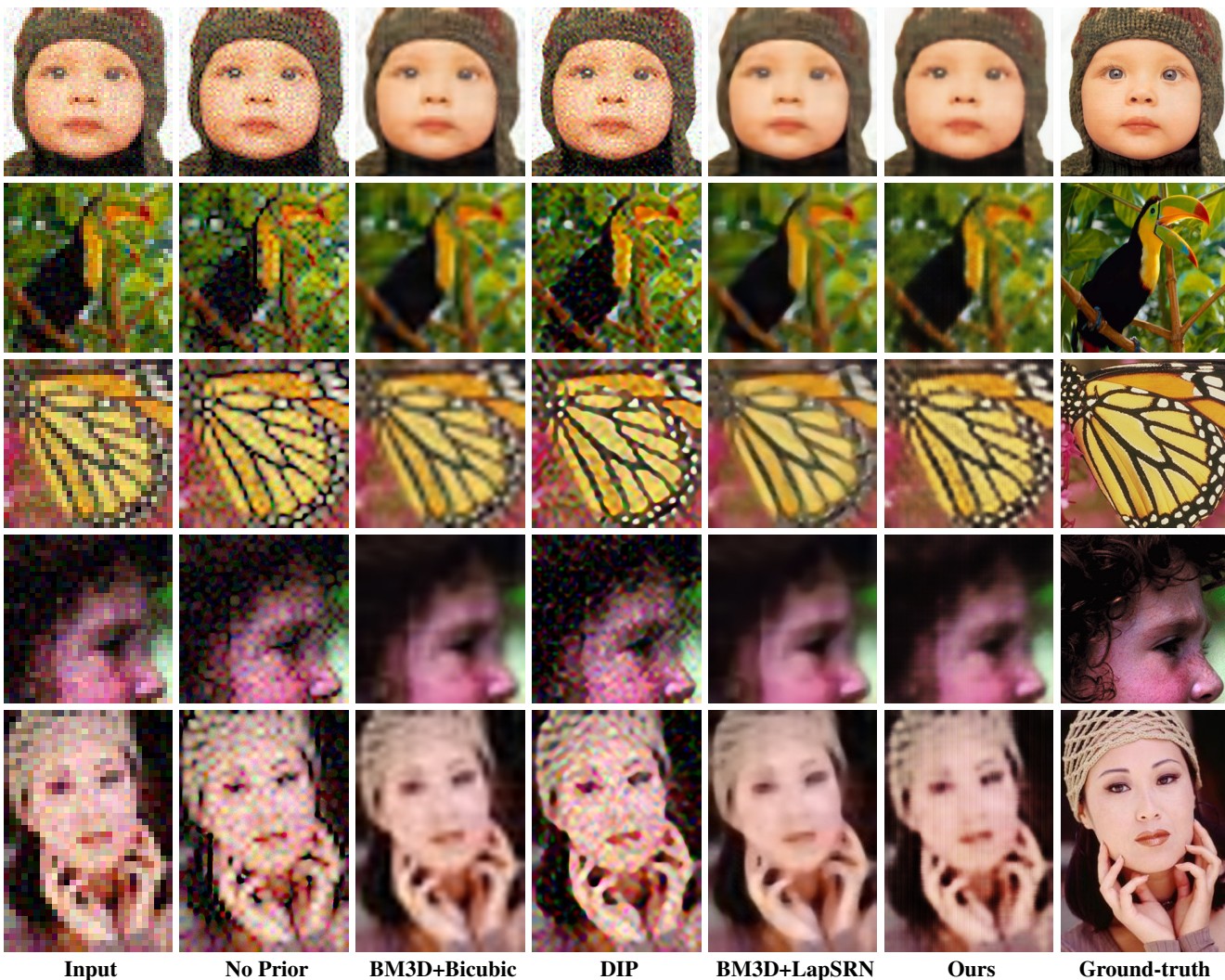

|Input|No Prior|BM3D+Bicubic|DIP|BM3D+LapSRN|Ours|Ground-truth|

*Figure 9.* ×8 super-resolution results on Set5 under additive white Gaussian noise ($\sigma = 15$). Our method achieves high reconstruction quality and produces clean results comparable to the BM3D+LapSRN pipeline, which uses a supervised super-resolution network.

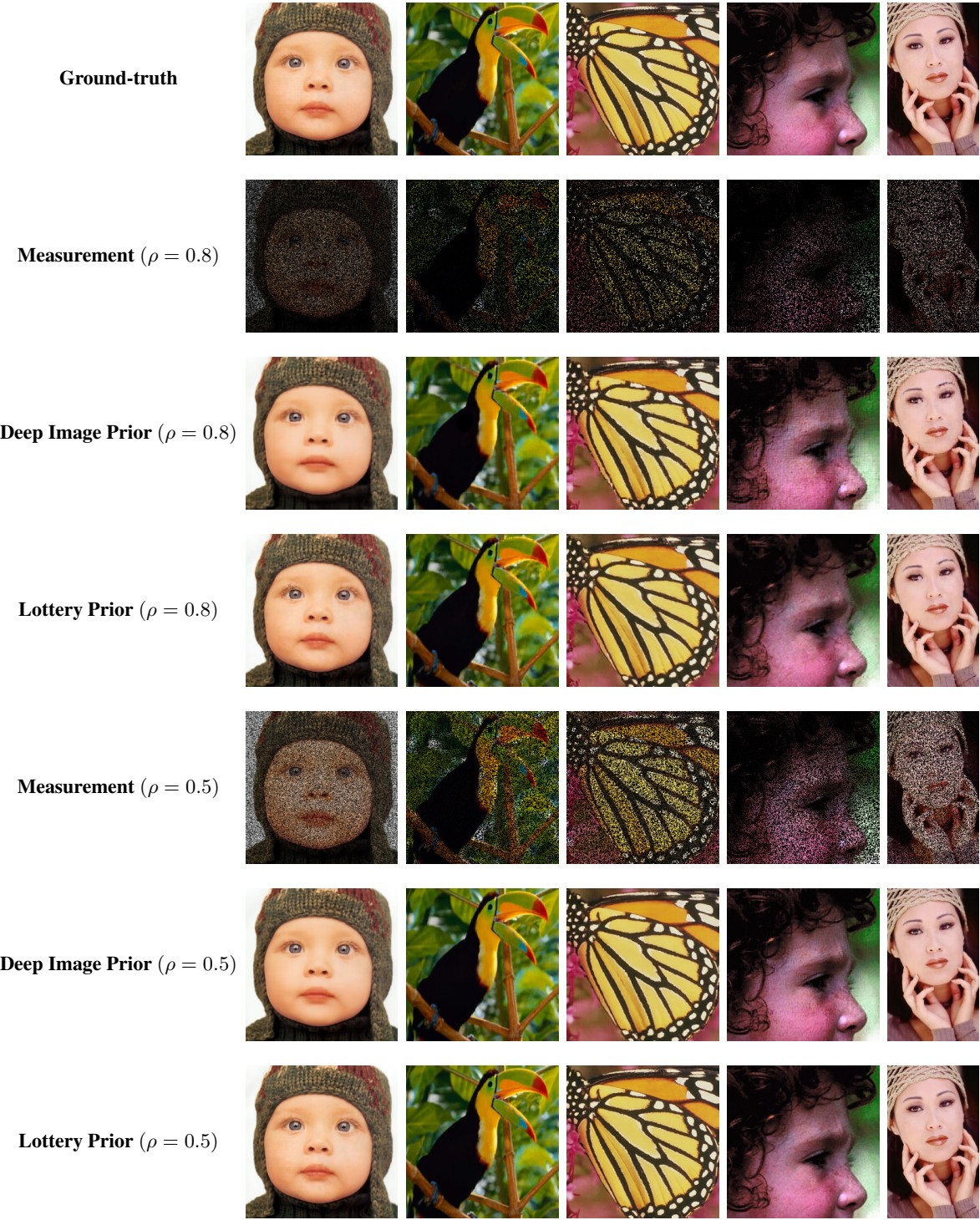

*Figure 10.* Inpainting results with different mask ratios ($\rho$) on Set5. Our *Lottery Prior* yields cleaner textures and fewer artifacts, especially under the heavier masking setting.

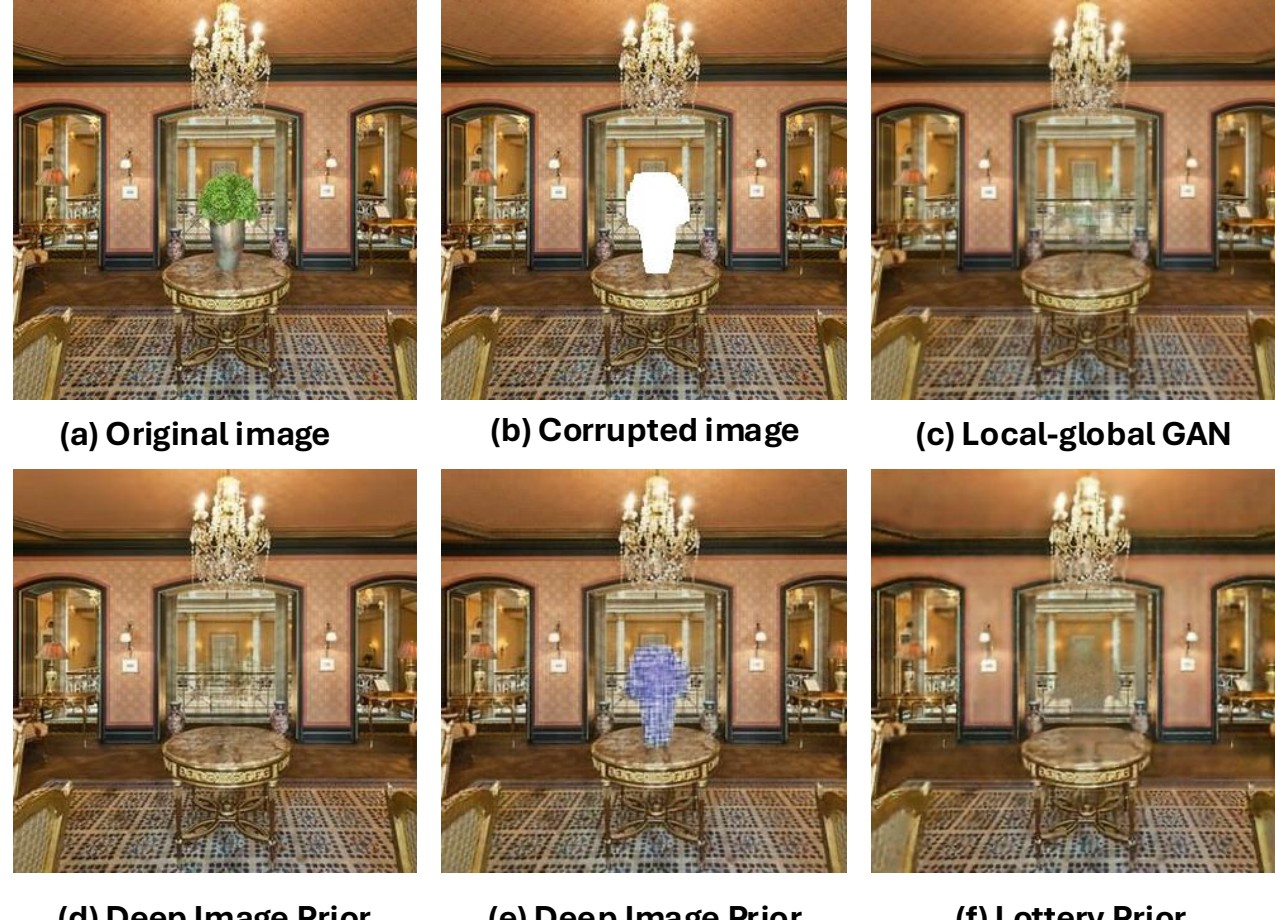

**(a) Original image**

**(b) Corrupted image**

**(c) Local-global GAN**

**(d) Deep Image Prior**

(good hyper-parameter example)

**(e) Deep Image Prior**

(bad hyper-parameter example)

**(f) Lottery Prior**

*Figure 11.* **Region inpainting: vase.** Deep Image Prior (DIP) can inpaint large missing regions despite the absence of external training, achieving results comparable to learned generative methods such as (Iizuka et al., 2017). However, its performance is sensitive to architectural choices and hyperparameters, as illustrated by the variation across (d) and (e). In contrast, *Lottery Prior*, while not designed for region-level semantic generation as in GAN-based methods, is still able to complete missing regions by exploiting contextual information from entropy modeling and the global fitting ability of an overfitted codec. Notably, this behavior is achieved with a substantially lighter architecture, suggesting that randomized masked networks can serve a role analogous to DIP as an implicit structural prior, with further gains possible by increasing model capacity.

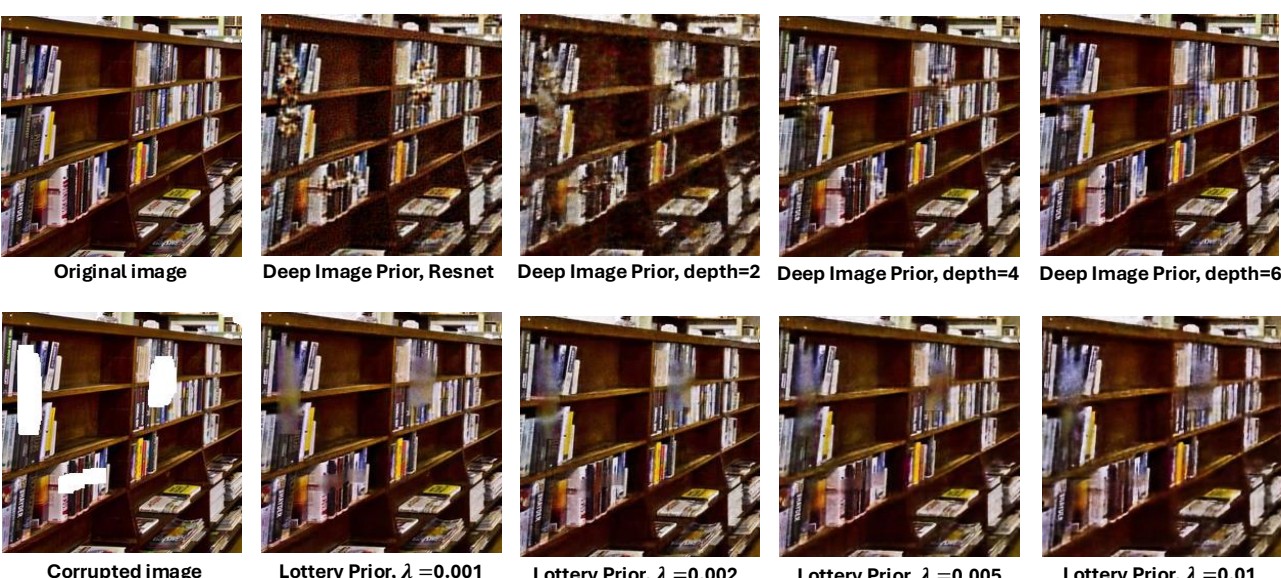

*Figure 12.* **Region inpainting: library.** The library scene poses a challenging inpainting task due to its complex structure. Deep Image Prior here requires very deep architectures and carefully tuned hyperparameters to achieve satisfactory results; otherwise, reconstructions degrade noticeably. In contrast, *Lottery Prior* controls reconstruction behavior via the rate constraint: larger $\lambda$ enforces stronger compression, improving consistency in the surrounding regions but suppressing background details. Interestingly, this behavior is similar to early stopping in DIP, complex scenes require earlier stopping, which leads to underfitting of background structures. From a compression perspective, this corresponds to operating at a higher compression rate, resulting in background smoothing. Motivated by DIP, increasing network capacity is expected to further improve inpainting quality for such complex scenes.

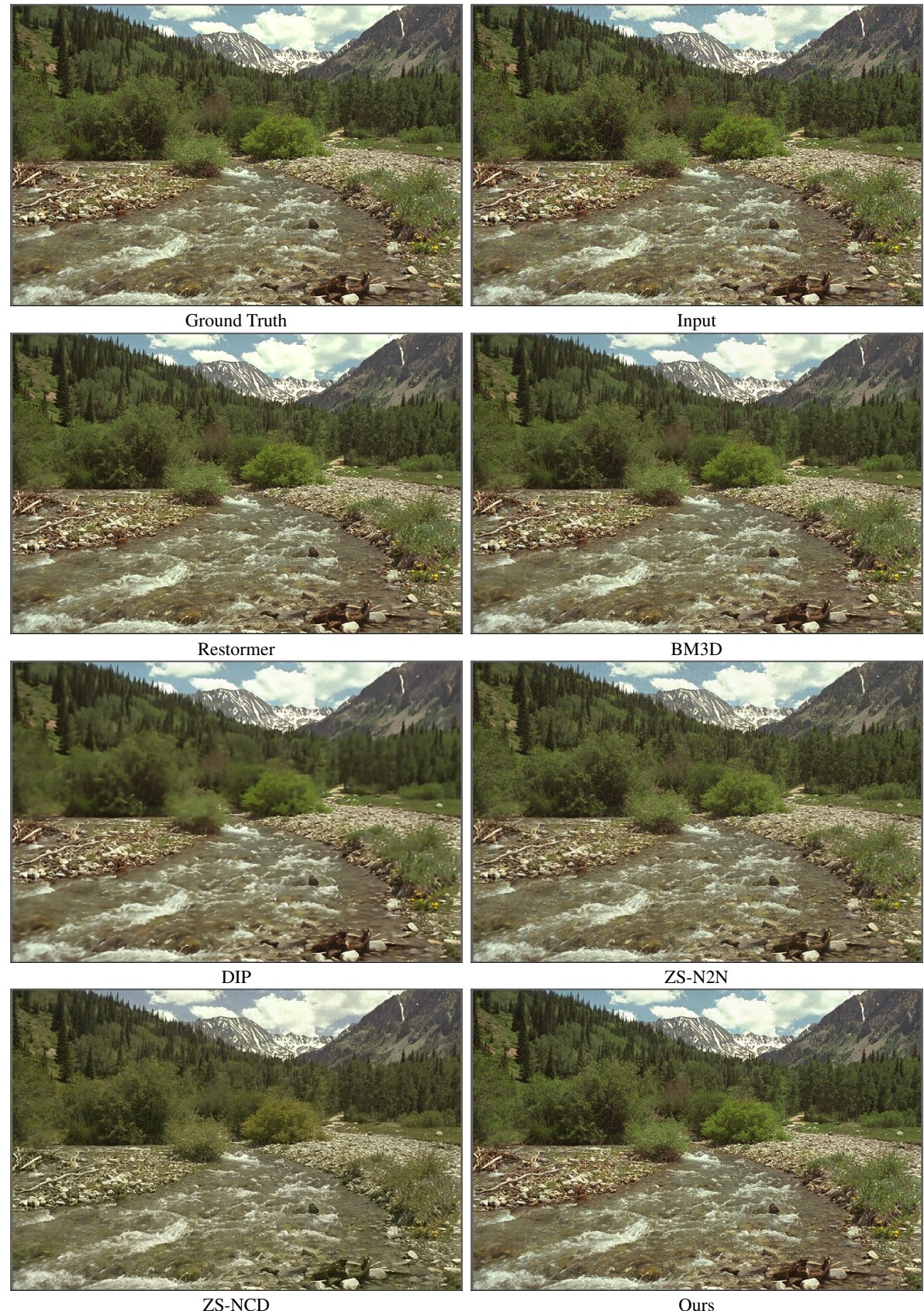

*Figure 13.* Denoising on Kodak under additive white Gaussian noise ($\sigma = 15$). Our approach achieves high fidelity and produces clean results comparable to the supervised Restormer approach.

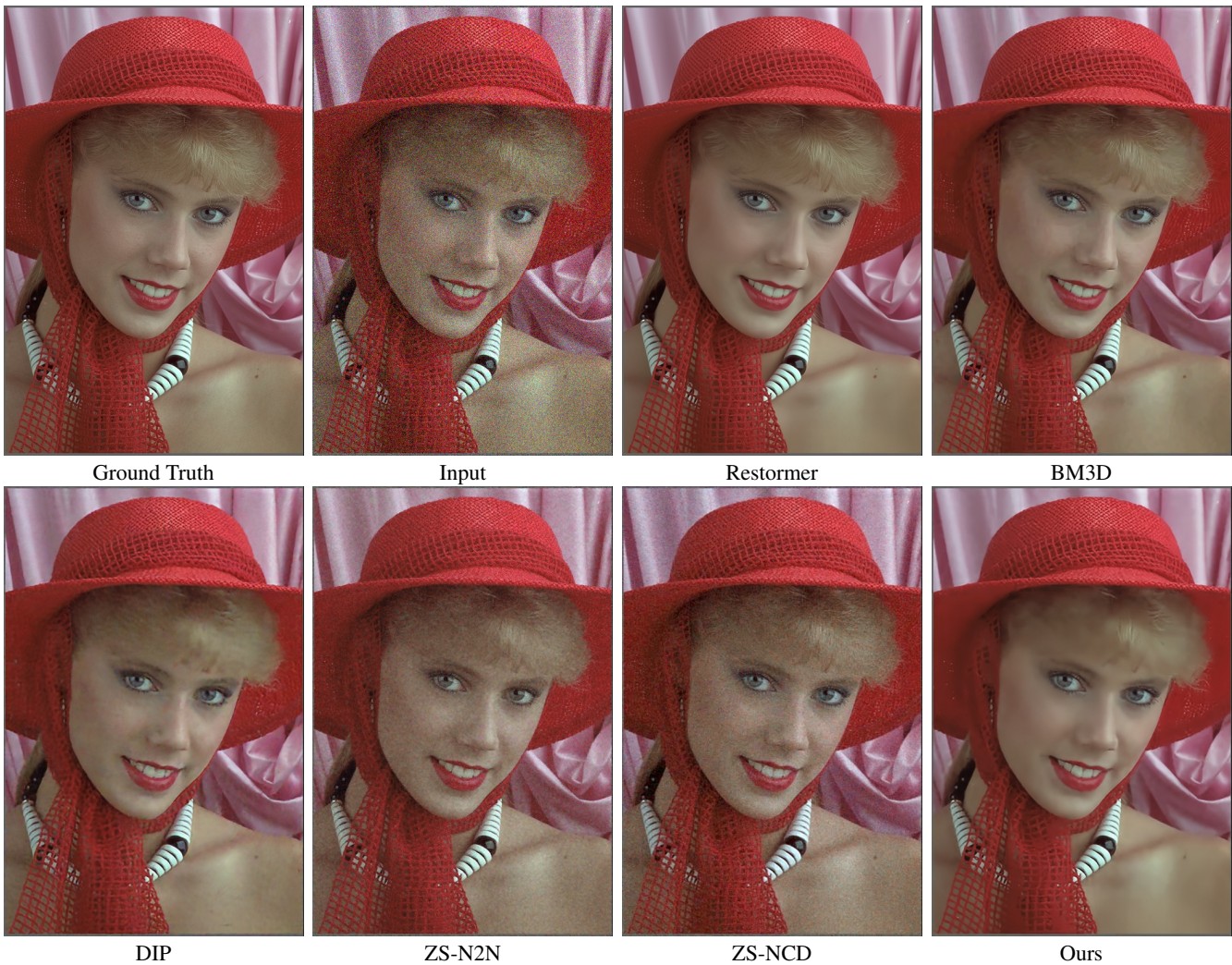

*Figure 14.* Denoising on Kodak under additive white Gaussian noise ($\sigma = 25$). Our approach achieves high fidelity and produces clean results comparable to the supervised Restormer approach.

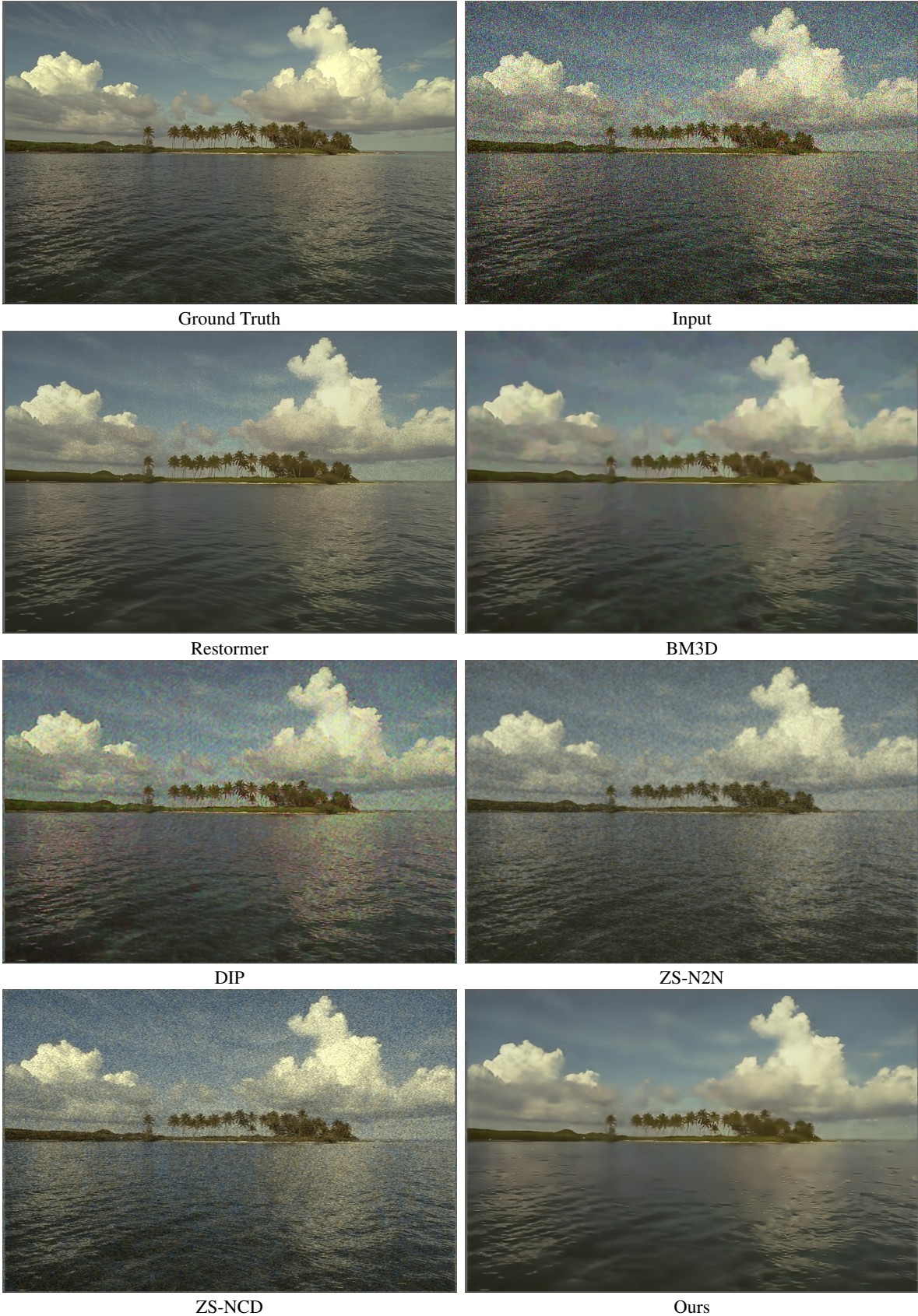

*Figure 15.* Denoising on Kodak under Poisson noise ($\alpha = 15$). Our approach achieves high fidelity and produces clean results comparable to the supervised Restormer approach.

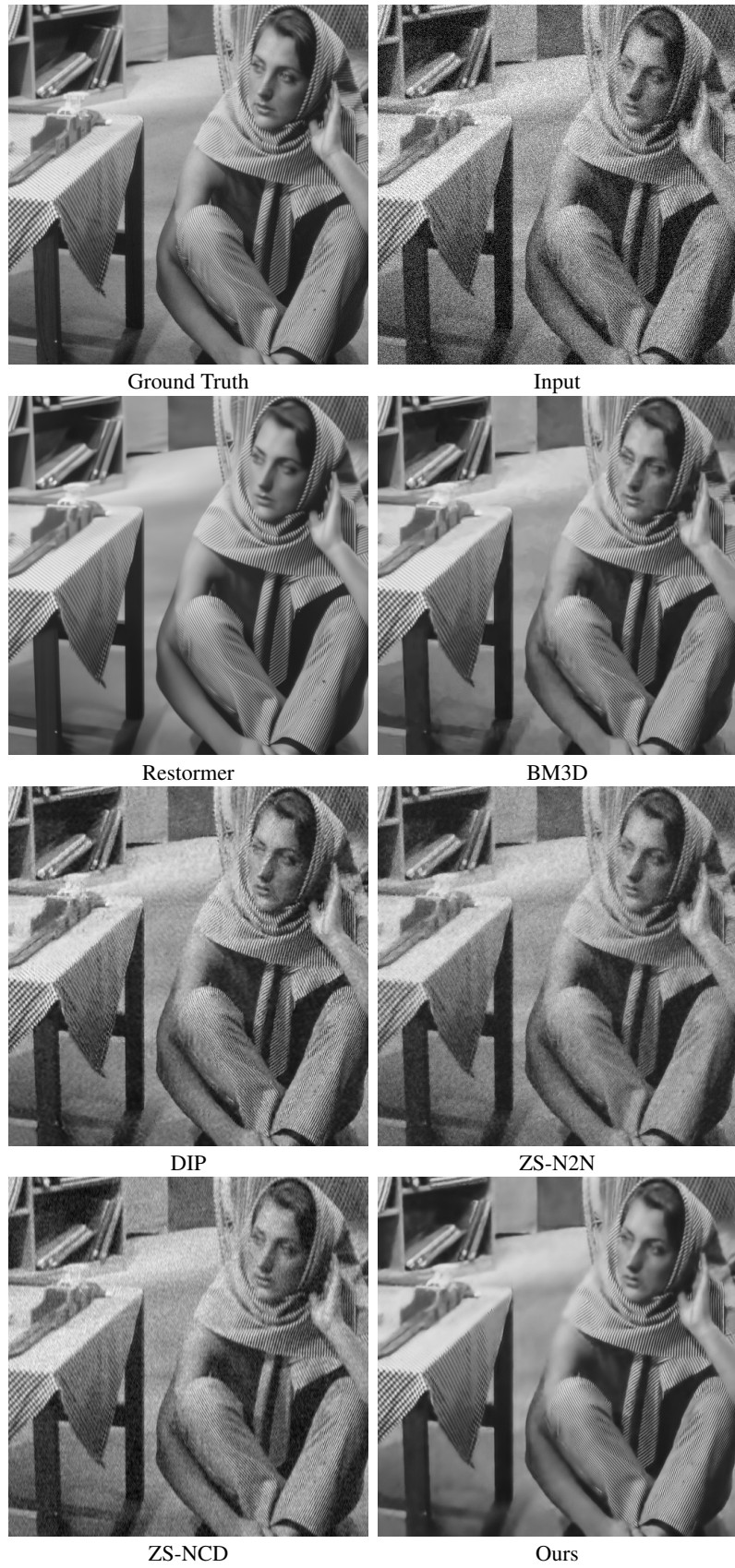

*Figure 16.* Denoising on Set11 under additive white Gaussian noise ($\sigma = 25$). Our approach achieves high fidelity and produces clean results comparable to the supervised Restormer approach.

*Table 15.* Set11 Denoising performance comparison under AWGN (PSNR and SSIM).

| σ | Method | 256 × 256 | | | | | | | 512 × 512 | | | | Average |
|---|--------|-----------|---|---|---|---|---|---|-----------|---|---|---|---------|
| | | C.man | House | Peppers | Starfish | Monarch | Airplane | Parrot | Barbara | Boats | Pirate | Couple | |
| | BM3D | 31.84/0.8974 | 34.94/0.8870 | 32.82/0.9118 | 31.16/0.9082 | 31.92/0.9409 | 31.09/0.9034 | 31.47/0.9032 | 33.04/0.9253 | 32.12/0.8604 | 31.94/0.8726 | 32.11/0.8795 | 32.22/0.8991 |
| | JPEG2K | 27.12/0.7474 | 29.48/0.7621 | 27.96/0.7907 | 26.75/0.8077 | 26.74/0.8166 | 26.58/0.7664 | 27.30/0.7778 | 26.76/0.7690 | 27.87/0.7390 | 27.92/0.7471 | 27.44/0.7449 | 27.45/0.7699 |
| | DIP | 27.94/0.7417 | 31.39/0.8111 | 29.80/0.8273 | 29.58/0.8605 | 29.93/0.8767 | 28.14/0.8047 | 28.37/0.7794 | 27.65/0.7538 | 29.48/0.7798 | 29.27/0.7817 | 28.65/0.7727 | 29.11/0.7990 |
| | DD | 29.41/0.8099 | 32.83/0.8406 | 26.97/0.8488 | 29.39/0.8739 | 30.01/0.8957 | 26.44/0.8228 | 29.32/0.8447 | 24.48/0.7089 | 29.45/0.7883 | 29.78/0.8085 | 29.06/0.7938 | 28.83/0.8215 |
| 15 | ZS-N2N | 30.14/0.8133 | 32.19/0.8138 | 30.58/0.8264 | 29.52/0.8639 | 30.15/0.8551 | 29.98/0.8298 | 30.19/0.8290 | 27.70/0.7772 | 30.06/0.7900 | 30.06/0.7957 | 29.59/0.7913 | 30.01/0.8169 |
| | ZS-N2S | 27.66/0.8272 | 31.08/0.8442 | 29.46/0.8675 | 28.83/0.8810 | 28.77/0.8961 | 27.34/0.8591 | 27.67/0.8528 | 28.75/0.8534 | 29.52/0.8139 | 29.41/0.8181 | 29.60/0.8311 | 28.92/0.8495 |
| | S2S | 20.29/0.6769 | 32.96/0.8633 | 23.96/0.8387 | 25.50/0.8250 | 30.05/0.9269 | 28.10/0.8611 | 20.20/0.7132 | 30.35/0.8865 | 27.74/0.7871 | 29.97/0.8192 | 25.82/0.7754 | 26.81/0.8158 |
| | ZS-NCD | 30.83/0.8554 | 34.45/0.8835 | 32.20/0.8844 | 31.34/0.8749 | 31.83/0.8966 | 30.07/0.8552 | 30.40/0.8464 | 31.14/0.8826 | 31.09/0.8014 | 30.82/0.8302 | 30.67/0.8279 | 31.35/0.8580 |
| | **Lottery Prior** | 31.82/0.8949 | 34.55/0.8779 | 32.90/0.9073 | 31.92/0.9074 | 32.73/0.9448 | 31.35/0.9003 | 31.48/0.8931 | 32.94/0.9183 | 32.11/0.8519 | 31.77/0.8582 | 31.71/0.8603 | 32.30/0.8922 |
| | BM3D | 29.54/0.8499 | 32.79/0.8561 | 30.13/0.8705 | 28.58/0.8578 | 29.36/0.9046 | 28.50/0.8584 | 28.94/0.8561 | 30.63/0.8887 | 29.87/0.8039 | 29.64/0.8082 | 29.74/0.8206 | 29.79/0.8523 |
| | JPEG2K | 24.49/0.6976 | 27.26/0.7269 | 24.93/0.7206 | 24.18/0.7167 | 24.06/0.7561 | 23.91/0.7126 | 24.38/0.7162 | 24.09/0.6825 | 25.61/0.6577 | 25.76/0.6566 | 25.28/0.6534 | 24.91/0.6997 |
| | DIP | 25.23/0.6043 | 28.93/0.7545 | 27.39/0.7579 | 26.39/0.7777 | 27.47/0.8169 | 25.57/0.6983 | 26.29/0.7409 | 24.75/0.6356 | 27.05/0.6843 | 27.06/0.6857 | 26.52/0.6847 | 26.60/0.7128 |
| | DD | 27.24/0.7521 | 30.48/0.8023 | 25.39/0.7591 | 26.86/0.8051 | 27.69/0.8526 | 24.93/0.7120 | 27.29/0.7863 | 23.81/0.6455 | 27.49/0.7163 | 27.87/0.7380 | 27.13/0.7131 | 26.93/0.7530 |
| 25 | ZS-N2N | 27.32/0.7089 | 29.36/0.7276 | 27.46/0.7240 | 26.61/0.7821 | 27.20/0.7634 | 27.02/0.7463 | 27.16/0.7149 | 25.49/0.6854 | 27.26/0.6779 | 27.48/0.6931 | 26.63/0.6673 | 27.18/0.7173 |
| | ZS-N2S | 26.24/0.7843 | 29.23/0.8073 | 27.77/0.8233 | 27.61/0.8463 | 27.35/0.8569 | 25.86/0.8023 | 26.27/0.7997 | 26.43/0.7759 | 28.23/0.7580 | 27.52/0.7526 | 27.74/0.7617 | 27.30/0.7971 |
| | S2S | 16.93/0.5998 | 29.12/0.8275 | 21.88/0.7666 | 21.14/0.6974 | 25.93/0.8606 | 24.12/0.7350 | 17.09/0.6069 | 25.79/0.7980 | 23.94/0.7061 | 23.94/0.7403 | 23.29/0.6979 | 23.32/0.7306 |
| | ZS-NCD | 28.78/0.8237 | 32.14/0.8547 | 29.62/0.8406 | 28.48/0.8134 | 29.02/0.8494 | 27.77/0.8126 | 28.14/0.8007 | 28.39/0.8192 | 28.85/0.7444 | 28.64/0.7630 | 28.37/0.7648 | 28.93/0.8079 |
| | **Lottery Prior** | 29.11/0.8360 | 32.53/0.8513 | 30.34/0.8681 | 29.24/0.8626 | 29.79/0.9054 | 28.64/0.8556 | 28.72/0.8337 | 30.20/0.8695 | 29.73/0.7881 | 29.50/0.7943 | 29.27/0.7976 | 29.73/0.8420 |

*Table 16.* Set11 denoising performance comparison under Poisson noise (PSNR and SSIM).

| α | Method | 256 × 256 | | | | | | | 512 × 512 | | | | Average |
|---|--------|-----------|---|---|---|---|---|---|-----------|---|---|---|---------|
| | | C.man | House | Peppers | Starfish | Monarch | Airplane | Parrot | Barbara | Boats | Pirate | Couple | |
| | BM3D | 26.64/0.7651 | 29.39/0.7668 | 27.13/0.7914 | 24.93/0.7519 | 26.32/0.8265 | 24.79/0.6730 | 26.26/0.7866 | 27.24/0.7860 | 26.82/0.6977 | 27.07/0.7048 | 26.67/0.7056 | 26.66/0.7505 |
| | JPEG2K | 21.98/0.6032 | 24.35/0.6106 | 22.12/0.6213 | 21.52/0.5887 | 21.24/0.6493 | 20.87/0.5818 | 22.02/0.6378 | 21.94/0.5688 | 23.09/0.5346 | 23.75/0.5510 | 23.01/0.5237 | 22.35/0.5882 |
| | DIP | 22.85/0.5382 | 26.32/0.6528 | 24.23/0.6138 | 23.23/0.6696 | 23.54/0.6875 | 22.07/0.4938 | 22.81/0.5723 | 22.59/0.5503 | 24.18/0.5533 | 24.95/0.5811 | 23.83/0.5362 | 23.69/0.5863 |
| | DD | 24.45/0.6261 | 27.59/0.7453 | 23.20/0.7269 | 23.86/0.7164 | 24.66/0.7286 | 22.22/0.6055 | 24.38/0.6830 | 22.89/0.6081 | 24.64/0.6023 | 25.61/0.6516 | 24.58/0.5986 | 24.37/0.6629 |
| 15 | ZS-N2N | 24.19/0.5818 | 25.41/0.5346 | 24.65/0.6016 | 23.12/0.6520 | 23.92/0.6441 | 23.12/0.5565 | 23.83/0.5821 | 23.05/0.5503 | 24.40/0.5403 | 24.87/0.5684 | 23.94/0.5305 | 24.04/0.5766 |
| | ZS-N2S | 24.94/0.7241 | 27.29/0.7317 | 25.71/0.7431 | 24.41/0.7417 | 25.37/0.7968 | 23.05/0.7051 | 24.98/0.7315 | 22.87/0.6087 | 26.08/0.6696 | 25.94/0.6655 | 25.09/0.6389 | 25.06/0.7051 |
| | S2S | 23.53/0.7325 | 22.01/0.7409 | 22.73/0.7300 | 18.20/0.6010 | 21.81/0.7813 | 16.18/0.5010 | 22.70/0.7304 | 22.10/0.7261 | 23.49/0.6529 | 24.72/0.6782 | 24.17/0.6843 | 21.75/0.6872 |
| | **ZS-NCD** | 25.73/0.7660 | 28.87/0.8015 | 26.54/0.7745 | 24.65/0.6988 | 25.86/0.7791 | 24.21/0.6568 | 25.52/0.7356 | 24.11/0.6562 | 25.48/0.6510 | 25.93/0.6705 | 25.29/0.6552 | 25.65/0.7132 |
| | **Lottery Prior** | 25.54/0.6940 | 29.19/0.7886 | 26.92/0.7759 | 25.02/0.7490 | 26.11/0.8070 | 25.05/0.7437 | 25.96/0.7677 | 25.81/0.7396 | 25.89/0.6419 | 26.56/0.6820 | 25.76/0.6702 | 26.16/0.7327 |
| | BM3D | 22.69/0.5154 | 22.82/0.4765 | 22.74/0.5930 | 22.11/0.6947 | 23.44/0.7213 | 19.60/0.3788 | 23.05/0.5991 | 23.09/0.6508 | 22.77/0.5123 | 23.89/0.5979 | 23.45/0.5753 | 22.70/0.5741 |
| | JPEG2K | 22.54/0.6267 | 24.97/0.6773 | 22.87/0.6076 | 22.26/0.6378 | 22.55/0.6641 | 21.59/0.5649 | 22.62/0.6373 | 22.55/0.5976 | 23.71/0.5685 | 24.20/0.5801 | 23.49/0.5566 | 23.03/0.6108 |
| | DIP | 24.21/0.5976 | 27.06/0.6553 | 25.76/0.6945 | 24.41/0.7312 | 25.21/0.7384 | 23.58/0.6290 | 24.69/0.6608 | 23.11/0.5903 | 25.24/0.6130 | 26.10/0.6528 | 24.91/0.5998 | 24.94/0.6512 |
| | DD | 25.59/0.6695 | 28.47/0.7606 | 24.20/0.7348 | 25.14/0.7667 | 26.16/0.8022 | 23.24/0.6306 | 25.89/0.7289 | 23.27/0.6257 | 25.84/0.6517 | 26.76/0.6961 | 25.77/0.6573 | 25.48/0.7022 |
| 25 | ZS-N2N | 25.54/0.6334 | 27.14/0.6234 | 25.82/0.6522 | 24.33/0.7158 | 25.51/0.7109 | 24.53/0.6274 | 25.55/0.6617 | 24.09/0.6173 | 25.60/0.6018 | 26.11/0.6354 | 25.16/0.5958 | 25.40/0.6432 |
| | ZS-N2S | 26.22/0.7776 | 27.81/0.7643 | 26.55/0.7768 | 24.82/0.7795 | 26.48/0.8254 | 24.77/0.7463 | 25.44/0.7839 | 23.24/0.6387 | 27.25/0.7143 | 27.13/0.7200 | 26.44/0.6986 | 26.01/0.7478 |
| | S2S | 25.09/0.7572 | 24.10/0.7398 | 24.91/0.7733 | 19.11/0.6491 | 23.64/0.8226 | 17.93/0.6279 | 21.13/0.7692 | 24.01/0.7860 | 25.30/0.7058 | 26.45/0.7232 | 25.77/0.7360 | 23.40/0.7355 |
| | **ZS-NCD** | 27.17/0.7635 | 30.09/0.8109 | 27.92/0.7932 | 26.27/0.7600 | 27.28/0.8093 | 24.93/0.6116 | 26.74/0.7551 | 26.24/0.7393 | 27.30/0.6993 | 27.32/0.7192 | 26.83/0.7123 | 27.10/0.7431 |
| | **Lottery Prior** | 27.77/0.8004 | 30.74/0.8219 | 28.29/0.8185 | 26.84/0.8042 | 27.81/0.8596 | 25.07/0.6364 | 27.05/0.7950 | 27.58/0.7914 | 27.78/0.7236 | 27.68/0.7259 | 27.45/0.7309 | 27.64/0.7734 |

*Table 17.* Set13 denoising performance comparison under AWGN (PSNR and SSIM).

| σ | Method | Baboon | Barbara | Bridge | Coastguard | Comic | Face | Flowers | Foreman | Man | Monarch | Peppers | PPT3 | Zebra | Average |
|---|---|---|---|---|---|---|---|---|---|---|---|---|---|---|---|
| | BM3D | 28.56/0.7797 | 33.07/0.9151 | 30.39/0.8723 | 30.18/0.8799 | 28.74/0.9289 | 30.28/0.7665 | 29.62/0.9040 | 35.83/0.9369 | 29.88/0.8323 | 31.13/0.9361 | 31.77/0.8199 | 34.49/0.9588 | 31.02/0.9198 | 31.15/0.8808 |
| | JPEG2K | 25.12/0.6539 | 27.01/0.7674 | 26.27/0.7426 | 25.79/0.7323 | 24.86/0.8017 | 27.35/0.6482 | 25.52/0.7986 | 30.86/0.8397 | 25.97/0.6983 | 25.63/0.7649 | 28.27/0.7161 | 28.00/0.8183 | 26.33/0.8243 | 26.69/0.7543 |
| | DIP | 27.25/0.7498 | 30.92/0.8403 | 30.79/0.9036 | 31.18/0.8692 | 30.79/0.9036 | 28.25/0.9091 | 29.62/0.7611 | 28.85/0.8817 | 33.18/0.8947 | 29.99/0.8333 | 30.30/0.7782 | 32.22/0.8908 | 30.63/0.9203 | 30.31/0.8570 |
| | DD | 26.35/0.7029 | 24.27/0.7066 | 29.16/0.8508 | 28.80/0.8421 | 26.44/0.8932 | 29.59/0.7398 | 27.34/0.8634 | 34.87/0.9274 | 28.68/0.8073 | 30.02/0.9151 | 31.07/0.8003 | 33.09/0.9277 | 30.22/0.9052 | 29.22/0.8371 |
| 15 | ZS-N2N | 28.63/0.7992 | 28.45/0.7803 | 32.08/0.9041 | 31.54/0.9141 | 28.70/0.9018 | 30.64/0.8048 | 29.67/0.8955 | 34.02/0.8817 | 31.63/0.8801 | 31.65/0.9120 | 31.28/0.8089 | 32.84/0.9009 | 31.27/0.9277 | 30.95/0.8701 |
| | ZS-N2S | 20.92/0.5844 | 21.14/0.5730 | 21.37/0.6468 | 20.78/0.4893 | 15.80/0.4987 | 22.03/0.5641 | 17.49/0.5154 | 8.43/0.3637 | 22.69/0.6725 | 12.21/0.6124 | 21.60/0.6438 | 11.58/0.4462 | 20.37/0.7871 | 18.18/0.5690 |
| | S2S | 22.36/0.5810 | 30.39/0.8769 | 22.74/0.7485 | 22.72/0.7108 | 17.44/0.7015 | 17.23/0.4383 | 21.14/0.7121 | 16.78/0.8102 | 15.65/0.4463 | 26.92/0.8838 | 24.20/0.7398 | 15.48/0.7528 | 14.84/0.5400 | 20.61/0.6879 |
| | ZS-NCD | 28.10/0.7831 | 33.85/0.9208 | 31.49/0.9051 | 32.65/0.9345 | 29.23/0.9355 | 30.51/0.7891 | 29.69/0.9077 | 35.85/0.9381 | 31.49/0.8867 | 33.21/0.9445 | 31.60/0.8281 | 35.07/0.9601 | 32.37/0.9448 | 31.93/0.8983 |
| | **Lottery Prior** | 29.36/0.8022 | 34.57/0.9234 | 32.91/0.9175 | 32.81/0.9253 | 31.21/0.9526 | 30.44/0.7456 | 31.66/0.9288 | 36.39/0.9479 | 32.26/0.8649 | 34.53/0.9589 | 32.07/0.8054 | 35.42/0.9707 | 32.02/0.9087 | 32.74/0.8963 |
| | BM3D | 26.56/0.6892 | 30.70/0.8824 | 28.06/0.7955 | 27.59/0.7664 | 25.90/0.8698 | 28.87/0.6978 | 26.93/0.8396 | 33.51/0.9087 | 27.35/0.7428 | 28.72/0.9056 | 30.11/0.7823 | 31.66/0.9324 | 28.61/0.8643 | 28.81/0.8213 |
| | JPEG2K | 23.98/0.5711 | 24.14/0.6801 | 24.14/0.6400 | 23.53/0.5801 | 21.36/0.7014 | 26.17/0.5692 | 22.77/0.6809 | 28.30/0.7960 | 23.51/0.5667 | 23.03/0.7101 | 26.47/0.6729 | 25.23/0.7707 | 23.58/0.7389 | 24.32/0.6676 |
| | DIP | 25.70/0.6734 | 27.27/0.7021 | 27.79/0.7999 | 27.86/0.8126 | 25.11/0.8382 | 28.16/0.6599 | 26.03/0.8164 | 31.17/0.8384 | 27.27/0.7446 | 29.02/0.8785 | 28.82/0.7313 | 29.20/0.8106 | 28.65/0.8820 | 27.85/0.7837 |
| | DD | 25.56/0.6589 | 23.56/0.6676 | 26.94/0.7758 | 26.77/0.7542 | 24.76/0.8397 | 28.37/0.6819 | 25.72/0.8058 | 32.22/0.8849 | 27.09/0.7366 | 27.68/0.8566 | 29.42/0.7615 | 29.44/0.8937 | 28.11/0.8641 | 27.40/0.7832 |
| 25 | ZS-N2N | 26.85/0.7287 | 26.73/0.7136 | 29.00/0.8282 | 28.30/0.8312 | 25.95/0.8462 | 28.71/0.7250 | 26.67/0.8248 | 31.31/0.8156 | 28.88/0.7940 | 29.03/0.8683 | 28.95/0.7282 | 29.78/0.8205 | 28.54/0.8768 | 28.36/0.8001 |
| | ZS-N2S | 19.22/0.4963 | 15.44/0.3973 | 22.05/0.5859 | 21.55/0.5581 | 16.64/0.5734 | 25.76/0.6423 | 17.98/0.6324 | 20.90/0.6792 | 22.30/0.6170 | 21.29/0.7778 | 22.45/0.6535 | 17.04/0.6502 | 22.41/0.7964 | 20.39/0.6200 |
| | S2S | 18.66/0.4947 | 24.97/0.7969 | 19.43/0.6054 | 20.69/0.5890 | 16.30/0.6168 | 15.08/0.3846 | 17.57/0.5267 | 15.53/0.7605 | 14.11/0.3892 | 21.34/0.7849 | 22.40/0.6795 | 13.60/0.6805 | 13.69/0.4890 | 17.95/0.5998 |
| | ZS-NCD | 26.54/0.7128 | 30.65/0.8388 | 28.78/0.8332 | 29.46/0.8718 | 26.83/0.8847 | 29.14/0.7367 | 27.05/0.8393 | 32.99/0.8870 | 28.83/0.8027 | 30.18/0.8917 | 29.62/0.7656 | 31.56/0.8898 | 29.63/0.9023 | 29.33/0.8351 |
| | **Lottery Prior** | 27.00/0.6891 | 32.60/0.9057 | 29.77/0.8381 | 29.66/0.8461 | 28.00/0.9080 | 28.82/0.6605 | 28.77/0.8765 | 33.04/0.9184 | 28.86/0.7658 | 31.54/0.9399 | 30.52/0.7776 | 32.08/0.9579 | 29.21/0.8542 | 30.00/0.8414 |

*Table 18.* Set13 denoising performance comparison under Poisson noise (PSNR and SSIM).

| α | Method | Baboon | Barbara | Bridge | Coastguard | Comic | Face | Flowers | Foreman | Man | Monarch | Peppers | PPT3 | Zebra | Average |
|---|---|---|---|---|---|---|---|---|---|---|---|---|---|---|---|
| | BM3D | 24.46/0.5651 | 26.63/0.7712 | 25.18/0.6462 | 24.45/0.5186 | 22.07/0.7101 | 27.46/0.6342 | 25.01/0.7017 | 29.05/0.7747 | 24.98/0.6405 | 25.60/0.8101 | 27.53/0.7041 | 26.16/0.7288 | 26.19/0.7802 | 25.64/0.6912 |
| | JPEG2K | 22.01/0.4873 | 21.82/0.5515 | 21.77/0.5163 | 21.69/0.3727 | 18.47/0.5237 | 24.97/0.5289 | 20.42/0.5841 | 23.83/0.6996 | 21.69/0.4723 | 20.21/0.5809 | 24.17/0.6012 | 20.73/0.5858 | 21.04/0.6381 | 21.76/0.5494 |
| | DIP | 24.30/0.5907 | 23.05/0.6010 | 25.46/0.7069 | 24.21/0.5996 | 21.79/0.7282 | 26.99/0.6150 | 23.41/0.7456 | 28.03/0.7686 | 25.10/0.6444 | 26.34/0.8136 | 26.26/0.6510 | 25.44/0.7088 | 26.37/0.8180 | 25.14/0.6916 |
| | DD | 23.89/0.5750 | 23.16/0.6479 | 24.83/0.6873 | 23.80/0.5544 | 21.52/0.7107 | 27.22/0.6387 | 23.67/0.7339 | 29.25/0.8522 | 25.03/0.6514 | 24.67/0.7745 | 27.09/0.7156 | 25.12/0.7640 | 25.43/0.8020 | 24.96/0.7006 |
| 15 | ZS-N2N | 24.82/0.6330 | 24.11/0.5768 | 25.82/0.7281 | 25.35/0.6946 | 21.99/0.7055 | 27.11/0.6779 | 23.67/0.7490 | 27.20/0.6491 | 26.60/0.7290 | 25.99/0.7740 | 26.01/0.6047 | 25.02/0.6004 | 26.13/0.8190 | 25.37/0.6878 |
| | ZS-N2S | 21.39/0.5390 | 17.46/0.4084 | 22.23/0.6428 | 21.83/0.5714 | 17.53/0.5771 | 25.14/0.6103 | 17.97/0.5082 | 24.33/0.7854 | 22.94/0.5959 | 21.15/0.7157 | 21.53/0.5467 | 19.43/0.6122 | 23.08/0.7725 | 21.23/0.6066 |
| | S2S | 16.58/0.5042 | 21.66/0.6523 | 18.07/0.6269 | 22.14/0.6206 | 15.01/0.5319 | 24.18/0.6548 | 18.87/0.6736 | 14.80/0.7562 | 24.78/0.7188 | 17.73/0.7807 | 21.51/0.6626 | 12.61/0.5630 | 22.09/0.7728 | 19.23/0.6553 |
| | ZS-NCD | 24.46/0.5807 | 27.63/0.7844 | 26.43/0.7443 | 26.40/0.7436 | 23.17/0.7781 | 27.53/0.6510 | 24.30/0.7680 | 29.58/0.8302 | 26.15/0.6997 | 26.46/0.7991 | 27.34/0.7196 | 26.68/0.7217 | 27.45/0.8440 | 26.44/0.7434 |
| | **Lottery Prior** | 24.49/0.6127 | 28.49/0.8330 | 26.04/0.7391 | 26.08/0.6449 | 23.09/0.7903 | 27.99/0.6595 | 24.93/0.7745 | 26.66/0.8711 | 26.81/0.7064 | 27.14/0.8942 | 28.57/0.7543 | 24.71/0.9084 | 27.57/0.8244 | 26.35/0.7702 |
| | BM3D | 20.61/0.4799 | 21.75/0.5241 | 22.93/0.6748 | 22.04/0.5620 | 19.94/0.6637 | 24.88/0.6358 | 22.35/0.7387 | 22.34/0.5167 | 23.11/0.5790 | 21.99/0.6799 | 22.75/0.5533 | 20.01/0.4239 | 23.48/0.7582 | 22.17/0.5992 |
| | JPEG2K | 22.59/0.5060 | 22.38/0.5732 | 22.81/0.5904 | 22.12/0.4455 | 19.59/0.6087 | 25.30/0.5533 | 21.51/0.6444 | 25.19/0.7343 | 22.10/0.4841 | 21.52/0.6226 | 24.78/0.6181 | 22.07/0.6515 | 22.44/0.7057 | 22.65/0.5952 |
| | DIP | 24.85/0.6243 | 24.02/0.5860 | 26.63/0.7646 | 25.41/0.6907 | 22.75/0.7640 | 27.42/0.6386 | 24.61/0.7935 | 29.22/0.8138 | 26.07/0.7119 | 27.02/0.8225 | 27.36/0.6839 | 26.86/0.7348 | 27.25/0.8467 | 26.13/0.7289 |
| | DD | 24.51/0.6017 | 23.30/0.6625 | 25.78/0.7389 | 24.96/0.6226 | 22.80/0.7680 | 28.18/0.6808 | 24.65/0.7872 | 30.34/0.8723 | 25.91/0.6824 | 26.31/0.8306 | 28.07/0.7242 | 26.92/0.7633 | 26.80/0.8309 | 26.04/0.7373 |
| 25 | ZS-N2N | 25.81/0.6876 | 25.11/0.6387 | 27.17/0.7874 | 26.65/0.7562 | 23.45/0.7635 | 28.28/0.7277 | 24.95/0.8033 | 28.68/0.7031 | 28.10/0.7878 | 27.67/0.8276 | 27.47/0.6713 | 26.90/0.6864 | 27.43/0.8511 | 26.75/0.7455 |
| | ZS-N2S | 19.29/0.4409 | 21.36/0.5393 | 21.65/0.6879 | 20.44/0.4223 | 16.56/0.4988 | 25.41/0.6417 | 15.88/0.5870 | 25.29/0.8037 | 19.35/0.6591 | 21.09/0.6962 | 24.71/0.6801 | 22.47/0.7677 | 22.00/0.7806 | 21.19/0.6312 |
| | S2S | 17.40/0.5115 | 24.01/0.7441 | 18.56/0.6458 | 22.99/0.6967 | 15.48/0.5679 | 25.07/0.6826 | 19.74/0.7193 | 15.06/0.7385 | 26.12/0.7550 | 18.86/0.8094 | 23.26/0.6999 | 12.85/0.6302 | 22.89/0.8084 | 20.18/0.6927 |
| | ZS-NCD | 25.23/0.6145 | 29.09/0.8189 | 27.60/0.7969 | 27.66/0.8053 | 24.50/0.8223 | 28.31/0.6908 | 25.65/0.8231 | 30.99/0.8571 | 27.21/0.7491 | 27.82/0.8309 | 28.40/0.7464 | 27.98/0.7490 | 28.42/0.8712 | 27.60/0.7827 |
| | **Lottery Prior** | 25.50/0.6398 | 30.12/0.8641 | 27.68/0.7852 | 27.78/0.7689 | 24.90/0.8432 | 28.82/0.6810 | 26.51/0.8242 | 28.84/0.8894 | 28.04/0.7571 | 29.03/0.9118 | 29.61/0.7697 | 26.90/0.9220 | 28.83/0.8586 | 27.89/0.8088 |

*Table 19.* Kodak24 Denoising performance comparison under AWGN denoising (PSNR and SSIM).

| Method (σ) | 01 | 02 | 03 | 04 | 05 | 06 | 07 | 08 | 09 | 10 | 11 | 12 | 13 | 14 | 15 | 16 | 17 | 18 | 19 | 20 | 21 | 22 | 23 | 24 | Average |
|---|---|---|---|---|---|---|---|---|---|---|---|---|---|---|---|---|---|---|---|---|---|---|---|---|---|
| JPEG2K (15) | 25.20/0.7120 | 29.23/0.6996 | 29.31/0.7678 | 29.09/0.7299 | 25.55/0.7728 | 26.71/0.7343 | 28.97/0.7880 | 25.24/0.7730 | 29.29/0.7756 | 29.27/0.7540 | 27.34/0.7033 | 29.28/0.7219 | 24.73/0.7477 | 26.65/0.7137 | 29.23/0.7490 | 28.29/0.7205 | 28.46/0.7709 | 26.54/0.7195 | 27.76/0.7400 | 29.28/0.8055 | 27.21/0.7749 | 27.83/0.6983 | 30.32/0.8039 | 26.38/0.7215 | 27.46/0.7457 |
| BM3D (15) | 29.46/0.8549 | 33.06/0.8266 | 35.19/0.9096 | 33.54/0.8583 | 30.37/0.9023 | 31.00/0.8664 | 34.59/0.9384 | 34.38/0.8952 | 31.70/0.8475 | 34.72/0.8611 | 32.79/0.8203 | 32.58/0.8419 | 30.60/0.8425 | 33.40/0.8781 | 32.69/0.8647 | 33.40/0.8898 | 31.18/0.9085 | 34.85/0.8558 | 32.18/0.8568 | 33.84/0.8936 | 31.46/0.8932 | 31.60/0.8355 | 35.90/0.9205 | 30.83/0.8800 | 32.37/0.8754 |
| DIP (15) | 26.10/0.7586 | 29.56/0.7297 | 30.98/0.8323 | 30.11/0.8023 | 26.17/0.8114 | 27.28/0.7715 | 30.92/0.8905 | 24.65/0.7766 | 31.54/0.8465 | 30.58/0.8890 | 28.45/0.7793 | 32.09/0.8177 | 23.42/0.6814 | 27.77/0.7891 | 29.10/0.7730 | 29.10/0.7730 | 30.93/0.8688 | 27.14/0.7826 | 28.25/0.7786 | 28.87/0.8349 | 28.03/0.8158 | 29.23/0.7820 | 31.88/0.8775 | 25.72/0.7853 | 28.71/0.8016 |
| DD (15) | 23.52/0.6317 | 27.46/0.6668 | 28.19/0.7644 | 26.64/0.6408 | 24.10/0.7155 | 24.10/0.7155 | 30.36/0.8890 | 31.96/0.8826 | 33.51/0.8872 | 30.79/0.9045 | 33.75/0.8532 | 33.48/0.8290 | 28.98/0.8885 | 31.14/0.8681 | 32.96/0.8401 | 33.23/0.8687 | 33.21/0.8743 | 31.93/0.7840 | 32.64/0.7855 | 32.66/0.8113 | 29.32/0.8045 | 27.16/0.7677 | 31.50/0.8850 | 32.54/0.7926 | 30.72/0.8016 |
| ZS-N2N (15) | 31.13/0.8951 | 33.19/0.8418 | 34.04/0.8582 | 32.93/0.8463 | 30.38/0.8890 | 31.96/0.8826 | 33.51/0.8872 | 33.75/0.8532 | 33.48/0.8290 | 28.98/0.8885 | 31.14/0.8681 | 32.96/0.8401 | 23.50/0.6967 | 27.77/0.7891 | 29.12/0.7789 | 30.93/0.8688 | 27.14/0.7826 | 28.25/0.7786 | 27.89/0.8158 | 29.23/0.7820 | 31.88/0.8775 | 29.29/0.7803 | 29.43/0.8416 | 31.83/0.8416 | 30.44/0.8582 |
| ZS-N2S (15) | 18.03/0.4389 | 25.76/0.6821 | 17.06/0.6127 | 25.30/0.6915 | 18.51/0.5545 | 16.16/0.3461 | 22.51/0.7272 | 17.54/0.5125 | 15.42/0.6011 | 19.79/0.6228 | 23.15/0.6176 | 8.06/0.2929 | 19.24/0.5868 | 22.86/0.6371 | 23.91/0.7191 | 23.05/0.6114 | 16.88/0.5174 | 14.44/0.5637 | 11.59/0.5257 | 5.47/0.0930 | 22.70/0.6687 | 20.22/0.5615 | 22.13/0.7441 | 17.36/0.5761 | 18.68/0.5540 |
| S2S (15) | 25.73/0.7941 | 25.01/0.7193 | 25.30/0.8177 | 27.41/0.8106 | 22.73/0.7274 | 19.32/0.7144 | 29.75/0.9184 | 19.82/0.7318 | 29.66/0.9927 | 25.72/0.8233 | 22.11/0.7244 | 22.15/0.7885 | 19.75/0.5547 | 24.55/0.7461 | 17.09/0.7066 | 17.15/0.8337 | 21.30/0.7254 | 24.15/0.7016 | 28.50/0.7811 | 10.63/0.6711 | 23.48/0.8304 | 23.60/0.7693 | 22.17/0.8469 | 18.37/0.7790 | 23.08/0.7695 |
| ZS-NCD (15) | 31.28/0.9059 | 33.93/0.8669 | 35.61/0.9215 | 34.19/0.8736 | 31.63/0.9286 | 32.07/0.8909 | 35.34/0.9382 | 31.72/0.8957 | 35.03/0.9088 | 34.71/0.9097 | 32.40/0.8895 | 34.89/0.8939 | 28.61/0.8931 | 31.95/0.8978 | 34.04/0.9004 | 34.04/0.9004 | 31.25/0.8860 | 33.26/0.8956 | 34.13/0.9096 | 32.45/0.8973 | 32.52/0.8767 | 27.45/0.8440 | 33.18/0.9026 | 33.18/0.9026 | 33.18/0.9026 |
| **Lottery Prior (15)** | 32.16/0.9130 | 34.85/0.8746 | 37.00/0.9323 | 35.02/0.8929 | 32.92/0.9349 | 33.17/0.9110 | 36.78/0.9550 | 32.43/0.9250 | 36.48/0.9267 | 36.27/0.9210 | 33.69/0.8882 | 35.62/0.8958 | 30.62/0.9053 | 32.91/0.8932 | 34.93/0.9029 | 34.79/0.9081 | 35.33/0.9150 | 32.71/0.8969 | 34.16/0.9058 | 33.39/0.9247 | 33.90/0.9216 | 33.44/0.8854 | 37.12/0.9308 | 33.21/0.9223 | 34.29/0.9118 |
| BM3D (25) | 26.98/0.7554 | 31.29/0.7717 | 32.74/0.8618 | 31.23/0.7994 | 27.56/0.8236 | 28.42/0.7789 | 31.97/0.9099 | 31.87/0.9026 | 27.74/0.8497 | 30.77/0.8496 | 26.51/0.8099 | 30.14/0.7659 | 28.28/0.7784 | 27.77/0.7115 | 30.33/0.7813 | 31.03/0.8382 | 27.82/0.7616 | 30.96/0.8366 | 30.17/0.7561 | 30.99/0.8077 | 28.01/0.6818 | 26.63/0.7666 | 32.76/0.8588 | 28.09/0.7966 | 29.58/0.8092 |
| DIP (25) | 25.26/0.7193 | 29.52/0.7298 | 29.58/0.7641 | 28.73/0.7327 | 25.97/0.7902 | 26.42/0.7245 | 29.27/0.8238 | 23.99/0.7493 | 29.50/0.7625 | 29.41/0.7732 | 27.37/0.7233 | 30.11/0.7467 | 22.97/0.6570 | 26.79/0.7397 | 29.90/0.7552 | 28.17/0.6514 | 29.63/0.8146 | 26.21/0.7263 | 26.90/0.7611 | 28.00/0.7185 | 26.97/0.7707 | 27.05/0.8093 | 29.49/0.8093 | 25.12/0.7364 | 27.62/0.7496 |
| DD (25) | 28.30/0.8249 | 30.67/0.7618 | 31.10/0.7676 | 30.28/0.7562 | 27.66/0.8317 | 29.01/0.7975 | 30.48/0.8057 | 27.82/0.8385 | 30.53/0.9088 | 30.24/0.7380 | 29.11/0.7645 | 30.69/0.7276 | 26.86/0.8113 | 28.32/0.7801 | 30.79/0.7901 | 30.46/0.7826 | 30.60/0.8466 | 28.07/0.7713 | 29.76/0.7902 | 30.95/0.8091 | 29.09/0.7451 | 30.98/0.7461 | 27.99/0.7707 | 29.54/0.7998 | 29.56/0.7798 |
| ZS-N2N (25) | 17.06/0.4493 | 24.96/0.6601 | 24.96/0.7601 | 24.70/0.6776 | 21.31/0.8206 | 20.89/0.6756 | 16.54/0.5188 | 20.46/0.6864 | 22.14/0.7248 | 23.98/0.6449 | 21.05/0.8047 | 24.51/0.7718 | 14.01/0.6122 | 23.08/0.7725 | 16.12/0.4638 | 18.55/0.6204 | 17.22/0.7635 | 24.10/0.7925 | 25.32/0.7214 | 19.03/0.6009 | 19.45/0.8093 | 29.09/0.7451 | 30.98/0.7461 | 30.22/0.7609 | 21.50/0.7198 |
| ZS-N2S (25) | 23.60/0.7118 | 20.04/0.6040 | 21.16/0.6932 | 23.86/0.7267 | 19.02/0.5867 | 17.49/0.7154 | 23.04/0.8244 | 17.92/0.6895 | 17.60/0.4487 | 24.39/0.8275 | 19.65/0.6347 | 21.04/0.7575 | 18.72/0.6112 | 22.24/0.7802 | 15.36/0.6307 | 25.10/0.6566 | 17.56/0.6245 | 20.93/0.5421 | 25.32/0.7214 | 10.00/0.6007 | 22.32/0.7695 | 22.64/0.6994 | 20.52/0.8077 | 17.67/0.7054 | 20.72/0.6949 |
| S2S (25) | 28.88/0.8364 | 31.87/0.7865 | 32.78/0.8226 | 30.38/0.7368 | 28.81/0.8615 | 29.58/0.8146 | 32.01/0.8524 | 28.87/0.8661 | 32.17/0.8171 | 32.38/0.8086 | 30.03/0.7961 | 32.64/0.8248 | 26.88/0.8315 | 29.50/0.8210 | 31.75/0.7962 | 31.14/0.7939 | 31.33/0.8108 | 29.02/0.8012 | 30.71/0.8008 | 31.77/0.8283 | 29.79/0.7852 | 25.67/0.8395 | 28.09/0.8789 | 30.79/0.8120 | 30.60/0.8144 |
| ZS-NCD (25) | | | | | | | | | | | | | | | | | | | | | | | | | |
| **Lottery Prior (25)** | 29.15/0.8358 | 32.60/0.8219 | 34.36/0.8998 | 32.53/0.8380 | 29.73/0.8787 | 30.13/0.8548 | 33.92/0.9293 | 29.41/0.8804 | 34.04/0.9001 | 33.82/0.8874 | 31.00/0.8225 | 33.27/0.8504 | 27.56/0.8013 | 30.05/0.8145 | 31.79/0.8479 | 32.18/0.8456 | 32.99/0.8869 | 29.80/0.8358 | 31.48/0.8387 | 29.76/0.8786 | 31.00/0.8789 | 30.79/0.8120 | 34.62/0.9070 | 31.48/0.8579 | 31.48/0.8579 |

*Table 20.* Kodak24 Denoising performance under Poisson noise (PSNR and SSIM).

| Method (α) | 01 | 02 | 03 | 04 | 05 | 06 | 07 | 08 | 09 | 10 | 11 | 12 | 13 | 14 | 15 | 16 | 17 | 18 | 19 | 20 | 21 | 22 | 23 | 24 | Average |
|---|---|---|---|---|---|---|---|---|---|---|---|---|---|---|---|---|---|---|---|---|---|---|---|---|---|
| JPEG2K (15) | 20.82/0.4172 | 25.15/0.6187 | 24.68/0.6894 | 23.43/0.5872 | 20.65/0.5202 | 21.66/0.4510 | 23.38/0.5545 | 19.29/0.5129 | 23.08/0.4707 | 23.18/0.4742 | 22.71/0.4834 | 23.41/0.6203 | 19.63/0.3773 | 22.63/0.5025 | 22.96/0.5573 | 24.58/0.5573 | 23.96/0.6000 | 22.59/0.5079 | 22.33/0.4785 | 21.03/0.5747 | 22.33/0.6105 | 22.38/0.5103 | 23.70/0.7252 | 21.32/0.4894 | 22.56/0.5249 |
| BM3D (15) | 24.18/0.5811 | 29.14/0.7007 | 30.21/0.8007 | 28.57/0.7097 | 24.18/0.6079 | 25.36/0.5866 | 28.76/0.8351 | 24.16/0.7193 | 28.38/0.7620 | 26.99/0.6664 | 29.27/0.7134 | 22.06/0.4810 | 25.34/0.5800 | 22.72/0.7134 | 25.54/0.6146 | 28.44/0.7103 | 27.81/0.6942 | 28.94/0.6336 | 26.97/0.6224 | 23.08/0.5985 | 27.02/0.6352 | 30.90/0.8314 | 34.84/0.8154 | 27.04/0.6900 | 26.88/0.6966 |
| DIP (15) | 24.26/0.6524 | 28.16/0.6587 | 28.70/0.7194 | 27.69/0.6423 | 24.37/0.7237 | 25.08/0.6237 | 27.75/0.7301 | 23.07/0.6912 | 27.66/0.6731 | 27.22/0.6660 | 25.20/0.7135 | 25.45/0.6435 | 28.03/0.6902 | 27.81/0.6691 | 27.56/0.7317 | 25.14/0.6761 | 25.90/0.6539 | 27.38/0.7276 | 25.82/0.7276 | 26.11/0.6091 | 23.58/0.7469 | 24.23/0.6551 | 26.37/0.6761 | 26.37/0.6761 | 26.37/0.6761 |
| DD (15) | 23.52/0.6317 | 27.46/0.6668 | 28.19/0.7644 | 26.64/0.6408 | 24.10/0.7155 | 24.21/0.5925 | 26.69/0.7177 | 22.50/0.6769 | 26.61/0.6161 | 26.03/0.6456 | 25.62/0.6349 | 25.96/0.5870 | 21.86/0.6909 | 27.63/0.7418 | 26.54/0.6755 | 26.61/0.6113 | 28.32/0.7801 | 30.79/0.7601 | 27.97/0.7038 | 26.73/0.6736 | 24.09/0.5915 | 27.50/0.6616 | 28.46/0.7573 | 25.43/0.8020 | 26.37/0.6760 |
| ZS-N2N (15) | 25.37/0.7129 | 29.02/0.7162 | 28.48/0.6611 | 27.39/0.6496 | 25.11/0.7421 | 25.89/0.6537 | 27.88/0.6982 | 24.73/0.7357 | 27.61/0.6054 | 27.30/0.5986 | 25.34/0.6973 | 25.92/0.6925 | 28.00/0.6750 | 27.91/0.6068 | 28.73/0.6919 | 26.11/0.7150 | 26.65/0.6696 | 27.83/0.7085 | 26.47/0.6993 | 23.83/0.7001 | 25.60/0.6604 | 27.93/0.6353 | 25.50/0.7063 | 23.86/0.6577 | 26.69/0.6577 |
| ZS-N2S (15) | 18.17/0.4762 | 23.25/0.5907 | 27.16/0.7496 | 24.53/0.6645 | 17.77/0.5489 | 22.42/0.5608 | 25.00/0.7159 | 15.75/0.3956 | 14.63/0.4183 | 22.84/0.6716 | 23.80/0.5989 | 24.71/0.6360 | 19.27/0.4577 | 22.51/0.5822 | 25.06/0.7098 | 26.49/0.6555 | 15.15/0.7713 | 22.36/0.6068 | 27.47/0.6694 | 24.92/0.7338 | 23.38/0.7072 | 23.98/0.6279 | 17.06/0.6786 | 20.40/0.5816 | 22.24/0.6170 |
| S2S (15) | 25.03/0.6917 | 28.20/0.7362 | 24.92/0.8189 | 26.33/0.7688 | 22.75/0.7643 | 17.10/0.6688 | 27.54/0.7357 | 27.61/0.6054 | 22.80/0.5989 | 24.71/0.6360 | 19.27/0.4577 | 18.80/0.7180 | 18.86/0.5559 | 16.07/0.7693 | 24.51/0.7364 | 28.09/0.8149 | 24.31/0.7900 | 25.09/0.7085 | 24.51/0.7085 | 10.74/0.7206 | 24.40/0.7108 | 22.82/0.7762 | 22.17/0.7450 | 18.13/0.6495 | 22.20/0.5780 |
| ZS-NCD (15) | 27.58/0.6860 | 29.16/0.7145 | 29.78/0.7998 | 29.01/0.7387 | 26.21/0.6907 | 29.49/0.8495 | 27.73/0.7407 | 22.91/0.6912 | 23.800/0.5969 | 30.35/0.7690 | 29.00/0.7796 | 27.91/0.6144 | 25.54/0.7812 | 28.23/0.7963 | 30.12/0.7691 | 19.70/0.7541 | 30.19/0.7561 | 25.60/0.6883 | 27.98/0.7466 | 29.15/0.7364 | 29.02/0.7346 | 28.70/0.7297 | 31.16/0.7770 | 27.62/0.7432 | 27.62/0.7432 |
| Lottery Prior (15) | 26.40/0.7115 | 30.53/0.7544 | 31.36/0.8520 | 30.12/0.7701 | 26.82/0.7961 | 26.14/0.7370 | 31.18/0.8886 | 25.00/0.7734 | 30.82/0.8462 | 30.38/0.8242 | 28.52/0.7383 | 29.40/0.7729 | 24.05/0.6223 | 27.48/0.7218 | 27.82/0.8089 | 29.28/0.7419 | 30.39/0.8220 | 27.14/0.7442 | 28.49/0.7597 | 23.97/0.8329 | 27.64/0.8016 | 27.97/0.7096 | 30.80/0.8781 | 25.95/0.7605 | 28.23/0.7778 |
| JPEG2K (25) | 21.37/0.4580 | 25.46/0.5785 | 25.60/0.6174 | 24.96/0.5626 | 22.53/0.5087 | 24.46/0.6331 | 20.53/0.5871 | 24.45/0.5683 | 24.50/0.5692 | 25.64/0.7360 | 23.60/0.4892 | 20.38/0.4892 | 23.64/0.5303 | 24.52/0.6224 | 24.67/0.5549 | 25.02/0.6481 | 23.29/0.5701 | 23.46/0.5558 | 22.91/0.6487 | 25.16/0.5770 | 23.80/0.5007 | 25.79/0.6751 | 22.15/0.5401 | 23.58/0.5680 | 23.58/0.5680 |
| BM3D (25) | 23.34/0.6677 | 26.19/0.6286 | 26.16/0.6566 | 25.34/0.5730 | 24.78/0.7690 | 22.90/0.5408 | 26.12/0.6097 | 29.49/0.6673 | 23.16/0.4315 | 26.49/0.5889 | 25.92/0.7057 | 20.97/0.3004 | 22.77/0.6244 | 26.42/0.6386 | 26.65/0.7228 | 25.18/0.5412 | 34.67/0.7299 | 26.80/0.7692 | 25.40/0.6470 | 26.12/0.6855 | 25.77/0.6556 | 14.16/0.5468 | 24.92/0.6625 | 27.03/0.8189 | 24.90/0.6399 |
| DD (25) | 25.53/0.7181 | 28.94/0.6890 | 29.84/0.7694 | 28.80/0.6975 | 25.70/0.7646 | 25.92/0.7008 | 28.40/0.7418 | 24.30/0.7297 | 28.41/0.7008 | 26.77/0.6669 | 28.89/0.6881 | 23.78/0.6952 | 26.05/0.7228 | 25.15/0.5423 | 28.60/0.7042 | 27.30/0.7327 | 27.24/0.6092 | 24.03/0.7066 | 28.07/0.7099 | 23.80/0.5007 | 23.62/0.7495 | 22.82/0.7762 | 24.57/0.6000 | 26.00/0.5600 | 24.98/0.6806 |
| ZS-N2N (25) | 24.74/0.7730 | 30.22/0.7610 | 29.87/0.7276 | 29.10/0.7617 | 26.99/0.8015 | 27.29/0.7272 | 29.32/0.7582 | 25.78/0.7790 | 29.07/0.7431 | 28.56/0.7174 | 27.07/0.7062 | 27.80/0.7542 | 29.53/0.7294 | 30.10/0.6851 | 29.99/0.8854 | 27.76/0.6866 | 28.26/0.7611 | 25.73/0.7459 | 28.68/0.7499 | 24.74/0.7693 | 24.89/0.7046 | 29.21/0.7026 | 29.43/0.7038 | 27.00/0.7298 | 28.13/0.7274 |
| ZS-N2S (25) | 19.86/0.5123 | 27.15/0.6869 | 27.66/0.7773 | 25.01/0.6794 | 16.99/0.5539 | 22.50/0.6374 | 18.34/0.5867 | 23.41/0.6613 | 23.10/0.6715 | 23.82/0.6477 | 08.29/0.4261 | 18.89/0.4529 | 22.97/0.6329 | 25.70/0.5498 | 25.59/0.7266 | 24.26/0.7312 | 22.07/0.5632 | 20.01/0.6144 | 23.72/0.6161 | 19.75/0.6136 | 23.78/0.6429 | 17.84/0.7240 | 14.82/0.5742 | 21.47/0.6277 | 21.47/0.6277 |
| S2S (25) | 26.06/0.7395 | 29.57/0.7862 | 25.94/0.8400 | 25.01/0.7881 | 13.80/0.7114 | 29.50/0.9017 | 17.50/0.7124 | 27.17/0.8418 | 26.25/0.6609 | 30.35/0.7690 | 29.00/0.7796 | 27.91/0.6144 | 25.54/0.7812 | 28.23/0.7963 | 30.12/0.7691 | 19.70/0.7541 | 26.97/0.7704 | 25.03/0.7026 | 28.50/0.8286 | 10.53/0.7026 | 24.52/0.8090 | 27.17/0.7929 | 31.31/0.8296 | 27.82/0.8012 | 25.75/0.7675 |
| ZS-NCD (25) | 27.28/0.7950 | 28.73/0.7061 | 31.52/0.8338 | 29.67/0.7387 | 28.10/0.8620 | 27.49/0.7226 | 30.51/0.8113 | 26.62/0.8213 | 29.80/0.6909 | 30.35/0.7690 | 27.69/0.7796 | 29.66/0.7809 | 27.40/0.7200 | 24.30/0.5963 | 28.64/0.7629 | 29.48/0.8298 | 30.50/0.7891 | 31.50/0.8457 | 28.45/0.7975 | 29.68/0.7892 | 26.41/0.8479 | 28.31/0.8286 | 28.94/0.7418 | 32.32/0.8919 | 27.46/0.8085 |
| Lottery Prior (25) | 27.45/0.7676 | 31.26/0.7732 | 32.74/0.8787 | 31.23/0.8014 | 28.20/0.8418 | 27.44/0.7755 | 32.35/0.9092 | 26.76/0.8295 | 32.09/0.8705 | 31.80/0.8519 | 29.72/0.7803 | 30.48/0.7949 | 25.38/0.6931 | 28.64/0.7629 | 29.48/0.8298 | 30.50/0.7891 | 31.50/0.8457 | 28.45/0.7975 | 29.68/0.7892 | 26.41/0.8479 | 28.31/0.8286 | 28.94/0.7418 | 32.32/0.8919 | 27.46/0.8085 | 29.55/0.8125 |

