# OpenReview forum: "Lottery Prior: Randomized Neural Compression for Zero-Shot Inverse Problems"
_ICML.cc/2026/Conference — ICML 2026 spotlight_

### Official Review · Reviewer_1528 · 2026-03-07

**Soundness:** 4
**Presentation:** 4
**Significance:** 2
**Originality:** 2
**Overall Recommendation:** 4
**Confidence:** 5

**Summary:**

The paper introduces Lottery Prior, a zero-shot framework for solving linear inverse problems where the degradation operator is known. Drawing on the Lottery Ticket Hypothesis, the authors model the solution as the output of a specific subnetwork identified within a fixed, over-parameterized random network. This subnetwork is discovered by optimizing a binary mask over the frozen weights. Specifically, a continuous score is learned for each weight, and only those with the highest scores are maintained to form the active topology. To enhance the network's expressivity, the optimization process also learns a latent vector that modulates the layers. This latent vector is regularized through an entropy term, calculated via an auto-regressive model, which enforces structural simplicity and prevents the model from overfitting. The authors demonstrate that this process identifies a subnetwork that acts as a powerful architectural prior for inverse tasks. Experimental results across denoising, super-resolution, and inpainting validate that Lottery Prior achieves strong performance compared to existing zero-shot methods while utilizing significantly fewer effective parameters.

**Compliance With Llm Reviewing Policy:**

Affirmed.

**Final Justification:**

This paper provides a well-written and mathematically sound proof-of-concept for applying the Lottery Ticket Hypothesis to zero-shot inverse problems, offering a compelling perspective on architectural priors. However, the rating is tempered by the incremental nature of the core technical framework, the reliance on outdated evaluation datasets, and the lack of comparison against modern diffusion-based baselines. Weighing this interesting conceptual framework against its notable empirical and computational limitations, I recommend a weak accept.

**Key Questions For Authors:**

1. How is the fixed sparsity ratio $r_a$ (the percentage of weights kept in the mask) determined for different tasks? Since more complex images likely require larger subnetworks to be well-represented, can $r_a$ be made adaptive to the input signal?

2. What is the average time and total compute required to solve a single inverse task? How does this latency compare to other competitive zero-shot methods?

3. Can the learned mask ($\tau$) and latent vector ($z$) be reused as a warm-start initialization for different images to accelerate the optimization process?

**Limitations:**

The authors provide a brief limitation section on ill-posed problems, but it lacks a critical discussion of the practical overhead. The framework requires significant time and computation to solve a single task due to the iterative optimization and ensemble-based refinement process. This section should be expanded to discuss these resource requirements against other competitive zero-shot methods.

**Strengths And Weaknesses:**

Strengths
1. The manuscript is well-organized and written with clarity, ensuring all essential context is presented in a readable way. Furthermore, the mathematical logic is straightforward to track and appears to be both robust and well-grounded.

2. The paper successfully adapts the Lottery Ticket Hypothesis to the domain of zero-shot inverse problems. It provides a compelling proof-of-concept that high-quality signal restoration can be derived from the structural topology of a fixed random network rather than the conventional training. It offers a fresh perspective on the role of architectural priors in unsupervised restoration.

Weaknesses
1. The core technical framework, including the subnetwork search, binary masking, and rewind modulation, is largely identical to the previously proposed LotteryCodec. Consequently, the application of these existing mechanisms to inverse problems represents a relatively incremental contribution rather than a fundamental architectural innovation.

2. The experimental validation relies on small, legacy datasets such as Set5, Set11, and Set13, which are no longer representative of modern imaging challenges. To demonstrate true robustness, the authors should evaluate the framework on larger, more diverse benchmarks like DIV2K or Urban100.

3. The paper fails to compare its results against contemporary state-of-the-art zero-shot techniques, particularly those leveraging pre-trained diffusion models (e.g. DPS). Without these comparisons, it is difficult to assess the competitive standing of a lottery-based prior against the current generation of generative solvers.

4. The use of an ensemble-based refinement process, which requires repeating the optimization multiple times to aggregate solutions, significantly inflates the time and compute required per image. This makes the method substantially less practical for real-world deployment

---

> ### Author Rebuttal · Authors · 2026-03-30
>
> We thank the reviewer for the valuable feedback and address the concerns point by point below.
>
> Response to weakness:
>
> - W1. Our goal is **not architectural novelty**, but to introduce a **new compression-based paradigm** and **theoretical framework** for inverse problems. Building on DIP-style untrained priors and LotteryCodec-style randomized compression, we propose a zero-shot inverse solver that leverages compression as a structural prior. To the best of our knowledge, this is **the first work** to systematically formulate inverse problems from a compression perspective beyond denoising. Our key contributions are:
>
>   (1) **Formulation.** While LotteryCodec was designed to compress a given signal as compactly as possible, Lottery Prior inverts the problem: instead of compressing a known signal, it searches for an unknown signal that is simultaneously compressible and consistent with the degraded observation. This inversion from compression-as-objective to compression-as-regularizer is the core conceptual contribution, and it requires new theoretical analysis that has no counterpart in the compression literature
>
>   (2) **Theory.** We derive non-asymptotic error bounds that characterize the rate–distortion–noise trade-off, providing theoretical grounding for the proposed paradigm.
>
>   (3) **Methodology.** We develop a randomized compression-driven framework that optimizes over an implicit codebook. Here, the architectural components serve only as implementations of the compression mechanism. Importantly, our approach does **not explicitly compress a signal**; instead, it searches for a solution that **satisfies both compressibility and data consistency**. While more advanced architectural designs may further enhance performance, they are beyond the scope of this work and left for future study.
>
> - W2. In response, we now provide  **new experiments** for additional dataset (**AFHQ**), MRI modalities (**Table R7**), and JPEG artifact removal task  (**Table R2**). For more details, we refer the reviewer to our response to Reviewer [nU7m] and [dhAe].
>
> - W3. We appreciate the reviewer raising this comparison. Our method operates in a dataset-free setting (no pretrained models), which is fundamentally different from DPS (which requires a pretrained diffusion model). Nevertheless, we agree that such methods provide valuable context. In the revision, we will include DPS in the related work. For more details regarding our setting, please refer to our response [R1–R2] to Reviewer [nU7m].
>
> - W4. We agree that the ensemble-based refinement introduces additional computational overhead. However, this component is **optional and not required** by the core method. In practice, **a single run** already achieves competitive (and for some tasks such as inpainting, state-of-the-art) performance. Moreover, compared to existing zero-shot approaches, our method remains **relatively efficient** due to its lightweight design. We provide detailed runtime and complexity analysis in our response to **[9L3k]**.
>
> Answers to questions:
>
> - Q1. The mask ratio is selected empirically, with $0.5$ yielding the best performance (Fig. 3), as it maximizes the entropy of the randomized network and thus the richness of the solution space. While the ratio can be made per-sample adaptive (e.g., via Gumbel-Softmax trick), we instead treat different ratios as inducing distinct random priors and exploit this diversity via a multi-prior ensemble for improved performance.
>
> - Q2. For detailed runtime and complexity analysis, please refer to **Tables R5–R6** in our response to Reviewer **[9L3k]**. In general, learning a single prior (Set11) takes around 1–2 minutes with satisfactory performance. Combining multiple priors (typically 5–10 minutes for denoising) can further improve performance, while a single run is sufficient for tasks such as inpainting. Additional speedups are expected through engineering optimizations.
>
> - Q3. This is an interesting suggestion that points to a promising direction for future research. We note that, in our current framework, random initialization is by design: the diversity of random initialization is what enables the ensemble. Warm starting from a shared initialization would sacrifice this diversity but could accelerate convergence for individual priors. Potential approaches to learning effective initialization include meta-learning [1] for fast per-instance optimization, or incorporating pretrained modules [2–3] to accelerate the coding process. We note that learning shared initializations typically goes beyond the pure zero-shot setting, and we will clarify this in the revision while leaving it for future work.
>
>     [1] "COIN++: neural compression across modalities." Transactions on Machine Learning Research, 2022.
>
>     [2] "Overfitted image coding at reduced complexity." European Signal Processing Conference, 2024.
>
>     [3] "HyperCool: Reducing Encoding Cost in Overfitted Codecs with Hypernetworks.", 2026

---

> > ### Author Rebuttal · Reviewer_1528 · 2026-04-03
> >
> > I appreciate the authors' detailed response and I have no further concerns.

---

### Official Review · Reviewer_dhAe · 2026-03-10

**Soundness:** 4
**Presentation:** 4
**Significance:** 3
**Originality:** 3
**Overall Recommendation:** 5
**Confidence:** 3

**Summary:**

Lottery Prior introduces a novel, zero-shot solver based on entropy-/complexity-regularized neural networks with a learnable, masked/modulation-based INR backbone. For this, the proposed architecture and learning setups borrow concepts recently introduced in  implicit (compression) codecs, such as Coin++, and merge them with the widely known observations of random, structural priors in the seminal paper Deep Image Prior (DIP). Apart from consistently outperforming benchmarks not only in denoising, but in other inverse problems such as super-resolution and inpainting, the authors formalize error bounds for the proposed inverse solver, adding a theoretical contribution.

**Compliance With Llm Reviewing Policy:**

Affirmed.

**Final Justification:**

While my initial assessment was already strongly positive, the authors have provided a meaningful rebuttal and addressed any remaining  questions or concerns. Further, and despite the initial positive review, I would like to highlight that the authors conducted novel experiments on a new domain, namely MRI reconstruction, which provides insights that Lottery Prior is also of relevance to medical inverse problems. This has, in my opinion, further strengthened the significance of their finding and proposed method, and I hope the authors will include the new insights in the final manuscript. Hence, the rebuttal has reinforced my initial evaluation and I would like to keep my initial scores of 5:accept.

**Key Questions For Authors:**

Q1: In Remark 3.6 l.224-229, the authors state that the codebooks need not be pre-designed or learned from data. Is this common knowledge in related literature? Does this only hold for natural signals, or also for other signals, such as MRI/CT?

Q2: Impact: While the experiments are extensive and appropriately convey the power of the proposed method, I would be very interested in seeing their performance in medical imaging, where degradation operators are highly non-linear and thus more difficult to solve. For instance, in under-sampled k-space (whether or not it is solved in k-space or image-space), or anisotropic super-resolution algorithms. The authors may consider adding an additional super-resolution experiment to contextualize these findings.

Q3: Could the authors add a brief discussion/limitation section on cases where degrading operators are (highly) non-linear? Do their signal-theoretic bounds still hold true in these cases? This is briefly mentioned in l. 272.274, and in limitations, perhaps the authors could give 1-2 examples?

Q4: Regarding l. 1705/1722, I would recommend adding metric names. While these share the structure with the results reported in the main manuscript, this helps people jump to the Appendix.

**Limitations:**

yes

**Strengths And Weaknesses:**

### Strengths

- This paper is clearly one of the sort of papers that is incredibly fun to review, given (1) its potential to change the inverse problem field, and (2) given its high-quality presentation, which nicely conveys its idea, contextualizes the framework within adjacent fields, and appropriately distinguishes its own contribution from the works it builds upon.
- The authors propose both (1) an empirical contribution, i.e., the combination of the regularizers and the implicit codec perspective to the unchallenged DIP-style architecture, and (2) the theoretical contribution regarding signal-theoretic properties of the proposed inverse solver.
- The paper is well written, original in its combination of components and paradigms, and lastly yields empirically very convincing results that lead me to believe that it will be of high significance to the field.


### Weaknesses
- If I have not missed any direct / competing relevant papers, I would say that there aren't any major flaws in the proposed paper. I have some minor concerns that could improve the paper, or could make its practical relevance perhaps even more relevant to other areas. I detail these in the questions for the authors.

---

> ### Author Rebuttal · Authors · 2026-03-31
>
> We sincerely thank the reviewer for the valuable feedback. Next, we address the questions point by point.
>
> - Q1. This observation is **implicitly assumed** in classical compression theory (e.g., [1][2]), where the codebook is treated as a given set, **without constraints** on its construction or requirements to be optimal or learned from external data. However, this point is rarely made explicit, and practical methods typically rely on pre-designed codebooks. Early zero-shot approaches (e.g., ZS-NCD for denoising) partially explore this direction by constructing a training set from the observation itself, but are mainly tailored to denoising and have limited generalizability to broader inverse problems.
>
>   In this work, inspired by untrained priors and INR-based compression, we make this implicit assumption explicit and operational by showing that an implicit neural representation, together with an entropy constraint, can be directly optimized to form a sample- or task-specific “codebook” without any external data. This can be interpreted as overfitting a feasible inverse solution that **satisfies both compressibility and data consistency**.
>
>   This mechanism relies on the signal being **structured or compressible**, and is therefore not restricted to natural images. It can, in principle, extend to domains such as MRI/CT, where similar structural regularities exist.
>
>   [1] Song, Dan, Ayfer Özgür, and Tsachy Weissman. The Performance of Compression-Based Denoisers. 2026
>
>   [2] Weissman, Tsachy, and Erik Ordentlich. The empirical distribution of rate-constrained source codes. 2004
>
> - Q2. We thank the reviewer for this valuable suggestion. To further assess our method in a medical imaging setting, we conduct experiments on fastMRI single-coil knee reconstruction (320x320-cropped Knee-file1000000 as an example) under $4\times$ and $8\times$ acceleration. The undersampled measurements follow $y = M\mathcal{F}(x) + n$, where $\mathcal{F}$ denotes the Fourier transform, $M$ the sampling mask, $n$ the acquisition noise ($n=0$ in our experiments). In this setting, the complex-valued image is parameterized by a single lottery prior and optimized under a $k$-space data-consistency objective with a rate constraint. The PSNR results are reported in **Table R7**, showing that the proposed Lottery Prior can extend to medical imaging inverse problems beyond natural image settings.
>
>   Our method achieves consistent improvement over DIP (+0.3dB at $4\times$, +1.26dB at $8\times$), demonstrating that the compression-based prior generalizes to MRI reconstruction. We note that both data-free approaches (DIP and ours) naturally have a performance ceiling compared to methods that exploit domain-specific training data or imaging physics priors. Bridging this gap through hybrid approaches (e.g., combining Lottery Prior with physics-informed unrolled reconstruction frameworks [1]) is a compelling direction for future work.
>
>   We also evaluated our method on other non-linear degradation problems, such as **JPEG artifact removal** (please refer to  **Table R2** in our response to [nU7m]). These results further support the generality of our framework across diverse and challenging zero-shot inverse problems.
>
>   [1] Huang, Peizhou, et al. "Self-supervised deep unrolled reconstruction using regularization by denoising." IEEE Transactions on Medical Imaging, 2023.
>
>     **Table R7. Experiment over MRI reconstruction with acceleration factors 4 and 8 on fastMRI.**
>
>     | **PSNR performance** | x4 | x8 |
>     |---|---:|---:|
>     | **Deep image prior** | 26.15 | 23.92 |
>     | **Lottery prior** | **26.45** | **25.18** |
>
> - Q3. We will **add a limitation section** as below:
>
>   ``Highly non-linear degradation operators, such as JPEG artifact removal or MRI reconstruction, remain challenging in the pure zero-shot setting. While our method shows initial effectiveness, fully recovering the signal using only compressibility and untrained priors is still difficult. Promising future directions include: (i) incorporating approximate or learned forward operator, (ii) integrating perceptual coding to better exploit the rate–distortion–perception trade-off for improved texture recovery, and (iii) jointly designing iterative or hybrid schemes to balance data consistency and prior regularization.
>
>     In addition, our current theoretical bounds are derived under linear operator assumptions, and extending them to non-linear settings is challenging because the ML estimator no longer reduces to a simple linear least-square projection, and the union-bound over the codebook may not yield a sufficiently tight bound when the likelihood surface is non-convex''
>
> - Q4. We will explicitly include the metric names in the revised manuscript for improved clarity.

---

> > ### Author Rebuttal · Reviewer_dhAe · 2026-04-01
> >
> > First of all, I would like to thank the reviewers for the nicely written rebuttal and the clarifications and replies to my question. I believe the new ablations and the discussion of limitations will add value to the reader in the revised manuscript.
> >
> > Furthermore, I would like to acknowledge the novel experiments on medical data, which confirm the validity of the proposed framework for applications outside of natural image domain. While the PSNR gains are not as pronounced for x4, they are encouraging for x8. To paint the full picture, I would recommend adding these results to the new camera ready version and complement them with perception-based metrics (LPIPS).
> >
> > Lastly, I would like to highlight the author's work in the rebuttal, and stress that my concerns have been resolved. I trust that the authors will incorporate the new aspects in the camera ready version. Given my initial, strongly positive rating (5: Accept), I would like to maintain this score.

---

### Official Review · Reviewer_9L3k · 2026-03-12

**Soundness:** 4
**Presentation:** 4
**Significance:** 4
**Originality:** 4
**Overall Recommendation:** 5
**Confidence:** 2

**Summary:**

This paper proposes a zero-shot inverse solver, dubbed the “Lottery prior,” to solve inverse problems without any training data. The paper builds off of ideas from Deep Image Prior and random networks for implicit codecs. Specifically, the method minimizes a measurement error (called measurement alignment) together with an entropy and complexity regularizers to impose implicit priors and improve restoration performance. The paper derives bounds for compression-based maximum-likelihood inverse solvers. In addition, extensive experiments are presented showing that for tasks such as denoising, super-resolution, and inpainting, the proposed method outperforms existing zero-shot methods such as BM3D and DIP.

**Compliance With Llm Reviewing Policy:**

Affirmed.

**Key Questions For Authors:**

1) Could the authors comment on the computational complexity and runtime of Lottery Prior compared to existing zero-shot methods, like DIP? How much longer does the proposed method take than existing methods?

**Limitations:**

Yes, the paper addresses its limitations, clearly stating that this method does not work as well for ill-conditioned forward operators. This was a question I had when first reading the paper, so it was great to see this point brought up and addressed in the limitations and future work section.

**Strengths And Weaknesses:**

Soundness:
- I did not check the proofs and math, so I cannot comment on the soundness of those.
- The extensive evaluation across a number of simple inverse problems (denoising, super-resolution, and inpainting) is thorough and informative. The paper compares against a number of relevant zero-shot methods, including deep methods like DIP and hand-crafted methods like BM3D. The proposed method does quite well across all of the tasks, and is only occasionally surpassed by BM3D, which is to be expected (BM3D is quite good and hard to beat!). Overall, the evaluation is extensive, thorough, and quite convincing that Lottery Prior improves the restoration quality for zero-shot inverse problems.

Presentation:
- The paper is well written and clearly states the problem. Figures are informative and easy to follow. The tables and results section are presented in a logical and concise way, with copious additional results in the supplement. Overall, the paper seems quite strong!

Significance:
- The paper is significant and has implications for a number of zero-shot inverse problems. Overall, I interpret this paper as a DIP-replacement method that uses randomness to improve the overall zero-shot reconstruction performance. It seems like this will be valuable for many applications where training data is hard to come by, such as scientific applications, including denoising in microscopy and astronomy.

Originality:
- The work provides new insights, proposes a new zero-shot method that surpasses existing methods, has some theoretical insights, and extensive evaluation.

Overall, the paper is quite strong, and I did not see any significant weaknesses.

---

> ### Author Rebuttal · Authors · 2026-03-30
>
> We sincerely thank the reviewer for the valuable question. Following the suggestion, we report the detailed runtime and computational complexity in **Table R5** (evaluated on the first 3 images of Set11 for denoising with $\sigma=15$, mask ratio $0.5$). To ensure fairness across different optimization schedules, we measure GPU runtime and complexity under a fixed budget of 1000 iterations. The backward pass is approximated as twice the forward MACs (following [1]).
>
> [1] Baydin, Atilim Gunes, et al. "Automatic differentiation in machine learning: a survey." Journal of machine learning research (2018).
>
> **Table R5. Runtime and complexity for various zero-shot methods to denoise a Set11 image, on NVIDIA 3090Ti GPU and AMD Ryzen 9 5900X CPU.**
>
> | **Methods** | **Run time (sec/1k steps)** | **Required steps** | **Per-step training complexity (MACs/pixel))** |
> |---|---:|---:|---:|
> | **ZS-NCD** | $96.91$ | 20k | 2350.08k |
> | **DIP** | $22.38$ | 2k-4k | 899.78k |
> | **Lottery prior** | $12.38$ | 5k-10k | 10.37k |
>
> Compared to ZS-NCD, which trains a **0.4M**-parameter network over overlapping patches (officially reported **40 minutes** per image), our method is significantly more efficient due to its lightweight design and avoidance of extensive patch-wise redundancy computation. Compared to DIP, which relies on a heavy multi-scale architecture (**2.2M** parameters), we consider its lightweight variant for fairness (**1–2 minutes**). In contrast, our model is highly compact (**0.7–3.3k** parameters), leading to **substantially lower per-iteration cost and better performance**.
>
> We further analyze performance across optimization steps (reported in **Table R6**, ratio=0.5, random seed: 20,40,60) and observe that satisfactory results are achieved within **5–10k** iterations, while longer runs provide additional but diminishing gains, indicating a favorable time–performance trade-off that **can be flexibly adjusted** based on the available computational budget.
>
> **Table R6. Denoising performance across various zero-shot learning steps.**
>
> | **Learning steps** | 500 | 1000 | 3000 | 5000 | 7000 | 10000 | 20000 | 30000 | 50000 |
> |---|---:|---:|---:|---:|---:|---:|---:|---:|---:|
> | **PSNR** | $31.32$ | $31.75$ | $32.19$ | $32.34$ | $32.39$ | $32.42$ | $32.49$ | $32.55$ | $32.60$ |
> | **SSIM** | $0.8649$ | $0.8710$ | $0.8766$ | $0.8793$ | $0.8798$ | $0.8802$ | $0.8817$ | $0.8825$ | $0.8834$ |
>
> Our framework also supports flexible prior aggregation: combining a few randomized priors (e.g., 5 mask ratios, **$\sim$ 10–20 minutes**) improves denoising, while a single prior (5–10k steps, **$\sim$2 minutes**) suffices for structured tasks such as inpainting, avoiding over-smoothing and achieving state-of-the-art performance.
>
> To summarize: in terms of total wall-clock time for a single image, ZS-NCD requires around $\sim40$ min (20K steps), DIP requires $\sim1-2$ min (2-4K steps), and a single Lottery Prior requires $\sim1-2$ min (5-10K steps at much lower per-step cost). For ensemble refinement with 5 priors, total time increases to $\sim5-10$ min, which remains competitive with ZS-NCD while achieving superior performance.

---

> > ### Author Rebuttal · Reviewer_9L3k · 2026-04-01
> >
> > The authors have provided a detailed and helpful response regarding my runtime question. I have no further concerns.

---

### Official Review · Reviewer_nU7m · 2026-03-12

**Soundness:** 3
**Presentation:** 3
**Significance:** 3
**Originality:** 3
**Overall Recommendation:** 5
**Confidence:** 4

**Summary:**

This paper presents a novel zero-shot image restoration method based on an implicit regularisation, which combines elements of bias from random neural representations a la deep image prior with a lightweight architecture, with explicit entropy/complexity regularisation. The proposed approach is studied in detail, theoretically and empirically through a range of prototypical image restoration tasks and comparisons with alternative approaches from the literature, where it performs well.

**Compliance With Llm Reviewing Policy:**

Affirmed.

**Final Justification:**

I thank the authors for their constructive engagement during the rebuttal period and for their thorough responses. After carefully reconsidering my original review and our subsequent discussion, I believe the main concerns have been adequately addressed. I therefore update my score to 5: accept, reflecting my revised overall assessment.

**Key Questions For Authors:**

How does the performance of the propose approach compare to state-of-the-art methods such as LATINO-PRO, Reconstruct Anything Model, and DPIR on a more challenging dataset such as FFHQ at resolution 1024x1024 or AFHQ at resolution 512x512?

How would the proposed approach perform on a non-linear problem such as noisy JPEG artefact removal or phase retrieval?

How sensitive are the obtained solutions w.r.t. to the choice of the optimiser and the choice of the network architecture?

**Limitations:**

Yes

**Strengths And Weaknesses:**

I find the paper well structured, clearly written and easy to follow. The presentation of prior work is broadly comprehensive, although there are some gaps that would need to be addressed. In particular, it would be important to also discuss recent developments in zero-shot inverse solvers based on foundation models. This should include both deep generative foundation models deployed within zero-shot inverse solvers, as well as foundation models developed specifically for image restoration. As an example of a state-of-the-art inverse solver based on a generative diffusion model, there is the LATINO-PRO solver (https://latino-pro.github.io) based on a Stable Diffusion XL prior with automatic prompt optimisation, see references therein. As an example of a foundational models developed specifically for image restoration, one could mention the Reconstruct Anything Model (https://matthieutrs.github.io/ram-page/), which can be deployed zero-shot or fine-tuned in a fully self-supervised manner from a single measurement. Of course, there many inverse solvers and models have been proposed in the literature. The literature on plug-and-play image restoration methods with state-of-the-art denoisers should also be discussed in far more detail (methods such as, e.g., DPIR https://github.com/cszn/DPIR).

With regard to soundness, I am of the view that the paper could be significantly strengthened by performing experiments with more modern image datasets and by including a wider range of methods that represent other approaches to zero-shot image restoration, even if these rely on much larger foundation models that could potentially perform better at a higher computational cost. For example, the authors could report comparisons with LATINO-PRO, Reconstruct Anything Model, and DPIR. I would also suggest including other metrics, such as LPIPS.

I find the proposed method timely and original. While the proposed implicit regularisation builds on established ideas, I find their combination novel and insightful.

---

> ### Author Rebuttal · Authors · 2026-03-30
>
> We sincerely thank the reviewer for the valuable feedback. In response, we will revise the paper by (a) clarifying the dataset-free zero-shot setting; (b) adding experiments and datasets, (c)  including perceptual metrics for evaluation; and (d) adding ablations on robustness.
>
> Next, we address the commented weakness and the questions point by point.
>
> - R1. We thank the reviewer for raising this important distinction. In our paper, “zero-shot” refers to an **instance-level, dataset-free** setting, where the inverse problem is solved from a single degraded observation **without any external training data or pretrained source knowledge** (as in DIP/ZS-NCD). This stands in contrast to recent foundation or generative model-based approaches that rely on large-scale pretrained priors. Dataset-free, instance-level solvers like ours are **particularly valuable** in domains where training data is scarce, mismatched, or nonexistent; for example, in exploratory scientific imaging where the goal is to image phenomena that have never been previously observed.
>
>     We recognize that the term `zero-shot' is also used in the literature for methods that apply pretrained models to unseen degradation operators without task-specific fine-tuning. To avoid ambiguity, we will clarify this terminological distinction in the revised paper and explicitly discuss both paradigms in the related work.
>
> - R2. Considering our instance-level, dataset-free setting, approaches that rely on externally pretrained models or learned priors  (e.g., LATINO-PRO, RAM, DPIR) are **not directly comparable** to our method. Nevertheless, we agree that these methods provide valuable context. In the revision, we will include them in Related Work. LPIPS results are reported in **Table R1**. Please note that, in our experiments, the proposed method already demonstrates leading performance in all three metrics, and in some cases, surpasses even strong supervised priors.
>
> **Table R1 Average PSNR / MS-SSIM/ LPIPS on Set5 ($\sigma=15$).**
> |Noisy Super-resolution|4x SR| 8x SR|
> |-|-|-|
> |Methods|PSNR/MS-SSIM/LPIPS|PSNR/MS-SSIM/LPIPS|
> |No prior|20.10/0.7125/0.5682 |18.66/0.6102/0.6037|
> |Bicubic| 23.32/0.8129/0.6129 |20.91/0.7158/0.7433|
> |BM3D+Bicubic|24.99/0.9078/0.4316|21.62/0.7795/0.6296|
> |Deep image prior|21.37/0.7610/0.5064|19.55/0.6460/0.6443|
> |**Lottery prior**|**25.87/0.9182/0.3042**|**22.07/0.7951/0.5180**|
> |LapSRN|24.21/0.8465/0.4158|21.30/0.7418/0.5787|
> |BM3D+LapSRN|25.17/0.9096/0.3392|21.66/0.7803/0.5189|
> Answers to questions:
> - Q1-Q2. Following the suggestion, we further evaluated our method on **AFHQ** (using the first image from each class) for the **JPEG artifact removal** task, with results reported in **Table R2**. The experiments show that our framework is **not restricted to linear operators** and **demonstrates strong generality**, consistently outperforming DIP under this setting (achieving a consistent +1.4dB PSNR improvement over DIP across all AFHQ categories).
>
>     We note that zero-shot JPEG artifact removal is **particularly challenging** due to its non-linear quantization-based degradation; nevertheless, the compression-based prior proves effective even in this regime. Further performance gain through iterative schemes or perceptual objective constitue a promising direction for future work.
>
> **Table R2 Noisy JPEG artifact removal experiment ($Q=5, \sigma=25$).**
> |PSNR/MS-SSIM|Cat|Dog|Wild|Avg|
> |-|-|-|-|-|
> |Deep image prior|25.17/0.8122|24.09/0.7728|23.13/0.7733|24.13/0.7861|
> |**Lottery prior**|**26.62/0.8807**|**25.27/0.820**|**24.73/0.8142**|**25.54/0.8323**|
>
> **Table R3 Ablation over different optimizers**
> |Optimizer|**Adam**|AdamW|RMSprop|
> |-|-|-|-|
> |PSNR(dB)|**32.60**|32.59|32.48|
> |SSIM|**0.8834**|0.8828|0.8792|
>
> **Table R4 Ablation over the width of the random network.**
> |Width of each layer|x0.5| x1|x2|**Combine Priors**|
> |-|-|-|-|-|
> |PSNR(dB)|32.54|32.60|32.59|**32.90**|
> |SSIM|0.8826|0.8834|0.8820|**0.8872**|
>
> - Q3. We conducted ablations on optimizer choice and architectural configurations to assess sensitivity, using the first three of Set11 for denoising ($\sigma$=15, mask ratio 0.5).
>
>     Performance across different optimizers (**Table R3**) varies by only 0.12 dB, with only minor differences in convergence speed, indicating robustness; we thus adopt the default Adam optimizer.
>
>     Varying backbone width (**Table R4**) shows that narrower models slightly underfit, while wider ones bring marginal gains or overfit (with only 0.06dB variation), indicating that the rate constraint, not specific architecture, is the primary regularizer, consistent with the minimum description length principle and our theoretical framework.
>
>     In practice, a moderate model size is preferred. We want to highlight that under a randomized network setting, different architectures can **induce a family of random codec priors**, judiciously combining multiple such priors may further improve performance, which is left for future work.

---

> > ### Author Rebuttal · Reviewer_nU7m · 2026-04-03
> >
> > I thank the reviewers for their detailed reply and additional metrics. I also appreciate that the focus herein is to advance methodology for problems where data training data is scarce or non-existent, such as exploratory scientific imaging. Having said that, I am of the view that modern foundation models such as RAM, which are trained on very large classes of images, are also suitable for such problems, especially because they can be fine-tuned in a fully self-supervised manner directly from the measurement data. To illustrate this point, the authors of RAM show that achieves good zero-shot performance in electron microscopy imaging. In opinion, the paper could be strengthen by showing comparisons with RAM, or an equivalent foundation model for computational imaging that has been validated in a scientific imaging context, with self-supervised fine-tuning.
> >
> > I keep my score, 4: Weak accept.

---

> > > ### Author Response · Authors · 2026-04-07
> > >
> > > We thank the reviewer for this valuable suggestion. We agree that modern foundation models are a promising direction for inverse problems, as they can exploit large-scale pretraining and then be fine-tuned in a fully self-supervised manner from the measurements. Although they require substantial external data and computational resources, they offer strong adaptability across unseen modalities and tasks.
> > >
> > > Accordingly, we include additional comparisons with a representative foundation model, RAM [1]. We evaluate on Mouse Nuclei fluorescence **microscopy denoising** [2] (**Table RR1**) and natural-image **demosaicing** (**Table RR2**), both **unseen** during RAM’s pretraining. For fine-tuning, RAM uses an additional N=10 measurements, following Table 3 of [1].
> > >
> > > Our experiments show that RAM achieves strong performance without fine-tuning across unseen tasks and modalities. In particular, it is competitive with BM3D on unseen-modality denoising and also achieves strong results on the unseen task of demosaicing, benefiting from its incorporation of forward-operator knowledge and extensive pretraining across diverse modalities and restoration tasks. In contrast, Lottery Prior **does not rely on any external data**, yet still achieves strong performance on both tasks, outperforming pretrained RAM by an average of 0.85 dB on microscopy denoising (when  $\sigma=25$ ) and 0.84 dB on demosaicing, which validates the generality of the proposed method.
> > >
> > > At the same time, RAM improves substantially after few-shot self-supervised fine-tuning (N=10), highlighting the advantage of pretrained priors combined with self-supervised fine-tuning. This setting is, however, **not directly comparable to ours**, since RAM relies on extensive supervised pretraining across diverse datasets and tasks, and is further adapted using additional target-domain measurements. We therefore present these results to **calibrate our method against pretrained and finetuned foundation-model priors, highlighting the different assumptions in prior knowledge and training resources**.
> > >
> > > We believe these additional experiments further strengthen the paper, and we will incorporate them, together with JPEG artifact removal and MRI results [3], into the revised manuscript. We thank the reviewer for the helpful suggestion and hope these additions address the concerns.
> > >
> > > [1] Terris, Matthieu, et al. Reconstruct Anything Model a lightweight general model for computational imaging. ICLR.
> > >
> > > [2] Buchholz, Tim-Oliver, et al. DenoiSeg: joint denoising and segmentation. ECCV.
> > >
> > > [3] Zbontar, Jure, et al. fastMRI: An open dataset and benchmarks for accelerated MRI.
> > >
> > > **Table RR1.** Modality adaptation experiments of the foundation models and Lottery Prior on Mouse Nuclei fluorescence microscopy image [2] denoising (first 5 samples from [2] for validation, and randomly sampled 10 from remaining as a fine-tuning set). Reported values are the average PSNR, and Lottery Prior is averaged with 3 priors, each with 10k steps.
> > >
> > > |Noise level|BM3D|Lottery Prior| RAM (Pretrained) | RAM (Finetuned N=10) |
> > > |--- |---|----|---|---|
> > > | $\sigma = 15$ | 33.91 | **34.13** | 33.89 | 34.98 |
> > > | $\sigma = 25$ | 29.85 | **30.42** | 29.57 | 29.91 |
> > >
> > > **Table RR2.** Task adaptation experiments of RAM and Lottery Prior on **demosaicing** tasks, where the models are evaluated on classic samples (randomly selected from Set5/13 for testing; remaining samples used for RAM finetuning). RAM is pretrained on the LSDIR dataset (**84,991 high-quality images**) across **6 inverse tasks** (**such as deblur or inpainting**)  and then fine-tuned in a self-supervised manner (**10 measurements**), while Lottery Prior is trained from a single degradation without external data.  The reported PSNR of Lottery prior here is obtained using a single prior.
> > >
> > > |  |  |  |  |  |  |  |
> > > |---|---:|---:|---:|---:|---:|---:|
> > > | PSNR | Baby | Butterfly | Woman | Coastguard | Foreman | Average |
> > > | **Lottery prior** | **33.17** | **31.15** | **32.77** | **31.59** | **32.43** | **32.22** |
> > > | RAM (Pretrained) | 32.58 | 30.77 | 31.86 | 30.63 | 31.08 | 31.38 |
> > > | RAM (Finetuned N=10) | 37.62 | 33.68 | 37.71 | 38.77 | 36.58 | 36.87 |

---

### Decision · Program_Chairs · 2026-04-30

**Decision:**

Accept (spotlight)

**Comment:**

A very interesting approach for solving inverse problems using random networks. All reviewers agreed that this is a strong paper that should be accepted